# TOWARDS BUILDING A GROUP-BASED UNSUPERVISED REPRESENTATION DISENTANGLEMENT FRAMEWORK

**Yang Tao**[1]*, **Xuanchi Ren**[2] , **Yuwang Wang**[3]†, **Wenjun Zeng**[4] , **Nanning Zheng**[1]
[1]Xi'an Jiaotong University, [2]HKUST, , [3]Microsoft Research Asia, [4]EIT

## ABSTRACT

Disentangled representation learning is one of the major goals of deep learning, and is a key step for achieving explainable and generalizable models. A well-defined theoretical guarantee still lacks for the VAE-based unsupervised methods, which are a set of popular methods to achieve unsupervised disentanglement. The Group Theory based definition of representation disentanglement mathematically connects the data transformations to the representations using the formalism of *group*. In this paper, built on the group-based definition and inspired by the *n-th dihedral group*, we first propose a theoretical framework towards achieving *unsupervised* representation disentanglement. We then propose a model, based on existing VAE-based methods, to tackle the unsupervised learning problem of the framework. In the theoretical framework, we prove three sufficient conditions on model, group structure, and data respectively in an effort to achieve, in an unsupervised way, disentangled representation per group-based definition. With the first two of the conditions satisfied and a necessary condition derived for the third one, we offer additional constraints, from the perspective of the group-based definition, for the existing VAE-based models. Experimentally, we train 1800 models covering the most prominent VAE-based methods on five datasets to verify the effectiveness of our theoretical framework. Compared to the original VAE-based methods, these *Groupified* VAEs consistently achieve better mean performance with smaller variances.

## 1 INTRODUCTION

Learning independent and semantic representations of which individual dimension has interpretable meaning, usually referred to as disentangled representations learning, is critical for artificial intelligence research (Bengio et al., 2013). Such disentangled representations are useful for many tasks: domain adaptation (Li et al., 2019; Zou et al., 2020), zero-shot learning (Lake et al., 2017), and adversarial attacks (Alemi et al., 2016), etc. Intuitively, a disentangled representation should reflect the factors of variations behind the observed data of the world, and one latent unit is only sensitive to changes of an individual factor.

Due to the facts that obtaining the ground-truth labels requires significant human effort and humans can learn those factors unsupervisedly, unsupervised representation disentanglement draws much attention from researchers recently. A lot of methods are proposed base on some intuitions. Most of the state-of-the-art methods (Higgins et al., 2017; Burgess et al., 2018; Kim & Mnih, 2018; Chen et al., 2018; Kumar et al., 2017) are based on Variational Autoencoder (VAE) (Kingma & Welling, 2013). These methods are *fully unsupervised* and can be applied to a variety of complex datasets (Lee et al., 2020). However, these methods suffer from the unidentifiability problem (Locatello et al., 2019b) due to a lack of theoretical guarantee. Another stream of works (Chen et al., 2016; Lin et al., 2020; Khrulkov et al., 2021; Lee et al., 2020) leverage generative adversarial network (GAN) (Goodfellow et al., 2014) to achieve disentanglement but are not interpretable. In general, a well-defined theoretical guarantee is needed for those methods.

The research of symmetry in physics demonstrates that infinitesimal transformations that conform to some symmetry groups on physical objects can reflect their nature (Anderson, 1972; Noether,

---

*Work done during internships at Microsoft Research Asia.
†Corresponding author

1915). Recently, inspired by this research on symmetry, Higgins et al. (2018) proposed a group-based definition of disentangled representation. They argue that the symmetries, i.e., the transformations that change certain aspects of data and keep other aspects unchanged, ideally reflect the underlying data structure. The group-based definition is a formal and rigorous mathematical definition of faithful and, ideally, interpretable representation of the generative factors of data, which is widely accepted (Greff et al., 2019; Mathieu et al., 2019; Khemakhem et al., 2020). Subsequently, due to the fact that the definition is defined by the world state (i.e., Ground Truth) and based on the assumption (Caselles-Dupré et al., 2019) that this definition should be useful for downstream tasks such as a Reinforcement Learning, Caselles-Dupré et al. (2019), Quessard et al. (2020), Painter et al. (2020) propose environment-based (to provide world state) methods to learn such disentangled representations in Reinforcement Learning settings. These inspire us to ask the following question: how would the definition benefit *unsupervised* representation disentanglement, and how to learn such a disentangled representation conforming to the definition in the setting of *unsupervised* representation learning?

In Group Theory[1], the *n-th dihedral group* (Judson, 2020) is a set of all permutations of polygons vertices, forming a permutation group under the operation of composition (Miller, 1973). The generators in an *n-th dihedral group*, i.e., flip and rotation, can be regarded as the disentangled factors and also transformations. In this paper, inspired by the *n-th dihedral group*, we answer the above questions and address the challenge by proposing a theoretical framework to make the definition practically applicable for *unsupervised* representation disentanglement. We then propose a model to tackle the learning problem of the framework and verify its effectiveness. We theoretically prove in Section 3.2 the three sufficient conditions towards achieving disentangled representation per group-based definition, which are referred to as model, group structure, and data constraint, respectively. With these conditions, we offer additional constraints from the perspective of the definition. The additional constraints encourage existing VAE-based models to satisfy the symmetry requirement that comes from the nature of factors. Finally, we provide a learning model based on the existing VAE-based methods in an effort to fulfill the three conditions (with the model and group structure constraint and a *necessary* condition for the data constraint satisfied). As an intuitive understanding, we introduce the additional constraints to reorganize the latent space to restrict its symmetry in an unsupervised way. These additional constraints indeed narrow down the solution space of VAE-based models. Detailed discussion in Sec. 5.4. Our model consistently achieves statistically better performance in prominent metrics (higher means and lower variances) than corresponding existing VAE-based models on five datasets, demonstrating that the group-based definition together with our proposed framework further encourages disentanglement.

Our main contributions are summarized as follows:

- To our best knowledge, we are the first to provide a theoretical framework to make the formal group-based mathematical definition of disentanglement practically applicable to *unsupervised* representation disentanglement.

- Our theoretical framework provides additional constraints from the perspective of group-based definition for the existing VAE-based methods.

- We propose a learning model of the framework by deriving and integrating additional loss into existing VAE-based models, in an effort to make the learned representation conform to the group-based definition without relying on the environment (as done in Caselles-Dupré et al. (2019); Quessard et al. (2020); Painter et al. (2020)).

## 2 RELATED WORKS

Different definitions have been proposed for disentangled representation (Bengio et al., 2013; Higgins et al., 2018; Suter et al., 2019). However, only the group-based definition proposed by Higgins et al. (2018) focuses on the disentangled representation itself and is mathematically rigorous, which is well accepted (Caselles-Dupré et al., 2019; Quessard et al., 2020; Painter et al., 2020; Diane Bouchacourt, 2021). Nevertheless, Higgins et al. (2018) do not propose a specific learning method based on their definition. Before this rigorous definition was proposed, there had been some success in identifying generative factors in static datasets (without interaction with environment), e.g., $\beta$-VAE (Higgins et al.,

---

[1] We assume some basic familiarity with the fundamentals of Group Theory and Group Representation Theory. Please refer to Appendix A for some basic concepts.

2017), Anneal-VAE (Burgess et al., 2018), $\beta$-TCVAE (Chen et al., 2018), and FactorVAE (Kim & Mnih, 2018). More recent works (Srivastava et al., 2020; Shao et al., 2020; Kim et al., 2019; Lezama, 2018; Rezende & Viola, 2018) also do not consider the group-based definition. Therefore, how group-based definition will facilitate these methods is still an open question. Besides, all these works suffer from the unidentifiability problem (Locatello et al., 2019b), which is a challenging problem in this literature. From group-based definition, our framework points out that, the unidentifiability problem could be solved once the data constraint is satisfied. However, in this work, we can only get a necessary condition for data constraint, and we still can not solve this challenging problem.

As pointed out in Quessard et al. (2020), it is not straightforward to reconcile the probabilistic inference methods with the group-based definition framework. Caselles-Dupré et al. (2019), Quessard et al. (2020), Painter et al. (2020) leverage the interaction with the environment (assuming it is available) as supervision instead of minimizing the total correlation as the VAE-based methods do. Consequently, the effectiveness of these methods is limited to the datasets with the environment available. *Our framework learns a representation conforming to the group-based definition without relying on the environment.* Pfau et al. (2020) propose a non-parametric method to unsupervisedly learn linear disentangled planes in data manifold under a metric. However, as pointed out by the authors, the method does not generalize to held-out data and performs poorly when trying to disentangle directly from pixels.

To summarize, the existing probabilistic inference methods lack theoretical support, while the application scope of existing methods based on the group-based mathematical definition Higgins et al. (2018) is very limited. To the best of our knowledge, *our work is the first to reconcile the probabilistic generative methods with the inherently deterministic group-based definition framework of Higgins et al. (2018).*

## 3 THE GROUP-BASED FRAMEWORK FOR UNSUPERVISED REPRESENTATION DISENTANGLEMENT

Our goal is to explore the benefit of the group-based definition for *unsupervised* representation disentanglement and learn such a disentangled representation. The background of the group-based definition is provided in Section 3.1. Section 3.2 presents the theoretical framework towards achieving *unsupervised* disentanglement, in which we derive three sufficient conditions on the model, group structure, and data, respectively. The conditions on the model and group structure provide additional constraints for the existing VAE-based models.

### 3.1 GROUP-BASED DEFINITION

We briefly review the group-based definition of disentangled representation Higgins et al. (2018). Considering a group $G$ acting on world state space $W$ (can be understood as ground-truth) of data space $O$ and representation space $Z$ via *group action* $\cdot_W$ and *group action* $\cdot_Z$ respectively. For a mapping $f = b \circ h$, where $b$ and $h$ denote the data generative process and encoding, we state: the mapping $f$ is *equivariant* between the actions on $W$ and $Z$ if

$$g \cdot f(w) = f(g \cdot w), \ \forall g \in G, \ \forall w \in W. \tag{1}$$

**Definition 1** *Assume $G$ can be decomposed as $G = G_1 \times G_2 \times \cdots \times G_m$. The set $Z$ is disentangled with respect to $G$ if: $(i)$ group action of $G$ on $Z$ exits. $(ii)$ the mapping $f$ is equivariant between the actions on $W$ and $Z$. $(iii)$ There is a decomposition $Z = Z_1 \times Z_2 \times \cdots \times Z_m$ such that each $Z_i$ is affected only by the corresponding $G_i$.*

It is challenging to apply the group-based definition to an *unsupervised* disentanglement setting in practice because the definition refers to the world state space $W$, the group action of $G$ on $W$, and mapping $b$ which are typically inaccessible in practice. We tackle the challenge by re-framing the definition in a new framework in the following section.

### 3.2 PROPOSED THEORETICAL FRAMEWORK

Since when the representation is disentangled, one latent unit in the representation space is only sensitive to changes of an individual generative factor, we make the following assumptions: $G$

is a direct product of $m$ *cyclic groups* (as suggested by Higgins et al. (2018) and for simplicity): $G = (\mathbb{Z}/n\mathbb{Z})^m = \mathbb{Z}/n\mathbb{Z} \times \mathbb{Z}/n\mathbb{Z} \times \cdots \times \mathbb{Z}/n\mathbb{Z}$, where $n$ is the assumed total number of possible values for a factor and $m$ is the total number of factors; we further assume $Z$ is a set with the same elements in $G$. Therefore, the *group actions* of $G$ on $Z$ can be set to be element-wise addition, i.e., $g \cdot z = \overline{g + z}, \forall z \in Z, g \in G$. For the *generator* of dimension $i$ of $G$, $g_i = (0, \ldots, \overline{1}, \ldots, 0)$, $g_i$ only affects the $i$-th dimension of $z$ by $\overline{g_i + z}$. In addition, the action of each generator $g_i$ on $w$ only affects a single dimension of $w$.

As we can seen from Equation 1 above, the group action is defined on $w$, which is often not accessible, making it difficult to apply the definition in practice. Therefore, for the *unsupervised* setting, we would like to use permutations on the data space $O$ (which only provides data without labels) to substitute the group actions on $W$. Specifically, inspired by the *n-th dihedral group* (Dummit & Foote, 1991), we construct a *permutation group* $\Phi$, serving the role of an "agent" of $G$. The actions of $G$ on $W$ can be performed by $\varphi_g \in \Phi$ on $O$, which can be formulated as

$$f(g \cdot w) = h(\varphi_g \cdot b(w)) = h(\varphi_g \cdot o), \forall w \in W, g \in G, \tag{2}$$

where $o$ denotes the data (e.g., image) corresponding to the world state $w$ through the mapping function $b$. If the above equation holds, we state that the "agent" permutation group $\Phi$ exists. We first give the conditions for the existence of this "agent" permutation group $\Phi$, then derive the additional condition to achieve such disentanglement. We accomplish these two objectives in Theorem 1 with the proof provided in Appendix B. Theorem 1 states that a *general permutation group* $\Phi$ on $O$ can serve as an agent group (*agent group exists*) if and only if both (i) and (ii) are satisfied. If the agent group exists, and its *permutations* (actions on $O$) can be defined by an autoencoder-like model as shown in the equation in (iii), then $Z$ is disentangled with respect to $G$.

**Theorem 1** *For the group $G = (\mathbb{Z}/n\mathbb{Z})^m$, a permutation group $\Phi$ on $O$, a representation space $Z$, a World State space $W$, and mapping $b$ and $h$, Equation 2 holds if and only if* (i) *$\Phi$ is isomorphic to $G$, and* (ii) *For each generator of dimension $i$ of $G$, $g_i$, there exists a $\varphi_i \in \Phi, i = 1, \ldots, m$, such that $\varphi_i \cdot b(w) = b(g_i \cdot w)$, $\forall w \in W$, and $\varphi_i$ is a generator of $\Phi$; Further, if Equation 2 holds and* (iii) *$\varphi_g \cdot b(w) = h^{-1}(g \cdot f(w)) \forall w \in W, \varphi_g \in \Phi$, then $Z$ is disentangled with respect to $G$, where $\varphi_g$ is the corresponding element in $\Phi$ of $g$ under the isomorphism.*

In Theorem 1, (i) states that the relation between the elements (i.e., group structure) is preserved between $\Phi$ and $G$, and we denote it as the *group structure constraint*; (ii) actually indicates a data constraint that all variations in the data can be generated by compositions of some basic *permutation generators* $\{\varphi_i\}_{i=1,\ldots,m}$. We denote it as the *data constraint*; (iii) states that the *permutations* in the agent group $\Phi$ are defined by encoding, action, and decoding, which is referred to as the *model constraint*. Note that in Theorem 1, only the data constraint refers to the world state $w$.

Here is a sketch of the proof: *data constraint* is a special case of Equation 2 for a *generator*, and *group structure constraint* is a relation-preserving constraint on compositions of *generators*, and satisfying both constraints will thus result in that Equation 2 holds for any general element in $\Phi$, and vice versa. Moreover, we can derive Equation 1 for disentanglement when combining the *model constraint* and Equation 2.

The model constraint specifies the way to permute the data. When the data is permuted, its world state changes. Therefore, how the world states transit between each other is modeled by the *model constraint* applied on the data. The isomorphism between $\Phi$ and $G$ ensures that the world state space $W$ and data space $O$ have the same symmetry. In this way, the model applied on the data learns the transition of the world states. Note that we aim to bring this group-based definition, which requires ground truth by default, into the *unsupervised* setting. Now only the data constraint refers to the world states, and it seems almost impossible to derive a sufficient condition for it without the labels. We thus make a trade-off in which we use a necessary condition in the next section.

## 4 GROUPIFIED VAE: A LEARNING METHOD OF THE FRAMEWORK

Let's look closer into the three constraints, respectively. Firstly, we consider the model constraint, $\varphi_g \cdot o = h^{-1}(g \cdot h(o)) \ \forall o \in O, \varphi_g \in \Phi$, which suggests that the action of $\Phi$ on $O$ can be implemented using an autoencoder-like network that performs encoding, action on its representation space, and decoding. Given an autoencoder-like network with an encoder $h$ and a decoder $d$, since $d$

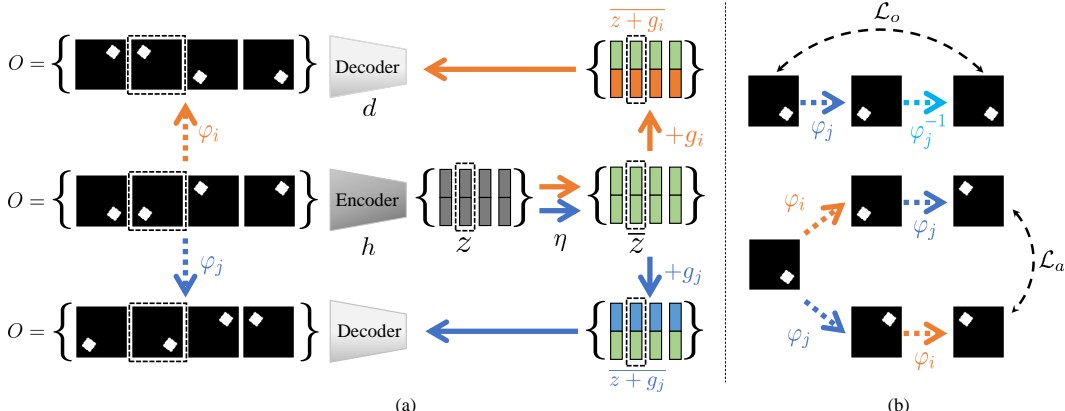

Figure 1: Overview of the implementation (*Groupified* VAE). (a) Illustration of *permutation group* $\Phi = \{\varphi_g | g \in G\}$ defined on a VAE-based model, where $G = (\mathbb{Z}/n\mathbb{Z})^m$. The *generators* $\varphi_i, \varphi_j \in \Phi$ are permutations on $O$. Specifically, when optimized, $\varphi_i$ and $\varphi_j$ are horizontal and vertical movements. $\varphi_i$ is defined as the solid orange arrows illustrate: encode an image $o$ to representation $z$, perform $\eta$ on $z$ to get $\overline{z}$, add $g_i$ to $\overline{z}$, and decode back to the image. This process can be regarded as an exchange of images in dataset (permutation), as the dashed orange arrow shows. These permutations form a group $\Phi$. (b) The Isomorphism Loss, which guarantees that $\Phi$ is isomorphic to $G$, includes Abel Loss $\mathcal{L}_a$ constraining the commutativity, and Order Loss $\mathcal{L}_o$ constraining the cyclicity.

is approximately the inversion of $h$, the *model constraint* can be formulated as

$$\varphi_g \cdot o = h^{-1}(g \cdot h(o)) \triangleq d(g \cdot h(o)), \forall o \in O, g \in G, \tag{3}$$

together with further implementation of $\Phi$ as described in Section 4.1, the *model constraint* can be fulfilled. Secondly, The *data constraint* requires that all variations in the data can be generated by compositions of some basic *permutations generators*. Previous VAE-based works (Higgins et al., 2017; Burgess et al., 2018; Kim & Mnih, 2018; Chen et al., 2018; Kumar et al., 2017) aim to generate the data with independent generative factors, which is in line with the *data constraint*. Intuitively, if the VAE-based model can generate the data from statistical independent basic latent units and each unit corresponds to the basic *permutation generator*, the *data constraint* may be satisfied. Based on the intuition above, we prove that if the world state is independently sampled per dimension, the minimization of total correlation is a necessary condition for the *data constraint* (see Appendix E). Therefore, we can leverage existing VAE-based models to fulfill the *data constraint* to some extent for the *unsupervised* setting. Lastly, to satisfy the *group structure constraint*, we derive a self-supervised *Isomorphism Loss* which can be incorporated into the VAE-based model as described in Section 4.2.

## 4.1 IMPLEMENTATION OF GROUP $\Phi$

The key is to implement the *group actions* of $G$ on $Z$ into the VAE-based models, we need to map the representation $z$ to a group that is isomorphic to $G$ (cyclic representation space). Therefore, we construct a function $\eta$ to achieve this mapping. Moreover, this mapping is required to be differentiable, in order for back-propagation to be adopted for optimization. According to Group Theory, there is an isomorphism between $G$ and the *n-th root unity group*: $\{\exp((2\pi iz)/n)|z \in \mathbb{Z}^m\}$, where $n, m$ are the same as in $G$. Therefore, the representation $z$ can be mapped to $\overline{z}$ by the function $\eta$ as $\overline{z} = \eta(z) = \exp((2\pi iz)/n)$ (see Figure 1 (a)). However, $\overline{z}$ can not be mapped to directly as it has complex numbers, but we can use Euler's formula: $\exp((2\pi iz)/n) = \sin((2\pi z)/n) + i\cos((2\pi z)/n)$ to map $z$ to its real and imaginary part, i.e., vector $\sin((2\pi z)/n)$ and $\cos((2\pi z)/n)$. The two vectors are concatenated and fed to the decoder.

For ease of implementation, the *permutation group* $\Phi$ can be approximately generated by compositions of *generators*, i.e., $\Phi = <\varphi_1, \varphi_2, \ldots, \varphi_m>$. Recall that the *generator* $\varphi_i$ of group $\Phi$ is defined as $\varphi_i \cdot o = d(g_i \cdot h(o)) = d(\overline{h(o) + g_i}), \forall o \in O$, where $g_i$ is *generator* of dimension $i$ in $G$, as shown in Figure 1 (a). For $\varphi_i$, we implement $g_i \cdot h(o)$ by adding 1 (without loss of generality) to the $i$-th dimension of $h(o)$, then make it cyclic by function $\eta$. Similarly, for $\varphi_i^{-1}$, we add the value of $n - 1$.

### 4.2 IMPLEMENTATION OF THE ISOMORPHISM

In this section, to satisfy the *group structure constraint* (isomorphism), we derive two equivalent constraints, which are then converted into an *Isomorphism Loss* $\mathcal{L}_I$. Many groups are uniquely determined by the properties of the *generators*, e.g., group $G = <a, b|a^2 = b^2 = e, ab = ba>$. In addition, since the group $\Phi$ is *isomorphic* to G, $\Phi$ is also expected to be commutative and cyclic. In light of this, we derive two constraints on *generators* that are equivalent to the isomorphism condition, as described in Theorem 2. Please refer to Appendix C for the proof.

**Theorem 2** *The defined permutation group* $\Phi = < \varphi_1, \varphi_2, \ldots, \varphi_m >$ *is isomorphic to* $G = (\mathbb{Z}/n\mathbb{Z})^m$ *if and only if: $(i)$ for $\forall$ generators* $\varphi_i, \varphi_j \in \Phi, 1 \leq i, j \leq m$, *we have* $\varphi_i\varphi_j = \varphi_j\varphi_i$, *and* $(ii)$ $\forall \varphi_i \in \Phi, 1 \leq i \leq m$, *we have* $\varphi_i^n = e$, *where $e$ is the identity element of group $\Phi$.*

The first constraint requires the group $\Phi$ to be an *abelian group* (Judson, 2020). Therefore, we denote it as Abel constraint and the loss derived from it as the Abel Loss $\mathcal{L}_a$. The second is a constraint on the *order* of elements. We thus denote it as the Order constraint and the loss derived from it as the Order Loss $\mathcal{L}_o$. See Appendix F for a more detailed implementation.

**Abel Loss.** For the Abel constraint: $\forall \varphi_i, \varphi_j \in \Phi, 1 \leq i, j \leq m$, we have $\varphi_i\varphi_j = \varphi_j\varphi_i$. We minimize $\|\varphi_i \cdot (\varphi_j \cdot o) - \varphi_j \cdot (\varphi_i \cdot o)\|, \forall o \in O$ to meet the Abel constraint, as shown in Figure 1 (b). The Abel Loss is the sum of the losses of all combinations of two *generators*. Denote the set of combinations as $C = \{(i, j)|1 \leq i, j \leq m\}$. The Abel Loss is defined as follows

$$\mathcal{L}_a = \sum_{o \in O} \sum_{(i,j) \in C} \|\varphi_i \cdot (\varphi_j \cdot o) - \varphi_j \cdot (\varphi_i \cdot o)\|. \tag{4}$$

**Order Loss.** For the Order constraint: $\forall \varphi_i \in \Phi, 1 \leq i \leq m$, we have $\varphi_i^n = e$, where $e$ is the *identity element* in group $\Phi$ (identity mapping). Note that with $n$ times composition of $\varphi_i$, it is difficult for the gradient to back-propagate. We thus use an approximation that uses 2 times of composition instead. When the autoencoder can do the reconstruction well, this approximation holds, see appendix E for details. Similar to Abel Loss, we minimize $\|\varphi_i \cdot (\varphi_i^{n-1} \cdot o) - o\|, \forall o \in O$ to satisfy the Order constraint. The whole process is illustrated in Figure 1 (b). However, the equation is not symmetrical and leads to bias. Therefore, we use the following symmetrical form instead:

$$\mathcal{L}_o = \sum_{o \in O} \sum_{1 \leq i \leq m} \left( \|\varphi_i \cdot (\varphi_i^{n-1} \cdot o) - o\| + \|\varphi_i^{n-1} \cdot (\varphi_i \cdot o) - o\| \right). \tag{5}$$

With the above two loss functions optimized, the isomorphism condition is satisfied, which can be illustrated by Theorem 3. Please refer to Appendix D for the proof.

**Theorem 3** *The following two conditions are equivalent: $(i)$ $\forall \varphi_i, \varphi_j \in \Phi, 1 \leq i, j \leq m$, we have* $\varphi_i\varphi_j = \varphi_j\varphi_i$ *and* $\forall \varphi_i \in \Phi, 1 \leq i \leq m$, *we have* $\varphi_i^n = e$ *$(ii)$ the Abel Loss function (Equation 4) and the Order Loss function (Equation 5) are optimized.*

Since the Abel Loss and Order Loss are equally important for satisfying the isomorphism condition, we assign equal weight to them. Thus, the **Isomorphism Loss** is $\mathcal{L}_I = \mathcal{L}_o + \mathcal{L}_a$. With the implementation of group $\Phi$, the *model constraint* is satisfied. We optimize the Isomorphism Loss to satisfy the *group structure constraint*. To further satisfy the data constraint to some extent as described in Section 4, we leverage VAE-based models and optimize their original loss (that minimizes the total correlation), denoted as $\mathcal{L}_{VAE}$. Therefore, the **Total Loss** is $\mathcal{L} = \mathcal{L}_{VAE} + \gamma_I \mathcal{L}_I$, where $\gamma_I$ is the weight of Isomorphism Loss. We denote the above VAE-based implementation as *Groupified* VAE.

## 5 EXPERIMENTS

We first verify the effectiveness of *Groupified* VAE quantitatively in learning disentangled representations on several datasets and several VAE-based models. Then, we show its effectiveness qualitatively on two typical datasets. After that, we perform a case study on the dSprites dataset to analyze the effectiveness, and conduct ablation studies on the losses and hyperparameters. For the performance comparison of two downstream tasks (abstract reasoning Van Steenkiste et al. (2019) and fairness evaluation Locatello et al. (2019a)), and more comprehensive results, please see Appendix I.

## 5.1 DATASETS AND BASELINE METHODS

To evaluate our method, we consider several datasets: dSprites (Higgins et al., 2017), Shapes3D (Kim & Mnih, 2018), Cars3D (Reed et al., 2015), and the variants of dSprites introduced by Locatello et al. (Locatello et al., 2019b): Color-dSprites and Noisy-dSprites. Please refer to Appendix G for the details of the datasets.

We choose the following four baseline methods as representatives of the existing VAE-based models, which are denoted as Original VAEs. We verify the effectiveness of our implementation based on those methods. $\beta$**-VAE** (Higgins et al., 2017) introduces a hyperparameter $\beta$ in front of the KL regularizer of the VAE loss. It constrains the VAE information capacity to learn the most efficient representation. **AnnealVAE** (Burgess et al., 2018) progressively increases the bottleneck capacity so that the encoder learns new factors of variation while retaining disentanglement in previously learned factors. **FactorVAE** (Burgess et al., 2018) and $\beta$**-TCVAE** (Chen et al., 2018) both penalize the total correlation (Watanabe, 1960), but estimate it with adversarial training (Nguyen et al., 2010; Sugiyama et al., 2012) and Monte-Carlo estimator respectively.

| dSprits | DCI | | BetaVAE | | MIG | | FactorVAE | |
|---|---|---|---|---|---|---|---|---|
| | Original | Groupified | Original | Groupified | Original | Groupified | Original | Groupified |
| $\beta$-VAE | $0.23 \pm 0.10$ | $\mathbf{0.46 \pm 0.085}$ | $0.75 \pm 0.083$ | $\mathbf{0.86 \pm 0.051}$ | $0.14 \pm 0.097$ | $\mathbf{0.37 \pm 0.089}$ | $0.51 \pm 0.098$ | $\mathbf{0.63 \pm 0.089}$ |
| AnnealVAE | $0.28 \pm 0.10$ | $\mathbf{0.39 \pm 0.056}$ | $0.84 \pm 0.050$ | $\mathbf{0.87 \pm 0.0067}$ | $0.23 \pm 0.10$ | $\mathbf{0.34 \pm 0.061}$ | $0.70 \pm 0.094$ | $0.68 \pm 0.058$ |
| FactorVAE | $0.38 \pm 0.10$ | $\mathbf{0.41 \pm 0.074}$ | $0.89 \pm 0.040$ | $\mathbf{0.89 \pm 0.020}$ | $0.27 \pm 0.092$ | $\mathbf{0.31 \pm 0.061}$ | $0.74 \pm 0.068$ | $\mathbf{0.75 \pm 0.075}$ |
| $\beta$-TCVAE | $0.35 \pm \mathbf{0.065}$ | $\mathbf{0.36} \pm 0.11$ | $0.86 \pm 0.026$ | $\mathbf{0.861} \pm 0.038$ | $0.17 \pm 0.067$ | $\mathbf{0.24 \pm 0.093}$ | $0.68 \pm 0.098$ | $\mathbf{0.70} \pm 0.098$ |

| Cars3d | DCI | | BetaVAE | | MIG | | FactorVAE | |
|---|---|---|---|---|---|---|---|---|
| | Original | Groupified | Original | Groupified | Original | Groupified | Original | Groupified |
| $\beta$-VAE | $0.18 \pm 0.059$ | $\mathbf{0.24 \pm 0.041}$ | $0.99 \pm 1.6e-3$ | $\mathbf{1.0 \pm 0.0}$ | $0.071 \pm 0.032$ | $\mathbf{0.11 \pm 0.032}$ | $0.81 \pm 0.066$ | $\mathbf{0.93 \pm 0.034}$ |
| AnnealVAE | $0.22 \pm 0.046$ | $\mathbf{0.25 \pm 0.046}$ | $0.99 \pm 4e-4$ | $\mathbf{0.99 \pm 1.5e-4}$ | $0.074 \pm 0.016$ | $\mathbf{0.10 \pm 0.014}$ | $0.82 \pm 0.062$ | $\mathbf{0.87 \pm 0.028}$ |
| FactorVAE | $0.21 \pm 0.054$ | $\mathbf{0.25 \pm 0.040}$ | $0.99 \pm 1e-4$ | $\mathbf{1.0 \pm 0.0}$ | $0.098 \pm 0.027$ | $\mathbf{0.11 \pm 0.033}$ | $0.90 \pm 0.039$ | $\mathbf{0.93 \pm 0.034}$ |
| $\beta$-TCVAE | $0.24 \pm 0.049$ | $\mathbf{0.26 \pm 0.046}$ | $\mathbf{1.0 \pm 0.0}$ | $\mathbf{1.0 \pm 0.0}$ | $0.10 \pm 0.021$ | $\mathbf{0.11 \pm 0.033}$ | $0.88 \pm 0.040$ | $\mathbf{0.93 \pm 0.034}$ |

| Shapes3d | DCI | | BetaVAE | | MIG | | FactorVAE | |
|---|---|---|---|---|---|---|---|---|
| | Original | Groupified | Original | Groupified | Original | Groupified | Original | Groupified |
| $\beta$-VAE | $0.44 \pm 0.176$ | $\mathbf{0.56 \pm 0.10}$ | $0.91 \pm 0.072$ | $0.90 \pm \mathbf{0.045}$ | $0.28 \pm 0.18$ | $\mathbf{0.42 \pm 0.15}$ | $0.82 \pm 0.098$ | $\mathbf{0.82 \pm 0.043}$ |
| AnnealVAE | $0.52 \pm \mathbf{0.051}$ | $\mathbf{0.60 \pm 0.078}$ | $0.82 \pm 0.076$ | $\mathbf{0.89} \pm 0.086$ | $0.48 \pm 0.047$ | $\mathbf{0.50 \pm 0.052}$ | $0.75 \pm 0.074$ | $\mathbf{0.83 \pm 0.066}$ |
| FactorVAE | $0.47 \pm 0.10$ | $\mathbf{0.49 \pm 0.065}$ | $0.86 \pm 0.055$ | $0.80 \pm 0.075$ | $0.33 \pm 0.13$ | $\mathbf{0.43 \pm 0.11}$ | $0.81 \pm 0.056$ | $0.79 \pm 0.066$ |
| $\beta$-TCVAE | $0.66 \pm 0.10$ | $\mathbf{0.72 \pm 0.061}$ | $0.97 \pm 0.039$ | $0.96 \pm 0.042$ | $0.40 \pm 0.18$ | $\mathbf{0.47 \pm 0.090}$ | $0.89 \pm 0.064$ | $\mathbf{0.90 \pm 0.046}$ |

Table 1: Performance (mean $\pm$ std) on different datasets and different models with different metrics. We evaluate $\beta$-VAE, AnnealVAE, FactorVAE, and $\beta$-TCVAE on dSprites, Cars3d, Shapes3d, Noisy-dSprites, and Color-dSprites for 1800 settings. These settings include different random seeds and hyperparameters, refer to Appendix G for the details. We only show the first three datasets here. For more results, please refer to Appendix I.

## 5.2 QUANTITATIVE EVALUATIONS

This section performs quantitative evaluations on the datasets and models introduced with different random seeds and different hyperparameters. Then, we evaluate the performance of the Original and *Groupified* VAEs in terms of several popular metrics: BetaVAE score (Higgins et al., 2017), DCI disentanglement Eastwood & Williams (2018) (DCI in short), MIG (Chen et al., 2018), and FactorVAE score (Kim & Mnih, 2018). We assign three or four hyperparameter settings for each model on each dataset. We run it with ten random seeds for each hyperparameter setting to minimize the influence of random seeds. Therefore, we totally run $((3 \times 10 \times 3 + 10 \times 3 \times 3) \times 2) \times 5 = 1800$ models. We evaluate each metric's mean and variance for each model on each dataset to demonstrate the effectiveness of our method. As shown in Table 1, these *Groupified* VAEs have better performance (numbers marked bold in Table 1) than the Original VAEs on almost all the cases.

On Shapes3d, the *Groupified* VAEs outperform the Original ones on all the metrics except for BetaVAE scores, suggesting some disagreement between BetaVAE scores and other metrics. Similar disagreement is also observed between the variances of MIG and other metrics on Cars3d. Note that the qualitative evaluation in Appendix J shows that the disentanglement ability of *Groupified* VAEs is better on Shapes3d and Cars3d.

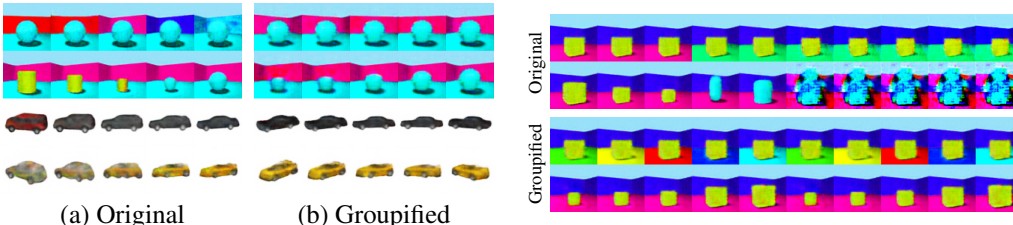

(a) Original          (b) Groupified

Figure 2: Visual traversal comparison between Original and *Groupified* β-TCVAE. The traversal results of *Groupified* VAEs are less entangled.

Figure 3: Traversal results of two factors (floor color, scale) of Original and *Groupified* β-TCVAE. The traversal results of *Groupified* VAEs are cyclic.

## 5.3 QUALITATIVE EVALUATIONS

We qualitatively show the *Groupified* VAEs achieve better disentanglement than the Original ones. As shown in Figure 2, the traversal results of *Groupified* β-TCVAE on Shape3d and Car3d are less entangled. For more qualitative evaluation, please refer to Appendix J. To verify that the *Groupified* VAEs learn a cyclic representation space (where $n = 10$), we provide the traversal results of [0,18] with a step of 2 for both the *Groupified* and Original β-TCVAE on Shape3d in Figure 3. We observe that the traversal results of *Groupified* VAEs are of high quality with a period of 10 (equal to $n$). However, the Original VAEs generate low-quality images without cyclicity. For the comparison of the results on CelebA (real-world datasets), please see appendix J.

## 5.4 VISUALIZATION OF THE LEARNED REPRESENTATION SPACE

To understand how our theoretical framework helps the existing VAE-based models to improve the disentanglement ability, we take dSprites as an example, visualize the learned representation space, and show the typical score distributions of the metrics. First, we visualize the space spanned by the three most dominant factors (x position, y position, and scale).

As shown in Figure 5 (for more results, please refers to Appendix L), the spaces learned by the Original VAEs collapse, while the spaces of the *Groupified* VAEs only bend a little bit. The main reason is that the Isomorphism Loss, serving as a self-supervision signal, suppresses the representation space distortion and encourages the disentanglement of the learned factors. As Figure 4 shows, the *Groupified* VAEs consistently achieve better mean performance with smaller variances. The isomorphism reduces the search space of the network so that the *Groupified* VAEs converge to the ideal disentanglement solution.

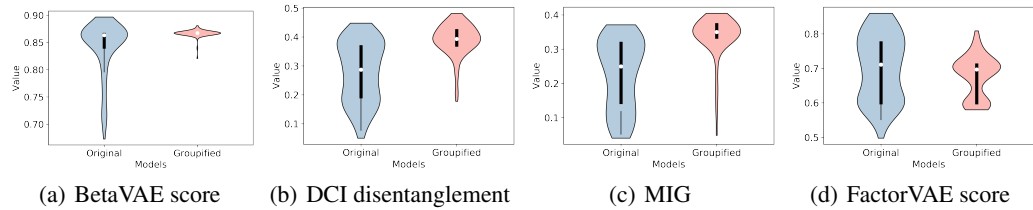

(a) BetaVAE score          (b) DCI disentanglement          (c) MIG          (d) FactorVAE score

Figure 4: Performance distribution of Original and *Groupified* AnnealVAE on dSprites (demonstrated by the Violin Plot (Hintze & Nelson, 1998)). Variance is due to different hyperparameters and random seeds. We observe that *Groupified* AnnealVAE improves the average performance with smaller variance in terms of BetaVAE score (a), DCI disentanglement (b), and MIG (c), and has a comparable mean performance with smaller variance in terms of FactorVAE score (d).

| | Original | Groupified | | | Factor Size $n = 10$ | | |
|---|---|---|---|---|---|---|---|
| | | $n = 5$ | $n = 10$ | $n = 15$ | w/o Abel | w/o Order | Groupified |
| DCI | $0.27 \pm 0.10$ | $0.34 \pm 0.062$ | $\mathbf{0.38 \pm 0.055}$ | $0.38 \pm 0.064$ | $0.28 \pm 0.11$ | $0.34 \pm 0.056$ | $\mathbf{0.38 \pm 0.055}$ |

Table 2: Ablation study on the factor size $n$ and Isomorphism Loss. DCI disentanglement is listed (mean ± std).

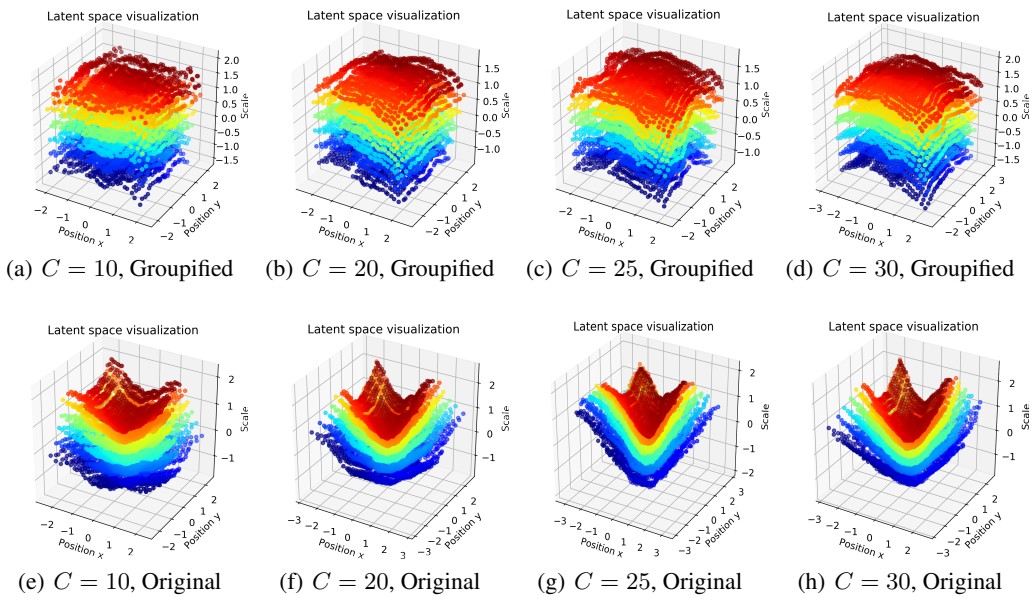

(a) $C = 10$, Groupified      (b) $C = 20$, Groupified      (c) $C = 25$, Groupified      (d) $C = 30$, Groupified

(e) $C = 10$, Original      (f) $C = 20$, Original      (g) $C = 25$, Original      (h) $C = 30$, Original

Figure 5: The representation space spanned by the learned factors by Original (bottom row) and *Groupified* AnnealVAE (top row). The position of each point is the disentangled representation of the corresponding image. An ideal result is all the points form a cube and color variation is continuous. The increase of $C$ (a hyperparameter of AnnealVAE) results in a collapse of representation space of the Original VAE. The collapse is suppressed by the Isomorphism Loss, which leads to better disentanglement.

## 5.5 ABLATION STUDY

We perform an ablation study on the assumed total number of possible values for a factor (factor size) $n$, Abel Loss $\mathcal{L}_a$, and Order Loss $\mathcal{L}_o$. We take the AnnealVAE trained on dSprites as an example. We only consider the DCI disentanglement metric here. We investigate the influence of factor size $n$. Besides, to evaluate the effectiveness of the two constraints, the models with the Abel Loss alone or Order Loss alone added are also evaluated. In this setting, we fix $n$ to 10. We compute the mean and variance of the performance for 30 settings of hyperparameters and random seeds. Table 2 shows that the isomorphism plays a role of cycle consistency in the representation space, leading to better disentanglement. The performance is robust to the factor size $n$, as the models learn to adapt to different $n$ in the training process. The models with only the Abel Loss or Order Loss applied have improved performance compared to the originals. The former (Abel Loss) performs better than the latter, suggesting that commutativity plays a more important role. Note that the number of factors $m$ can be learned and is not a hyperparameter. See Appendix F for details. $\gamma_I$ is empirically set to 1.

## 6 CONCLUSION

In this paper, we have opened the possibility of applying group-based definition to *unsupervised* disentanglement by proposing a theoretical framework. The group structure and model constraint in the framework are effective for existing VAE-based *unsupervised* disentanglement methods. In addition, by establishing the feasibility of learning the representation conforming to the definition in *unsupervised* settings, we have exhibited the consistently better mean performance with lower variance attributed to the definition. We believe our work constitutes a promising step towards *unsupervised* disentanglement with theoretical guarantee. As to the limitation, we only provide a necessary condition for the *data constraint*, as a result, we can not address the unidentifiability problem. Tackling the unidentifiability problem with the group-based definition is beyond the scope of this work, we will leave it as future work. In addition, a natural extension of our framework is to use *lie group* Hall (2015) (which is also a manifold) to extend our framework.

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

## A   PRELIMINARIES

***Group***: A set $G$ together with a binary operation $\circ$ as: $\circ : G \times G \to G$ satisfying the following properties:

- Associativity: $\forall\, a, b, c \in G, s.t.(a \circ b) \circ c = a \circ (b \circ c)$.
- Identity: $\exists\, e \in G, s.t.\,\forall\, a \in G, e \circ a = a \circ e = a$.
- Inverse: $\forall\, a \in G, \exists\, a^{-1} \in G : a \circ a^{-1} = a^{-1} \circ a = e$.

It is customary to represent a group with a set $G$ and the binary operation $\circ$ as a pair $(G, \circ)$. When the binary operation is clear, we represent group $(G, \circ)$ as $G$ and use multiplication to represent the binary operation $\circ$, i.e., $a \circ b = ab, \; \forall\, a, b \in G$.

***Group Action***: Let $(G, \circ)$ be a group and $P$ be a set. By the group actions of $(G, \circ)$ on $P$, we mean a mapping:

$$\cdot_P : G \times P \to P, \tag{6}$$

such that

- $\forall a, b \in G, p \in P, (a \circ b) \cdot p = a \cdot (b \cdot p)$.
- $e \cdot p = p$, where $e$ is the identity element of $G$.

***Symmetry Group and Permutation Group***: Let $\Sigma$ be a nonempty set, the bijections from $\Sigma$ to itself are called **Permutations**. $S(\Sigma)$ denotes the set containing all the permutations on $\Sigma$. $S(\Sigma)$ forms a group under the binary operation: composition of functions, which is called **Symmetry Group**. A subgroup of $S(\Sigma)$ is called **Permutation Group**.

***Abelian Group***: If the commutative law ($\forall\, a, b \in G, a \circ b = b \circ a$) holds in a group $(G, \circ)$, such a group is called an abelian group.

***Subgroup***: If a subset $H$ of a group $G$ is itself a group under the operation of $G$, we say that $H$ is a subgroup of $G$.

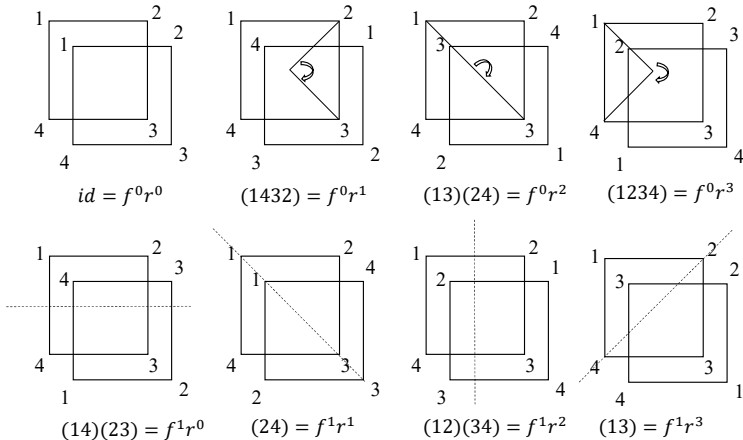

Figure 6: 4-th Dihedral Group ($D_4$): The groups of symmetries for square. Note that permutation $(123)$ means $1 \to 2, 2 \to 3, 3 \to 1$.

***Equivalence Relation***: An equivalence relation on a set $B$ is a subset $U \subset B \times B$ satisfying: (It is customary to represent $(a, b) \in U$ as $a \sim b$)

- Reflexive: $\forall\, a \in B, (a, a) \in U$.
- Symmetric: $(a, b) \in U \iff (b, a) \in U$.
- Transitive: $(a, b) \in U$ and $(b, c) \in U \Rightarrow (a, c) \in U$.

***Equivalence Class***: Let $\sim$ be an equivalence relation on a set $B$. We take an $a \in B$, and then the equivalence class containing $a$ is the subset $\overline{a} \triangleq \{b \in C | b \sim a\} \subset B$.

***Homomorphism*** Let $(G, \cdot)$ and $(H, \circ)$ be two groups. A homomorphism $f$, from $G$ to $H$, is a mapping $f : G \to H$, such that $f(x \cdot y) = f(x) \circ f(y)$, $\forall x, y \in G$.

***Isomorphism***: A homomorphism $f : G \to H$ which is bijective is called an isomorphism. Two groups are said to be isomorphic if there exists an isomorphism between them.

**Related Groups**:
($i$) ***Additive group of integers modulo $n$***: Let $n$ denote a positive integer. We define an equivalence relation on $\mathbb{Z}$ as $a \sim b \Leftrightarrow a = b \ (mod\ n)$ ($a$ and $b$ have the same remainder modulo $n$). The relation divides $\mathbb{Z}$ into $n$ equivalent classes: $\overline{0}, \overline{1}, \ldots, \overline{n-1}$, where $\overline{i}$ represents the equivalent class containing $i$, i.e., $\overline{i} = \{m \in \mathbb{Z} | m = i \ (mod\ n)\}$. Let $Z_n = \{\overline{0}, \overline{1}, \ldots, \overline{n-1}\}$, and $Z_n$ forms a group under the binary operation: $\overline{a} + \overline{b} = \overline{a + b}$, denoted by $\mathbb{Z}/n\mathbb{Z}$.
($ii$) ***Group of $n$-th root unity***: Let $n$ denote a positive integer and $C_n = \{e^{\frac{2\pi i a}{n}} | 0 \le a \le n-1, a \in \mathbb{Z}\}$ be $n$ roots of $x^n = 1$, and then $C_n$ forms a group under complex multiplication. For the mapping: $f : C_n \to (Z_n, +)$ defined by $f(e^{\frac{2\pi i a}{n}}) = \overline{a}$, we have

$$f\left(e^{\frac{2\pi i a}{n}} \cdot e^{\frac{2\pi i b}{n}}\right) = f\left(e^{\frac{2\pi i (a+b)}{n}}\right) = \overline{a+b} = \overline{a} + \overline{b} = f\left(e^{\frac{2\pi i a}{n}}\right) \cdot f\left(e^{\frac{2\pi i b}{n}}\right) \tag{7}$$

Therefore, $f$ is a homomorphism, and thus is an isomorphism.

***Congruence Class***: Let $n$ denote a positive integer, and we define an equivalence relation on $\mathbb{Z}$ as $a \sim b \Leftrightarrow a = b \ (mod\ n)$. The relation divides $\mathbb{Z}$ into $n$ equivalent classes: $\overline{0}, \overline{1}, \ldots, \overline{n-1}$, which are congruence classes and the elements of the additive group of integers modulo $n$.

***Subgroup Generated by $B$***: If $B$ is a nonempty subset of the group $(G, \circ)$, the set defined by $ = \{a \in G | a = a_1 \circ a_2 \circ \cdots \circ a_n \text{ with either } a_i \in B \text{ or } a_i^{-1} \in B\}$ forms a subgroup of $G$, which is called the subgroup generated by $B$, e.g., $G = <a, b | a^2 = e, ab = ba>$ in the toy example of Section 1. Here $a_1, \ldots a_n$ are called **Generators**, e.g., $a, b$ in $G$.

***Cyclic Group*** A group $G$ is called cyclic if there is an element $a \in G$ such that $G = \{a^n | n \in \mathbb{Z}\} = <a>$, e.g., $\mathbb{Z}/n\mathbb{Z}$. Such an element $a$ is called a generator of $G$.

***Symmetry***: A symmetry of a geometric figure is a rearrangement of the figure preserving the arrangement of its sides and vertices as well as its distances and angles.

***Dihedral Group***: The permutation group formed by the symmetries of a regular $n$-sided polygon, denoted by $D_n$.

$D_4$ **as an example**: The vertices of a square are numbered by $\{1, 2, 3, 4\}$, which is analogous to an image dataset, the symmetries are analogous to image transformations. We often abbreviate permutation $1 \to 2, 2 \to 3, 3 \to 4, 4 \to 1$ as $(1234)$. The elements of group $D_4$ are shown in Figure 6, from which we know that all of the transformations are compounded by two basic permutations: horizontal flip $f$ and rotate 90 degrees clockwise $r$. Please note that $f$ and $r$ are analogous to disentangled factors. What's more, the Group can be generated by these basic permutations $f, r$ with some properties, i.e., $D_4 = <f, r | f^2 = 1, r^4 = 1, fr = r^{-1}f>$. The constraints $f^2 = 1, r^4 = 1, fr = r^{-1}f$ are analogous to the group constraints in this paper.

## B   PROOF FOR THEOREM 1

In our setting, the group-based definition is equivalent to the equivariant condition $g \cdot f(w) = f(g \cdot w)$ for the following reason. Since the group actions of $G$ on $Z$ is the element wise addition, ($a$) the group action of $G$ on $Z$ exists and ($b$) group actions of $G_i$ only affect $Z_i$ (refer to Section 3.2). These two conditions of the group-based definition thus hold.

*Proof.* In the following, we prove in Step 1 that the necessary and sufficient conditions of $\Phi$ playing the "agent" role , i.e., $f(g \cdot w) = h(\varphi \cdot b(w))$, are ($i$) $\Phi$ is isomorphic to $G$, and ($ii$) there exist $\varphi_i \in \Phi$, s.t. $\varphi_i \cdot b(w) = b(g_i \cdot w)$, $i = 1, 2, \ldots, m$. We then prove in Step 2 that if equation $f(g \cdot w) = h(\varphi \cdot b(w))$ and the definition of $\varphi \in \Phi$ hold, then $Z$ is disentangled per group-based definition.

**Step 1**: In the following, we prove that the necessary and sufficient conditions of $f(g \cdot w) = h(\varphi \cdot b(w))$ are $(i)$ and $(ii)$ above. Note that this holds even without a specific definition of $\Phi$.

$\Rightarrow$) For a general permutation group $\Phi$ on $O$, we assume there exist $\varphi_i$ such that $\varphi_i \cdot b(w) = b(g_i \cdot w), i = 1, 2, \ldots, m$. Assume that there exists an isomorphism $\tau : G \to \Phi$, where group $G = (\mathbb{Z}/n\mathbb{Z})^m$. From Group Theory, we know that there exists $\mu \in Aut(\Phi)$ such that the following equation holds, where $Aut(\Phi)$ denotes the group of automorphisms of $\Phi$, i.e., $\mu$ is an isomorphism $\mu : \Phi \to \Phi$. We denote the composition of $\mu, \tau$ as $\sigma$, and have

$$\sigma(g_i) = \mu(\tau(g_i)) = \varphi_i, \ i = 1, 2, \ldots, m. \tag{8}$$

Since we have $\varphi_i \cdot b(w) = b(g_i \cdot w), \ i = 1, 2, \ldots, m$, the following equation holds,

$$\sigma(g_i) \cdot b(w) = \varphi_i \cdot b(w) = b(g_i \cdot w), \ i = 1, 2, \ldots, m. \tag{9}$$

Since both $\mu, \tau$ are isomorphism, $\sigma = \mu \circ \tau$ is also an isomorphism, which indicates that $\varphi_i$ is a generator of $\Phi$. For $\forall \varphi \in \Phi$, since $\sigma$ is a bijection, there exists $g \in G = (\mathbb{Z}/n\mathbb{Z})^m$, such that $\sigma(g) = \varphi$ and $g = \sum_i k_i g_i, k_i \in \mathbb{Z}$, where $g_i, i = 1, 2, \ldots, m$ denotes the generators in $G$. In order to calculate $\varphi \cdot b(w)$, we consider a specific example $\sigma(g_1 + g_2)$.

$$\sigma(g_1 + g_2) \cdot b(w) = \sigma(g_1) \cdot (\sigma(g_2) \cdot b(w)) = \sigma(g_1) \cdot b(g_2 \cdot w) = b(g_1 \cdot (g_2 \cdot w)) = b((g_1 + g_2) \cdot w). \tag{10}$$

Therefore, for the general element $\varphi$, we have

$$\varphi \cdot b(w) = \sigma(\sum_i k_i g_i) \cdot b(w) = b(\sum_i k_i g_i \cdot w) = b(g \cdot w). \tag{11}$$

In general, we have $\varphi \cdot b(w) = b(g \cdot w)$ and thus we have $h \circ b(g \cdot w) = f(g \cdot w) = h(\varphi \cdot b(w))$ when taking $h$ on both sides.

$\Leftarrow$) Here we prove that $(i)$ $G$ is isomorphic to $\Phi$ and $(ii)$ there exists $\varphi_i$ such that $\varphi_i \cdot b(w) = b(g_i \cdot w), i = 1, 2, \ldots, m$ are two necessary conditions of equation $f(g \cdot w) = h(\varphi \cdot b(w)) = h(\varphi \cdot x)$. We take $f^{-1}$ on both sides of the equation. For convenience, we rewrite $g \cdot w, \varphi \cdot b(w)$ as $g(w)$ and $\varphi(b(w))$, then we have

$$g(w) = f^{-1} \circ h \circ \varphi \circ b(w) = b^{-1} \circ \varphi \circ b(w). \tag{12}$$

Note that the notation $\circ$ here denotes the composition of functions. We define the mapping $\tau$ between $\Phi$ and $G$ as follows:

$$\tau(\varphi) = g = b^{-1} \circ \varphi \circ b. \tag{13}$$

Note that $b$ is a bijection, thus $\tau$ is a bijection. We take $\varphi_i, \varphi_j \in \Phi$ and we have

$$\begin{aligned} \tau(\varphi_i \circ \varphi_j) &= b^{-1} \circ \varphi_i \circ \varphi_j \circ b \\ &= (b^{-1} \circ \varphi_i \circ b) \circ (b^{-1} \circ \varphi_j \circ b) \\ &= \tau(\varphi_i) \circ \tau(\varphi_j). \end{aligned} \tag{14}$$

Therefore, $\tau$ is a homomorphism and thus is an isomorphism. i.e., $\Phi$ is isomorphic to $G$. Recall that group $G$ has the form of $G = (\mathbb{Z}/n\mathbb{Z})^m$, where it is a direct product of $m$ cyclic groups. For $\forall \varphi \in \Phi$, we have

$$\varphi \cdot b(w) = h^{-1} \circ f(g(w)) = b(g(w)). \tag{15}$$

For generators $g_i \in G, i = 1, 2, \ldots, m$ and the corresponding $\varphi_i = \tau^{-1}(g_i)$, we have a specific one derived from the above equation:

$$\varphi_i \cdot b(w) = b(g_i(w)) = b(g_i \cdot w). \tag{16}$$

Q.E.D.

**Step 2**: As discussed in Section 1 and Section 3, we define $\varphi$ as $\varphi \cdot x = \varphi \cdot b(w) = h^{-1}(\sigma^{-1}(\varphi) \cdot h(x)) = h^{-1}(g \cdot f(w))$, where $\sigma$ is the same as in Step 1. Since $\sigma$ is a bijection, for $\forall \varphi \in \Phi$, $\sigma^{-1}(\varphi)$ uniquely exists, and thus $\varphi$ is well-defined. We bring it into $f(g \cdot w) = h(\varphi \cdot b(w))$, and derive $f(g \cdot w) = g \cdot f(w)$, i.e., the representation space $Z$ is disentangled with respect to $G$. Specifically, for the existing $\varphi_i, i = 1, 2, \ldots, m$, from equation 8, we have $\sigma^{-1}(\varphi_i) = g_i$ and $\varphi_i \cdot x = h^{-1}(g_i \cdot f(w))$.

Q.E.D.

## C PROOF FOR THEOREM 2

In order to easily distinguish a general element and a generator, we use $\phi$ to stand for the general element in $\Phi$ and $\varphi_i$ to stand for generators in $\Phi$. In addition, for mapping $\tau : G \to \Phi$, notation $ker(\tau)$ is generally the inverse image of $e$, i.e., $ker(\tau) = \{g \in G | \tau(g) = e\}$, where $e$ is the identity element in $\Phi$.

*Proof.* Note that Section 3.4 describes the implementation of the elements in $\Phi$, but there is no guarantee that $\Phi$ is a group. Therefore, here we first prove in Step 1 that set $\Phi =< \varphi_1, \varphi_2, \ldots, \varphi_m | \varphi_i \varphi_j = \varphi_j \varphi_i, \varphi_i^n = e, i = 1, 2, \ldots, m$ and $j = 1, 2, \ldots, m >$ forms a group under the composition of functions, where $e$ satisfies that $e\varphi_i = \varphi_i$, then prove in Step 2 the necessary and sufficient condition for the isomorphism is to meet both the Abel constraint and the Order constraint.

**Step 1**: To prove a set forms a group under some operation, we only need to verify that the elements of the set satisfy the following three properties: 1. Associativity 2. Identity 3. Inverse.

1. We first verify the Associativity property: $\forall \phi_s, \phi_t, \varphi_l \in \Phi, s.t. (\phi_s \phi_t)\phi_l = \phi_s(\phi_t \phi_l)$.

Recall that in Section 3.2, the representation space is a set with the same elements in $G$. The group action of $G$ on $Z$ is the addition. We have $g \cdot z \in Z, \forall g \in G, z \in Z$, and then we have $d(g \cdot z) \in O, \forall g \in G, z \in Z$. Therefore, $\varphi_i$ is a permutation: $\varphi_i : O \to O$, i.e., $\varphi_i \in S(O)$, where $S(O)$ is the symmetry group on images set $O$, then for $\forall \varphi_i, \varphi_j, \varphi_k \in \{\varphi_1, \varphi_2, \ldots \varphi_m\}$, we have

$$(\varphi_i \varphi_j)\varphi_k = \varphi_i(\varphi_j \varphi_k), \tag{17}$$

the generators of $\Phi$ thus satisfy the Associativity property. However, whether the general elements of $\Phi$ satisfy the Associativity property is unknown. We take $\forall \phi_s, \phi_t, \phi_l \in \Phi$, where $\phi_t = \prod_i \varphi_i^{t_i}, \phi_s = \prod_i \varphi_i^{s_i}, \phi_l = \prod_i \varphi_i^{l_i}$ and $t_i, s_i, l_i \in \mathbb{Z}$, and we have

$$(\phi_t \phi_s)\phi_l = \left(\prod_i \varphi_i^{t_i} \prod_i \varphi_i^{s_i}\right) \prod_i \varphi_i^{l_i} = \prod_i \varphi_i^{t_i} \left(\prod_i \varphi_i^{s_i} \prod_i \varphi_i^{l_i}\right) = \phi_t(\phi_s \phi_l). \tag{18}$$

Therefore, the set of mappings $\Phi$ satisfies the Associativity property under the composition of functions.

2. We then verify the Identity property: $\exists e \in \Phi, s.t. \forall \phi \in \Phi, e\phi = \phi e = \phi$.

For generators $\varphi_i, \varphi_j \in \{\varphi_1, \varphi_2, \ldots \varphi_m\}$, we have $\varphi_i^n \varphi_j = e\varphi_j = \varphi_j$. Therefore, for general elements, we take $\forall \phi \in \Phi, \phi = \prod_i \varphi_i^{k_i}$, and we have:

$$\varphi_i^n \phi = e \prod_i \varphi_i^{k_i} = (e\varphi_j)\varphi_j^{k_j-1} \prod_{i\{i \neq j\}} \varphi_i^{k_i} = \varphi_j^{k_j} \prod_{i\{i \neq j\}} \varphi_i^{k_i} = \prod_i \varphi_i^{k_i} = \phi. \tag{19}$$

This states that the identity element of $\Phi$ is $e = \varphi_i^n \in \Phi$. Therefore, the set of mappings $\Phi$ satisfies the Identity property.

3. We finally verify the Inverse property: $\forall \phi \in \Phi, \exists \phi^{-1} \in G : \phi\phi^{-1} = \phi^{-1}\phi = e$.

For the generators $\varphi_i \in \{\varphi_1, \varphi_2, \ldots \varphi_m\}$, we have $\varphi_i^k \varphi_i^{n-k} = \varphi_i^n = e, 1 \leq k \leq n, k \in \mathbb{Z}$. For any general element $\phi \in \Phi, \phi = \prod_i \varphi_i^{k_i}$, we have:

$$\phi \prod_i \varphi_i^{n-k_i} = \prod_{i\{i \neq j\}} \varphi_i^{k_i}(\varphi_j^{k_j}\varphi_j^{n-k_j}) \prod_{i\{i \neq j\}} \varphi_i^{n-k_i} = \cdots = e, \tag{20}$$

we denote $\prod_i \varphi_i^{n-k_i}$ as $\phi^{-1}$ and have $\phi\phi^{-1} = e$. Similarly, we also have $\phi^{-1}\phi = e$. These two equations state that for any general element $\phi \in \Phi, \phi = \prod_i \varphi_i^{k_i}$, we have an inverse element $\phi^{-1} = \prod_i \varphi_i^{n-k_i}$ in $\Phi$. Therefore, the set of mappings $\Phi$ satisfies the Inverse property.

To summarize, $\Phi$ is a group.

Q.E.D.

**Step 2**: In this step, we prove that the necessary and sufficient condition for the isomorphism $(\mathbb{Z}/n\mathbb{Z})^m \sim< \varphi_1, \varphi_2, \ldots, \varphi_m >$ is the satisfaction of $\varphi_i \varphi_j = \varphi_j \varphi_i, i = 1, 2, \ldots, m$ and $j = 1, 2, \ldots, m$, and $\varphi_i^n = e, i = 1, 2, \ldots, m$.

$\Rightarrow$) Assume the isomorphism is $\tau : G = (\mathbb{Z}/n\mathbb{Z})^m \to \Phi =< \varphi_1, \varphi_2, \ldots, \varphi_m >$, for generators $\varphi_i, \varphi_j \in \{\varphi_1, \varphi_2, \ldots \varphi_m\}$, there exist $g_\alpha = \sum_i \alpha_i g_i, g_\beta = \sum_i \beta_i g_i \in G, s.t. \tau(g_\alpha) = \varphi_i, \tau(g_\beta) = \varphi_j$, where $g_i, i = 1, 2, \ldots, m$ denotes the generators in $G$, which is the m-dimensional one-hot vector $(0, \ldots, \bar{1}, \ldots, 0) \in G$ with 1 in position i and 0 elsewhere, and $\alpha_i, \beta_i \in \mathbb{Z}$. Since $G$ is an Abelian group, we have

$$\varphi_i \varphi_j = \tau(g_\alpha)\tau(g_\beta) = \tau(g_\alpha g_\beta) = \tau(g_\beta g_\alpha) = \tau(g_\beta)\tau(g_\alpha) = \varphi_j \varphi_i. \tag{21}$$

For generators $\varphi_i \in \{\varphi_1, \varphi_2, \ldots \varphi_m\}$, we have $g_\alpha = \sum_i \alpha_i g_i \in G, s.t. \tau(g_\alpha) = \varphi_i$, the $n$ times composition of itself is

$$\varphi_i^n = \tau(g_\alpha)^n = \tau(ng_\alpha) = \tau(\sum_i \alpha_i n g_i) = \tau(\sum_i \alpha_i e_G) = \tau(e_G) = e, \tag{22}$$

where $e_G, e$ are identity elements of $G$ and $\Phi$ respectively. In Equation 22, $\tau(e_G) = e$ holds because $\tau$ is an isomorphism, and $ker(\tau) = \{e_G\}$. The sufficiency is proven.

$\Leftarrow$) In the following, we prove that when two conditions are satisfied simultaneously, the mapping $\tau$ we define is an isomorphism. Considering the mapping $\tau : \Phi \to G$, defined as

$$\begin{cases} \tau : \varphi_i \mapsto g_i; \\ \tau : \varphi_i \varphi_j \mapsto g_i + g_j. \end{cases} \tag{23}$$

For general elements $\phi_t, \phi_s \in \Phi$, where $\phi_t = \prod_i \varphi_i^{t_i}, \phi_s = \prod_i \varphi_i^{s_i}$ and $t_i, s_i \in \mathbb{Z}$, we have

$$\tau(\phi_t \phi_s) = \tau\left(\prod_i \varphi_i^{t_i} \prod_i \varphi_i^{s_i}\right) = \tau\left(\prod_i \varphi_i^{t_i+s_i}\right) = \sum_i (t_i + s_i)g_i. \tag{24}$$

Note that these terms can be merged since $\varphi_i \varphi_j = \varphi_j \varphi_i$. The summation on the right side of Equation 24 is partitioned into two parts as follows

$$\sum_i (t_i + s_i)g_i = \sum_i t_i g_i + \sum_i s_i g_i = \tau\left(\prod_i \varphi_i^{t_i}\right) + \tau\left(\prod_i \varphi_i^{s_i}\right) = \tau(\phi_t) + \tau(\phi_s). \tag{25}$$

Consequently, we have $\tau(\phi_t \phi_s) = \tau(\phi_t) + \tau(\phi_s)$, which states that $\tau$ is a homomorphism. Then, we only need to prove that mapping $\tau$ is bijective. First, we prove $\tau$ is injective, i.e., $ker(\tau) = \{e_G\}$, after that, we prove $\tau$ is surjective, i.e., one can find the inverse image of any element.

Since it is not hard to obtain $\tau(\phi) = \tau(e \cdot \phi) = \tau(e) + \tau(\phi)$, we have $\tau(e) = (0, \ldots, 0) = e_G$, i.e., $e \in \tau^{-1}(e_G)$. Assume there is $\phi_l = \prod_i \varphi_i^{l_i}, s.t. \tau(\phi_l) = e_G$, we have

$$\tau(\phi_l) = \sum_i l_i g_i = e_G \Rightarrow l_i | n \ (l_i \text{ are divisible by } n) \Rightarrow \phi_l = \prod_i \varphi_i^{l_i} = \prod_i e = e. \tag{26}$$

The equation above states that $\tau^{-1}(e_G) = \{e\} \ (\tau : \Phi \to \Phi)$ and mapping $\tau$ is injective. We take $\forall g \in G, g = (\overline{k_1}, \overline{k_1}, \ldots, \overline{k_m}) \in (\mathbb{Z}/n\mathbb{Z})^m$, and have

$$g = (\overline{k_1}, \ldots, 0) + \cdots + (0, \ldots, \overline{k_m}) = \sum_i k_i g_i = \tau\left(\prod_i \varphi_i^{k_i}\right). \tag{27}$$

Hence, we have $\tau^{-1}(g) = \prod_i \varphi_i^{k_i} \in \Phi$, which indicates that mapping $\tau$ is surjective. Since mapping $\tau$ is bijective and homomorphism, $\tau$ is an isomorphism.

Q.E.D.

## D   PROOF FOR THEOREM 3

*Proof.* In the following, we prove that the necessary and sufficient condition for satisfying Abel and Order constraints is that both Abel and Order Loss are minimized.

$\Rightarrow$) For the condition: $\forall \varphi_i, \varphi_j \in \Phi, 0 \le i, j \le m$, we have $\varphi_i \varphi_j = \varphi_j \varphi_i$. The constraint on any image $o \in O$ can be obtained by

$$\varphi_i(\varphi_j(o)) = \varphi_j(\varphi_i(o)) \Rightarrow \varphi_i(\varphi_j(o)) - \varphi_j(\varphi_i(o)) = 0. \tag{28}$$

For the set of combinations of factors $C = \{(i,j) | 1 \leq i, j \leq m\}$ and the set containing images $O$, we have

$$\mathcal{L}_a = \sum_{o \in O} \sum_{(i,j) \in C} \|\varphi_i \cdot (\varphi_j \cdot o) - \varphi_j \cdot (\varphi_i \cdot o)\| \text{ is minimized.} \tag{29}$$

We obtain the Abel loss. For the Order loss, we first consider $n$ times composition of the same mapping $\varphi_i$, we have

$$\varphi_i^n = e \Rightarrow \varphi_i^{n-1} \varphi_i = \varphi_i \varphi_i^{n-1} = e. \tag{30}$$

Therefore, we then have $\varphi_i^{-1} = \varphi_i^{n-1}$. For a single image $o \in O$, we have

$$\varphi_i(\varphi_i^{n-1}(o)) = \varphi_i(\varphi_i^{-1}(o)) = o \Rightarrow \varphi_i(\varphi_i^{-1}(o)) - o = 0. \tag{31}$$

Thus, for the set of factors and the set containing all images $O$, we have

$$\sum_{o \in O} \sum_{0 \leq i \leq m} \|\varphi_i \cdot (\varphi_i^{-1} \cdot o) - o\| \text{ is minimized.} \tag{32}$$

To eliminate the bias of optimization, we optimize the symmetry form of the Order Loss, and we have

$$\mathcal{L}_o = \sum_{o \in O} \sum_{0 \leq i \leq m} (\|\varphi_i \cdot (\varphi_i^{-1} \cdot o) - o\| + \|\varphi_i^{-1} \cdot (\varphi_i \cdot o) - o\|) \text{ is optimized.} \tag{33}$$

The Order Loss is obtained.

$\Leftarrow$)When the Abel Loss $\mathcal{L}_a$ is optimized, for $\forall o \in O$, we have

$$\varphi_i(\varphi_j(o)) - \varphi_j(\varphi_i(o)) = 0 \Rightarrow \varphi_i(\varphi(o)) = \varphi_j(\varphi_i(o)). \tag{34}$$

Therefore, for $\forall \varphi_i, \varphi_j \in \Phi, i = 1, 2, \ldots, m$ and $j = 1, 2, \ldots, m$, we have $\varphi_i \varphi_j = \varphi_j \varphi_i$, and we obtain the Abel constraint. When the Order Loss $\mathcal{L}_o$ is minimized, for $\forall o \in O$, we have

$$\varphi_i(\varphi_i^{-1}(o)) - o = 0 \Rightarrow \varphi_i(\varphi_i^{-1}(o)) = o. \tag{35}$$

This implies that

$$\varphi_i^n = \varphi_i \circ \varphi_i^{n-1} = \varphi_i \circ \varphi_i^{-1} = e. \tag{36}$$

Therefore, we obtain the Order constraint. The Group constraints are satisfied.

## E  THE DATA CONSTRAINT

In this section, we prove that if each dimension of the world state is independently sampled ($p(w) = \Pi_i p(w_i)$, where $p$ denotes the probability mass function (for discrete random variable) or probability density function (for continuous random variable)), then the minimization of total correlation ($p(z) = \Pi_i p(z_i)$) is a necessary condition to satisfy the data constraint (see Theorem 4 and Theorem 5 below).

Our goal is to learn a representation $z$ conforming to the group-based definition of disentanglement with an encoder $h$ and a decoder $d$. Then the data constraint can be formulated as follows. For a generator $g_i \in G$, we have $b(g_i \cdot w) = \varphi_i \cdot b(w) = h^{-1}(g_i \cdot z)$, where $z = f(w)$ and $\varphi_i$ is the corresponding permutation of $g_i$ under the isomorphism between $G$ and $\Phi$. We reorganize the formula and have $h \circ b(g_i \cdot w) = f(g_i \cdot w) = g_i \cdot z$.

**Theorem 4**  *Assume that the action of generator $g_i$ on $z$ is $g_i \cdot z = \overline{z + g_i}$ (element-wise addition), and the action of each generator $g_i$ on $w$ only affects a single dimension of $w$ (see the assumptions in Section 3.2). Then the equation $f(g_i \cdot w) = g_i \cdot z, \forall w \in W$ is equivalent to: for each $i = 1, 2, \ldots, n$, there exists a bijective function $\gamma_i$ s.t. $z_i = \gamma_i(w_j)$ for some $j$.*

*Proof.* $\Rightarrow$) Without loss of generality, we assume $g_i$ only affects the $j$-th dimension of $w$. If the equation $f(g_i \cdot w) = g_i \cdot z, \forall w \in W$ holds, it's obvious that, for each $i$, there exists a bijective function $\gamma_i$ s.t. $z_i = \gamma_i(w_j)$.

$\Leftarrow$) In the following, we use $(g_i \cdot w)_i$ ($(g_i \cdot z)_i$) to denote the $i$-th dimension of vector $g_i \cdot w$ ($g_i \cdot z$). If for $i = 1, \ldots, n$, the functions $z_i = \gamma_i(w_j)$ hold for some $j$, by using an index permutation $j = \pi(i)$ we can rewrite the function as $z_i = \gamma_i(w_{\pi(i)})$. Therefore, $z$ can be formulated as follows

$$f(w_1, \ldots, w_n) = z = (z_1, \ldots, z_n) = (\gamma_1(w_{\pi(1)}), \ldots, \gamma_n(w_{\pi(n)})). \tag{37}$$

Please note that how the world state transits on dimension $i$ dose not affect how disentangled the representation is. Therefore, we define action of generator $g_i$ on $w$ as follows.

$$g_i \cdot w = g_i \cdot (w_1, \ldots, w_n) = (w_1, \ldots, (g_i \cdot w)_j, \ldots, w_n) = (w_1, \ldots, \gamma_i^{-1}((g_i \cdot z)_i), \ldots, w_n), \tag{38}$$

Then, we take $f$ on both sides of the above equation, and apply Equation 37, and have

$$
\begin{aligned}
f(g_i \cdot w) \quad &= f(w_1, \ldots, \gamma_i^{-1}((g_i \cdot z)_i), \ldots, w_n) = (\gamma_1(w_{\pi(1)}), \ldots, \gamma_i(\gamma_i^{-1}((g_i \cdot z)_i)), \ldots, \gamma_n(w_{\pi(n)})) \\
&= (z_1, \ldots, (g_i \cdot z)_i, \ldots, z_n) = g_i \cdot z
\end{aligned}
\tag{39}
$$

Q.E.D.

Note that it is obvious that the following theorem (Theorem 5) holds for discrete random variables and bijective functions. However, since the world state space is dense and the encoder $h$ and decoder $d$ are differentiable in general, we also prove for the case where $w$ and $z$ are treated as continuous random variables. Theorem 5 states that the minimization of total correlation ($p(z) = \Pi_i p(z_i)$) is a necessary condition to make equations $z_i = \gamma_i(w_{\pi(i)})$ satisfied, where $i = 1, 2, \ldots, n$ and $\pi(i)$ is an index permutation.

**Theorem 5**  *For the independent random variables $w_1, w_2, \ldots, w_n$, considering the functions $z_i = \delta_i(w_1, w_2, \ldots, w_n)$, where $i = 1, 2, \ldots, n$ and each $\delta_i$ is a bijective, differentiable function. For $i = 1, \ldots, n$, if there exists an index permutation $\pi(i)$ s.t., we have $\frac{\partial z_{\pi(i)}}{\partial w_i} \neq 0$ and $\frac{\partial z_{\pi(i)}}{\partial w_k} = 0$, where $k = 1, 2 \ldots, n$ but $k \neq i$, then the new random variables $z_1, z_2, \ldots, z_n$ are independent.*

*Proof.* We treat the random variables $w_1, w_2, \ldots, w_n$ as an $n$-dimensional random vector $W = (w_1, w_2, \ldots, w_n)$. Similarly, we write $Z = (z_{\pi(1)}, z_{\pi(2)}, \ldots, z_{\pi(n)})$, which is rearranged by index permutation $\pi(i)$. According to Change of Variable Theorem For Random Vectors (Mood, 1950), we have

$$p(Z)|J(W)| = p(W) = \Pi_i p(w_i) \tag{40}$$

where $J(W)$ is the jacobian matrix of $Z$ w.r.t. $W$, the $(i, j)$-th entry of it is $\frac{\partial z_{\pi(i)}}{\partial w_j}$. Since for each $i = 1, \ldots, n$, we have $\frac{\partial z_{\pi(i)}}{\partial w_i} \neq 0$ and $\frac{\partial z_{\pi(i)}}{\partial w_k} = 0$, where $k = 1, 2 \ldots, n$ but $k \neq i$, then the jacobian

matrix can be formulated as follows

$$J(W) = \Pi_i \frac{\partial z_{\pi(i)}}{\partial w_i} \tag{41}$$

According to Change of Variable Theorem For Random Variable ( Mood (1950)), we have

$$p(z_{\pi(i)}) |\frac{\partial z_{\pi(i)}}{\partial w_i}| = p(w_i) \tag{42}$$

Bring Equation 41 and Equation 42 into Equation 40, we have

$$p(Z) = \frac{\Pi_i p(w_i)}{\Pi_i |\frac{\partial z_{\pi(i)}}{\partial w_i}|} = \frac{\Pi_i p(z_{\pi(i)}) \Pi_i |\frac{\partial z_{\pi(i)}}{\partial w_i}|}{\Pi_i |\frac{\partial z_{\pi(i)}}{\partial w_i}|} = \Pi_i p(z_{\pi(i)}) = \Pi_i p(z_i) \tag{43}$$

Q.E.D.

Please note that the inverse proposition of the theorem above does not hold. As a counterexample, for zero mean independent gaussian random variables $w_1, w_2$ with common variance $\sigma^2$, new random variables $z_1 = \sqrt{w_1^2 + w_2^2}$ (the norm of vector $(w_1, w_2)$), $z_2 = \tan^{-1}(w_2/w_1)$ (the angle between vector $(w_1, w_2)$ and $(0, 1)$) are independent (Mood, 1950) but $\frac{\partial z_i}{\partial w_j} \neq 0$ for all $i, j \in \{1, 2\}$. In addition, the sufficiency would be satisfied for some specific settings, e.g., there are only two independent uniformly distributed random variables $w_1, w_2$, and the functions $z_1, z_2$ are linear functions.

Combining Theorems 4 and 5, we have that the minimization of total correlation $(p(z) = \Pi_i p(z_i))$ is a necessary condition to satisfy the data constraint.

## F  DETAILS OF IMPLEMENTATION

### F.1  ABEL LOSS DETAILS

As mentioned in the main paper, the Abel Loss of the VAE-based models is as follows:

$$\mathcal{L}_a = \sum_{o \in O} \sum_{(i,j) \in C} \|\varphi_i \cdot (\varphi_j \cdot o) - \varphi_j \cdot (\varphi_i \cdot o)\|, \tag{44}$$

where $\varphi_i \cdot (\varphi_j \cdot o) = \varphi_i(\varphi_j(o))$ represents the top path of Figure 7 (a), and $\varphi_j \cdot (\varphi_i \cdot o) = \varphi_j(\varphi_i(o))$ represents the bottom path of Figure 7 (a). For better optimization, we constrain such consistency on their representation (straight dotted double arrow in Figure 7 (a)) instead of the reconstructed images. Besides, we constrain the consistency between the representations of intermediate images (curved dotted double arrow in Figure 7 (a)).

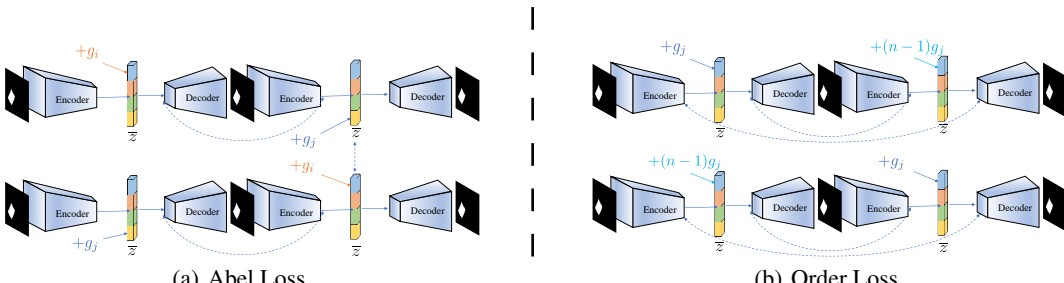

(a) Abel Loss  (b) Order Loss

Figure 7: Overview of the Isomorphism Loss. The Abel Loss and Order Loss constrain the commutativity and cyclicity of permutation group $\Phi$, respectively. The dotted lines in the figure represents reconstruction loss.

### F.2  ORDER LOSS DETAILS

As mentioned in the main paper, the Order Loss of VAE-based models is as follows:

$$\mathcal{L}_o = \sum_{o \in O} \sum_{i \in I} \left( \|\varphi_i \cdot (\varphi_i^{-1} \cdot o) - o\| + \|\varphi_i^{-1} \cdot (\varphi_i \cdot o) - o\| \right), \tag{45}$$

where $\varphi_i \cdot (\varphi_i^{-1} \cdot o)$ represents the lower path of Figure 7 (b), and $\varphi_i^{-1} \cdot (\varphi_i \cdot o)$ represents the upper path of Figure 7 (b). Similar to the Abel Loss, we do not constrain such consistency on the reconstructed images for better optimization but on their representations instead (the long curved dotted line in Figure 7 (b)). Besides, we constrain the consistency between the representations of intermediate images (short curved dotted lines in Figure 7 (b)).

### F.3 THE NUMBER OF FACTORS $m$

For the given VAE-based models, dimensional KL divergence (indicating the meaningful dimensions) increases during the training process. Therefore, we use $m$ dimensions of which the corresponding dimensional KL divergence $KL_i \geq T$, where $T$ is a hyperparameter, which is empirically set to 30 in our experiments.

# G DETAILS OF EXPERIMENTS

## G.1 DATASET DETAILS

In all the experiments, we resize the images to $64 \times 64$ resolution. We introduce all the datasets used in our paper in detail.

**dSprites** Higgins et al. (2017): dSprites contains 737,280 binary 2D shapes (heart, oval and square) images with five ground truth factors: shape (3 values), scale (6 values), orientation (40 values), x-position (32 values), y-position (32 values). Then we introduce the variants of dSprites (Color dSprites and Noisy dSprites) created by Locatello et al. Locatello et al. (2019b).

**Color dSprites**, the shapes of the images in dSprites are randomly colored.

**Noisy dSprites**, the background of the images in dSprites is noise.

**Shapes3D** Kim & Mnih (2018): Shapes3D contains 480,000 images of 3D shapes with 6 ground truth factors: shape (4 values), scale (8 values), orientation (15 values), floor color (10 values), wall color (15 values), object color (10 values).

**Cars3D** Reed et al. (2015): This dataset consists of 183 car CAD models, each rendered from 24 azimuth directions and 4 elevations.

| Encoder | Decoder |
|---|---|
| Input: $64 \times 64 \times$ number of channels | Input: 10 or 20 |
| $4 \times 4$ conv, 32 ReLU, stride 2 | FC, 256 ReLU |
| $4 \times 4$ conv, 32 ReLU, stride 2 | FC, 256 ReLU |
| $4 \times 4$ conv, 32 ReLU, stride 2 | FC, $4 \times 4 \times 32$ ReLU |
| $4 \times 4$ conv, 32 ReLU, stride 2 | $4 \times 4$ deconv, 32 ReLU, stride 2 |
| FC 256 ReLU | $4 \times 4$ deconv, 32 ReLU, stride 2 |
| FC 256 ReLU | $4 \times 4$ deconv, 32 ReLU, stride 2 |
| FC $2 \times 10$ | $4 \times 4$ deconv, number of channels, stride 2 |

Table 3: Architecture of the encoder and decoder of VAEs. For Original VAE, the dimension of input of the decoder is 10. For *Groupified* VAE, the dimension is 20. Note that the number of representation dimensions of *Groupified* VAE is still 10, which is the same as Original VAE, the comparison with Original VAE is fair Watters et al. (2019).

## G.2 ARCHITECTURE FOR ENCODER AND DECODER

We follow Locatello et al. Locatello et al. (2019b) to use the same architecture of VAEs in all of the experiments: the activation function used is ReLU except for the last layer of decoder, as shown in Table 3. For the details of the Discriminator in FactorVAE, please refer to Table 5 (a) and (c).

| Model | Parameter | Value |
|---|---|---|
| $\beta$-VAE | $\beta$ | [10; 20; 30;] |
| AnnealedVAE | C | [10; 20; 30;] |
| | start | [3e4; 4e4; 5e4;] |
| | end | [2e4; 3e4; 4e4;] |
| FactorVAE | $\gamma$ | [5; 10; 15] |
| $\beta$-TCVAE | $\beta$ | [6; 9; 12] |
| | random seed | [1; 2; 3; 4; 5; 6; 7; 8; 9;] |
| | group | [True; False] |

Table 4: Hyperparameters and random seeds for every model.

| (a) Optimizer for Discriminator | | (b) General hyperparameters for VAE | | (c) Architecture of Discriminator |
|---|---|---|---|---|
| Parameter | Values | Parameter | Values | Discriminator |
| Batch size | 64 | Batch size | 64 | FC, 1000 LReLU |
| Optimizer | Adam | Representation dimension | 10 | FC, 1000 LReLU |
| Adam: beta1 | 0.9 | Optimizer | Adam | FC, 1000 LReLU |
| Adam: beta2 | 0.999 | Adam: beta1 | 0.9 | FC, 1000 LReLU |
| Adam: epsilon | 1.0e-8 | Adam: beta2 | 0.999 | FC, 1000 LReLU |
| Adam: learning rate | 0.0001 | Adam: epsilon | 1.0e-8 | FC, 1000 LReLU |
| | | Adam: learning rate | 0.0001 | FC, 2 |
| | | Decoder type | Bernoulli | |

Table 5: Shared hyperparameters in all experiments. LReLU stands for leaky ReLU.

### G.3 Experiment settings

We run using different hyperparameters and random seeds for every VAE-based model implemented by Pytorch Paszke et al. (2017). As shown in Table 4, for $\beta$-VAE, we assign 3 choices for $\beta$ and 10 random seeds for both the Original and *Groupified* VAEs: $3 \times 10 \times 2 = 60$ settings for each dataset. Similarly, we also assign 60 settings for FactorVAE and $\beta$-TCVAE. For AnnealVAE, we assign three choices for $C$ and 3 choices for the start and end pair, also assign 10 random seeds. In summary, for all 5 datasets, we run $(((3 \times 10 \times 2) \times 3) + 3 \times 3 \times 10 \times 2) \times 5 = 1800$ models. For other hyperparameters, please refer to Table 5 (b).

## H Relation to some previous works

### H.1 Symmetry-based disentanglement

Caselles-Dupré et al. (2019) argue that the symmetry-based disentanglement requires interaction with the environments. Specifically, for a given disentangled representation $z$ w.r.t. a world with some group action on $W$, there are multiple other worlds (same world states and symmetry) of a static dataset that have different group actions on each dimension of $W$. For those worlds, the representation $z$ is not disentangled, per the group-based definition. However, in our work, we do not assume that a static dataset has a world equipped with some specific fixed group action on $W$ in advance, instead we use permutation group $\Phi$ as an agent to learn a world with proper group actions. For this learned world, there is only one representation that satisfies the definition. Therefore, we do not need an environment to provide group actions on $W$ to determine which world it is. Please see our proof in Theorem 1. In the testing phase, since we input the images to the model directly to derive their representation, the disentanglement of the representation does not rely on how the world state transitions between each other (i.e., as a result of group actions on $W$). Therefore, our framework can learn such a disentangled representation without interaction with the environments.

### H.2 Lie group VAE

Zhu et al. (2021) argue that the representation space being a vector space is sub-optimal since it requires the model to learn to discard different scales of variations. They propose to use lie group as the representation space instead and use Hessian Penalty to encourage disentanglement. Our framework is complimentary to their proposed method. The lie group representation can be applied to extend our framework. Here we leave it for future work. In addition, our proposed representation mapped by the sine and cosine function ($exp((2\pi i z)/n)$ for a real vector $z$) is also a lie group.

# I   MORE QUANTITATIVE RESULTS AND SCORE DISTRIBUTION

## I.1   QUANTITATIVE RESULTS

The performance of the Original and *Groupified* VAEs on all five datasets is shown in Table 6. Our method outperforms the original one on most of the datasets in terms of nearly all the metrics.

| dSprits | DCI | | BetaVAE | | MIG | | FactorVAE | |
|---|---|---|---|---|---|---|---|---|
| | Original | Groupified | Original | Groupified | Original | Groupified | Original | Groupified |
| $\beta$-VAE | $0.23 \pm 0.10$ | $\mathbf{0.46 \pm 0.085}$ | $0.75 \pm 0.083$ | $\mathbf{0.86 \pm 0.051}$ | $0.14 \pm 0.097$ | $\mathbf{0.37 \pm 0.089}$ | $0.51 \pm 0.098$ | $\mathbf{0.63 \pm 0.089}$ |
| AnnealVAE | $0.28 \pm 0.10$ | $\mathbf{0.39 \pm 0.056}$ | $0.84 \pm 0.050$ | $\mathbf{0.87 \pm 0.0067}$ | $0.23 \pm 0.10$ | $\mathbf{0.34 \pm 0.061}$ | $\mathbf{0.70 \pm 0.094}$ | $0.68 \pm 0.058$ |
| FactorVAE | $0.38 \pm 0.10$ | $\mathbf{0.41 \pm 0.074}$ | $0.89 \pm 0.040$ | $\mathbf{0.89 \pm 0.020}$ | $0.27 \pm 0.092$ | $\mathbf{0.31 \pm 0.061}$ | $0.74 \pm 0.068$ | $\mathbf{0.75 \pm 0.075}$ |
| $\beta$-TCVAE | $0.35 \pm 0.065$ | $\mathbf{0.36 \pm 0.11}$ | $0.86 \pm 0.026$ | $\mathbf{0.861 \pm 0.038}$ | $0.17 \pm 0.067$ | $\mathbf{0.24 \pm 0.093}$ | $0.68 \pm 0.098$ | $\mathbf{0.70 \pm 0.098}$ |

| Cars3d | DCI | | BetaVAE | | MIG | | FactorVAE | |
|---|---|---|---|---|---|---|---|---|
| | Original | Groupified | Original | Groupified | Original | Groupified | Original | Groupified |
| $\beta$-VAE | $0.18 \pm 0.059$ | $\mathbf{0.24 \pm 0.041}$ | $0.99 \pm 1.6e-3$ | $\mathbf{1.0 \pm 0.0}$ | $0.071 \pm 0.032$ | $\mathbf{0.11 \pm 0.032}$ | $0.81 \pm 0.066$ | $\mathbf{0.93 \pm 0.034}$ |
| AnnealVAE | $0.22 \pm 0.046$ | $\mathbf{0.25 \pm 0.046}$ | $0.99 \pm 4e-4$ | $\mathbf{0.99 \pm 1.5e-4}$ | $0.074 \pm 0.016$ | $\mathbf{0.10 \pm 0.014}$ | $0.82 \pm 0.062$ | $\mathbf{0.87 \pm 0.028}$ |
| FactorVAE | $0.21 \pm 0.054$ | $\mathbf{0.25 \pm 0.040}$ | $0.99 \pm 1e-4$ | $\mathbf{1.0 \pm 0.0}$ | $0.098 \pm 0.027$ | $\mathbf{0.11 \pm 0.033}$ | $0.90 \pm 0.039$ | $\mathbf{0.93 \pm 0.034}$ |
| $\beta$-TCVAE | $0.24 \pm 0.049$ | $\mathbf{0.26 \pm 0.046}$ | $\mathbf{1.0 \pm 0.0}$ | $\mathbf{1.0 \pm 0.0}$ | $0.10 \pm 0.021$ | $\mathbf{0.11 \pm 0.033}$ | $0.88 \pm 0.040$ | $\mathbf{0.93 \pm 0.034}$ |

| Noisy dSprits | DCI | | betaVAE | | MIG | | FactorVAE | |
|---|---|---|---|---|---|---|---|---|
| | Original | Groupified | Original | Groupified | Original | Groupified | Original | Groupified |
| BetaVAE | $0.056 \pm 0.018$ | $\mathbf{0.087 \pm 0.051}$ | $0.624 \pm 0.090$ | $\mathbf{0.647 \pm 0.055}$ | $0.030 \pm 0.022$ | $\mathbf{0.065 \pm 0.055}$ | $0.355 \pm 0.093$ | $\mathbf{0.407 \pm 0.071}$ |
| Anneal VAE | $0.053 \pm 0.013$ | $\mathbf{0.060 \pm 0.022}$ | $0.631 \pm 0.036$ | $\mathbf{0.644 \pm 0.031}$ | $0.035 \pm 0.027$ | $\mathbf{0.047 \pm 0.032}$ | $0.434 \pm 0.080$ | $\mathbf{0.481 \pm 0.087}$ |
| FactorVAE | $\mathbf{0.114 \pm 0.062}$ | $0.099 \pm 0.057$ | $0.682 \pm 0.081$ | $\mathbf{0.684 \pm 0.070}$ | $\mathbf{0.077 \pm 0.046}$ | $0.066 \pm 0.046$ | $0.437 \pm 0.098$ | $\mathbf{0.468 \pm 0.098}$ |
| $\beta$-TCVAE | $0.081 \pm 0.036$ | $\mathbf{0.111 \pm 0.053}$ | $0.605 \pm 0.053$ | $\mathbf{0.635 \pm 0.050}$ | $0.040 \pm 0.030$ | $\mathbf{0.068 \pm 0.042}$ | $0.353 \pm 0.091$ | $\mathbf{0.431 \pm 0.097}$ |

| Shapes3d | DCI | | BetaVAE | | MIG | | FactorVAE | |
|---|---|---|---|---|---|---|---|---|
| | Original | Groupified | Original | Groupified | Original | Groupified | Original | Groupified |
| $\beta$-VAE | $0.44 \pm 0.176$ | $\mathbf{0.56 \pm 0.10}$ | $0.91 \pm 0.072$ | $0.90 \pm \mathbf{0.045}$ | $0.28 \pm 0.18$ | $\mathbf{0.42 \pm 0.15}$ | $0.82 \pm 0.098$ | $\mathbf{0.82 \pm 0.043}$ |
| AnnealVAE | $0.52 \pm 0.051$ | $\mathbf{0.60 \pm 0.078}$ | $0.82 \pm \mathbf{0.076}$ | $\mathbf{0.89 \pm 0.086}$ | $0.48 \pm 0.047$ | $\mathbf{0.50 \pm 0.052}$ | $0.75 \pm 0.074$ | $\mathbf{0.83 \pm 0.066}$ |
| FactorVAE | $0.47 \pm 0.10$ | $\mathbf{0.49 \pm 0.065}$ | $0.86 \pm \mathbf{0.055}$ | $0.80 \pm 0.075$ | $0.33 \pm 0.13$ | $\mathbf{0.43 \pm 0.11}$ | $0.81 \pm 0.056$ | $0.79 \pm 0.066$ |
| $\beta$-TCVAE | $0.66 \pm 0.10$ | $\mathbf{0.72 \pm 0.061}$ | $0.97 \pm 0.039$ | $0.96 \pm 0.042$ | $0.40 \pm 0.18$ | $\mathbf{0.47 \pm 0.090}$ | $0.89 \pm 0.064$ | $\mathbf{0.90 \pm 0.046}$ |

| Color dSprits | DCI | | betaVAE | | MIG | | FactorVAE | |
|---|---|---|---|---|---|---|---|---|
| | Original | Groupified | Original | Groupified | Original | Groupified | Original | Groupified |
| BetaVAE | $0.174 \pm 0.097$ | $\mathbf{0.328 \pm 0.130}$ | $0.798 \pm 0.094$ | $\mathbf{0.844 \pm 0.050}$ | $0.103 \pm 0.058$ | $\mathbf{0.243 \pm 0.118}$ | $0.591 \pm 0.148$ | $\mathbf{0.648 \pm 0.092}$ |
| Anneal VAE | $0.268 \pm 0.103$ | $\mathbf{0.337 \pm 0.114}$ | $0.843 \pm 0.038$ | $\mathbf{0.856 \pm 0.031}$ | $0.219 \pm 0.084$ | $\mathbf{0.252 \pm 0.104}$ | $\mathbf{0.718 \pm 0.065}$ | $0.692 \pm 0.094$ |
| FactorVAE | $0.294 \pm 0.101$ | $\mathbf{0.322 \pm 0.104}$ | $0.861 \pm 0.038$ | $\mathbf{0.862 \pm 0.029}$ | $0.203 \pm 0.080$ | $\mathbf{0.236 \pm 0.091}$ | $\mathbf{0.739 \pm 0.068}$ | $0.730 \pm 0.080$ |
| $\beta$-TCVAE | $0.338 \pm 0.052$ | $\mathbf{0.395 \pm 0.082}$ | $0.876 \pm 0.024$ | $\mathbf{0.881 \pm 0.031}$ | $0.169 \pm 0.040$ | $\mathbf{0.269 \pm 0.090}$ | $0.711 \pm 0.086$ | $\mathbf{0.786 \pm 0.050}$ |

Table 6: Performance (mean $\pm$ std) on different datasets and different models with different metrics. We evaluate $\beta$-VAE, AnnealVAE, FactorVAE, and $\beta$-TCVAE on dSprites, Cars3d, Shapes3d, Noisy-dSprites, and Color-dSprites for 1800 settings. These settings include different random seeds and hyperparameters.

## I.2   ABSTRACT REASONING & FAIRNESS

As pointed out by Locatello et al. Locatello et al. (2019b), the disentangled representation's downstream tasks should also be verified. Therefore, we verify the effectiveness of the representations learned by the *Groupified* VAEs on Shapes3d in two downstream tasks: abstract reasoning Van Steenkiste et al. (2019) and fairness evaluation Locatello et al. (2019a). As Table 7 shows, the performance of the abstract reasoning models fine-tuned on the representation learned by the *Groupified* FactorVAEs is better than the original ones. In terms of fairness evaluation, we can observe that the unfairness scores of the representation learned by the *Groupified* FactorVAEs are lower than the Original ones.

| | Abstract reasoning$\uparrow$ | Unfairness scores$\downarrow$ |
|---|---|---|
| Original | $0.948 \pm 0.031$ | $0.023 \pm 0.007$ |
| Groupified | $\mathbf{0.954 \pm 0.028}$ | $\mathbf{0.018 \pm 0.008}$ |

Table 7: Downstream task performance on the models trained on the representation learned by original and groupified FactorVAE.

## I.3   SCORE DISTRIBUTION

The detailed distribution of the performance is shown in this section (demonstrated by the Violin Plot Hintze & Nelson (1998)). The performance distributions on dSprits, Car3d, Noisy dSprites,

Color-dSprites, and Shapes3d are shown in Figure 8, Figure 9, Figure 10, Figure 11 and Figure 12, respectively.

## I.4 COMPARISON WITH METHOD INTERACTION WITH THE ENVIRONMENT

Our work considers an unsupervised setting, which is a more practical one. To understand the price to pay, we compare our unsupervised *Groupified* models to the methods that use the interaction with the environment as supervision. Here we provide the comparison between RGrVAE (Painter et al., 2020) and our *Groupified* $\beta$-TCVAE on dSprites, Shapes3D, and Color dSprites as shown in Table 8.

| Datasets | DCI | | BetaVAE | | MIG | | FactorVAE | |
|---|---|---|---|---|---|---|---|---|
| | RGrVAE | Groupified | RGrVAE | Groupified | RGrVAE | Groupified | RGrVAE | Groupified |
| dSprites | $0.52 \pm 0.058$ | $0.36 \pm 0.110$ | $0.97 \pm 0.039$ | $0.86 \pm 0.038$ | $0.08 \pm 0.042$ | $0.24 \pm 0.093$ | $0.86 \pm 0.073$ | $0.70 \pm 0.098$ |
| Shapes3D | $0.83 \pm 0.056$ | $0.72 \pm 0.061$ | $1.00 \pm 0.000$ | $0.96 \pm 0.042$ | $0.25 \pm 0.031$ | $0.47 \pm 0.090$ | $0.98 \pm 0.032$ | $0.90 \pm 0.046$ |
| Color dSprites | $0.11 \pm 0.072$ | $0.40 \pm 0.082$ | $0.53 \pm 0.294$ | $0.88 \pm 0.031$ | $0.03 \pm 0.028$ | $0.27 \pm 0.090$ | $0.31 \pm 0.309$ | $0.79 \pm 0.050$ |

Table 8: Performance (mean $\pm$ variance) on different datasets of RGrVAE and *Groupified* $\beta$-TCVAE with different metrics. These settings include different random seeds and hyperparameters.

Since there are no results reported on Shapes3D and Color dSprites, we conduct experiments on these two datasets with the official implementation[2] using the recommended hyper-parameters. In addition, the results reported in Painter et al. (2020) are of models trained with 16 latent units and 3 random seeds. We also conduct experiments on dSprites with 10 latent units and 10 random seeds, which is our setting. From Table 8, we observe that there is still a gap between RGrVAE and *Groupified* $\beta$-TCVAE, especially on Shapes3D. However, the latent learned by RGrVAE is not as pure as *Groupified* $\beta$-TCVAE (lower MIG). Additionally, RGrVAE performs poorly because a factor (color) is not modeled in the environment of Color dSprites.

## I.5 COMPARISON WITH CONTROLVAE

In this section, we provide a comparison between Original and *Groupified* ControlVAE (Shao et al., 2020). For ControlVAE, we use the official implementation[3] and follow the default setting, $C_{max} = 25$. We follow Locatello et al. (2019b) to set the hyperparameter interval to 10. For other parameters, we follow Shao et al. (2020). The results on dSprites, Shapes3D, and Color dSprites are presented in Table 9.

| Datasets | DCI | | BetaVAE | | MIG | | FactorVAE | |
|---|---|---|---|---|---|---|---|---|
| | Original | Groupified | Original | Groupified | Original | Groupified | Original | Groupified |
| dSprites | $0.31 \pm 0.093$ | $0.46 \pm 0.115$ | $0.83 \pm 0.084$ | $0.92 \pm 0.061$ | $0.16 \pm 0.062$ | $0.27 \pm 0.123$ | $0.62 \pm 0.087$ | $0.74 \pm 0.110$ |
| Shapes3D | $0.59 \pm 0.144$ | $0.85 \pm 0.165$ | $0.86 \pm 0.142$ | $0.97 \pm 0.149$ | $0.21 \pm 0.212$ | $0.72 \pm 0.151$ | $0.59 \pm 0.174$ | $0.88 \pm 0.171$ |
| Color dSprites | $0.47 \pm 0.111$ | $0.54 \pm 0.055$ | $0.94 \pm 0.045$ | $0.96 \pm 0.024$ | $0.28 \pm 0.095$ | $0.35 \pm 0.055$ | $0.80 \pm 0.070$ | $0.84 \pm 0.022$ |

Table 9: Performance (mean $\pm$ variance) on different datasets of ControlVAE and *Groupified* ControlVAE with different metrics. These settings include different random seeds and hyperparameters.

From Table 9, we observe that our method consistently improved the performance of ControlVAE under the same hyper-parameters, especially on Shapes3D.

## I.6 DETAILED RESULTS UNDER DIFFERENT HYPER-PARAMETERS

In order to provide a more convincing comparison, we compare our method and the original one at different levels of regularization parameters. We take $\beta$-TCVAE as an example. As Table 10, 11, and 12 show, our method consistently improves the performance of the original methods on most of the metrics, especially on MIG and DCI.

---

[2] https://github.com/MattPainter01/UnsupervisedActionEstimation
[3] https://github.com/shj1987/ControlVAE-ICML2020

| Regulize strength | DCI | | BetaVAE | | MIG | | FactorVAE | |
|---|---|---|---|---|---|---|---|---|
| | Original | Groupified | Original | Groupified | Original | Groupified | Original | Groupified |
| $\beta = 6$ | $0.33 \pm 0.079$ | $0.34 \pm 0.106$ | $0.86 \pm 0.025$ | $0.85 \pm 0.032$ | $0.15 \pm 0.042$ | $0.20 \pm 0.093$ | $0.68 \pm 0.100$ | $0.68 \pm 0.105$ |
| $\beta = 9$ | $0.37 \pm 0.051$ | $0.36 \pm 0.103$ | $0.85 \pm 0.028$ | $0.87 \pm 0.026$ | $0.17 \pm 0.053$ | $0.24 \pm 0.097$ | $0.63 \pm 0.092$ | $0.70 \pm 0.072$ |
| $\beta = 12$ | $0.35 \pm 0.057$ | $0.38 \pm 0.114$ | $0.87 \pm 0.024$ | $0.86 \pm 0.051$ | $0.20 \pm 0.071$ | $0.30 \pm 0.098$ | $0.71 \pm 0.084$ | $0.71 \pm 0.104$ |

Table 10: Performance (mean $\pm$ variance) on dSprites of Original and *Groupified* $\beta$-TCVAE with different metrics. The results under different regularize strength are reported.

| Regulize strength | DCI | | BetaVAE | | MIG | | FactorVAE | |
|---|---|---|---|---|---|---|---|---|
| | Original | Groupified | Original | Groupified | Original | Groupified | Original | Groupified |
| $\beta = 6$ | $0.56 \pm 0.080$ | $0.68 \pm 0.071$ | $0.98 \pm 0.028$ | $0.97 \pm 0.036$ | $0.31 \pm 0.123$ | $0.42 \pm 0.098$ | $0.88 \pm 0.046$ | $0.90 \pm 0.052$ |
| $\beta = 9$ | $0.67 \pm 0.089$ | $0.76 \pm 0.048$ | $0.97 \pm 0.037$ | $0.96 \pm 0.048$ | $0.38 \pm 0.196$ | $0.51 \pm 0.084$ | $0.90 \pm 0.078$ | $0.89 \pm 0.045$ |
| $\beta = 12$ | $0.75 \pm 0.053$ | $0.73 \pm 0.026$ | $0.96 \pm 0.044$ | $0.93 \pm 0.048$ | $0.51 \pm 0.151$ | $0.51 \pm 0.082$ | $0.92 \pm 0.059$ | $0.90 \pm 0.040$ |

Table 11: Performance (mean $\pm$ variance) on Shapes3D of Original and *Groupified* $\beta$-TCVAE with different metrics. The results under different regularize strength are reported.

| Regulize strength | DCI | | BetaVAE | | MIG | | FactorVAE | |
|---|---|---|---|---|---|---|---|---|
| | Original | Groupified | Original | Groupified | Original | Groupified | Original | Groupified |
| $\beta = 6$ | $0.35 \pm 0.051$ | $0.43 \pm 0.073$ | $0.89 \pm 0.009$ | $0.89 \pm 0.016$ | $0.17 \pm 0.038$ | $0.29 \pm 0.073$ | $0.73 \pm 0.094$ | $0.82 \pm 0.027$ |
| $\beta = 9$ | $0.36 \pm 0.019$ | $0.40 \pm 0.077$ | $0.89 \pm 0.013$ | $0.88 \pm 0.019$ | $0.17 \pm 0.037$ | $0.27 \pm 0.104$ | $0.74 \pm 0.077$ | $0.79 \pm 0.048$ |
| $\beta = 12$ | $0.31 \pm 0.061$ | $0.35 \pm 0.081$ | $0.86 \pm 0.031$ | $0.87 \pm 0.020$ | $0.16 \pm 0.044$ | $0.24 \pm 0.083$ | $0.66 \pm 0.060$ | $0.75 \pm 0.051$ |

Table 12: Performance (mean $\pm$ variance) on Color dSprites of Original and *Groupified* $\beta$-TCVAE with different metrics. The results under different regularize strength are reported.

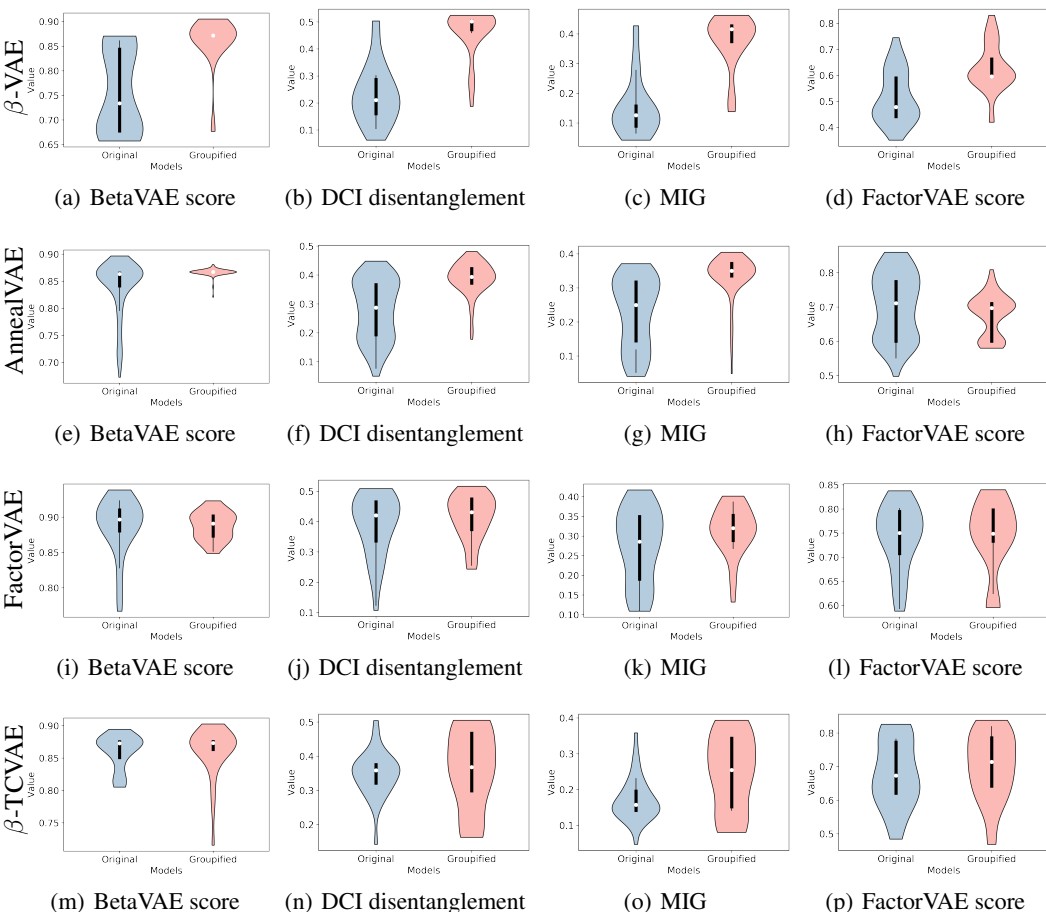

(a) BetaVAE score    (b) DCI disentanglement    (c) MIG    (d) FactorVAE score

(e) BetaVAE score    (f) DCI disentanglement    (g) MIG    (h) FactorVAE score

(i) BetaVAE score    (j) DCI disentanglement    (k) MIG    (l) FactorVAE score

(m) BetaVAE score    (n) DCI disentanglement    (o) MIG    (p) FactorVAE score

Figure 8: Performance distribution on dSprites. Variance is due to different hyperparameters and random seeds. We consider four metrics: BetaVAE score, DCI disentanglement, MIG, and FactorVAE score. We observe that *Groupified* VAEs outperform the original ones.

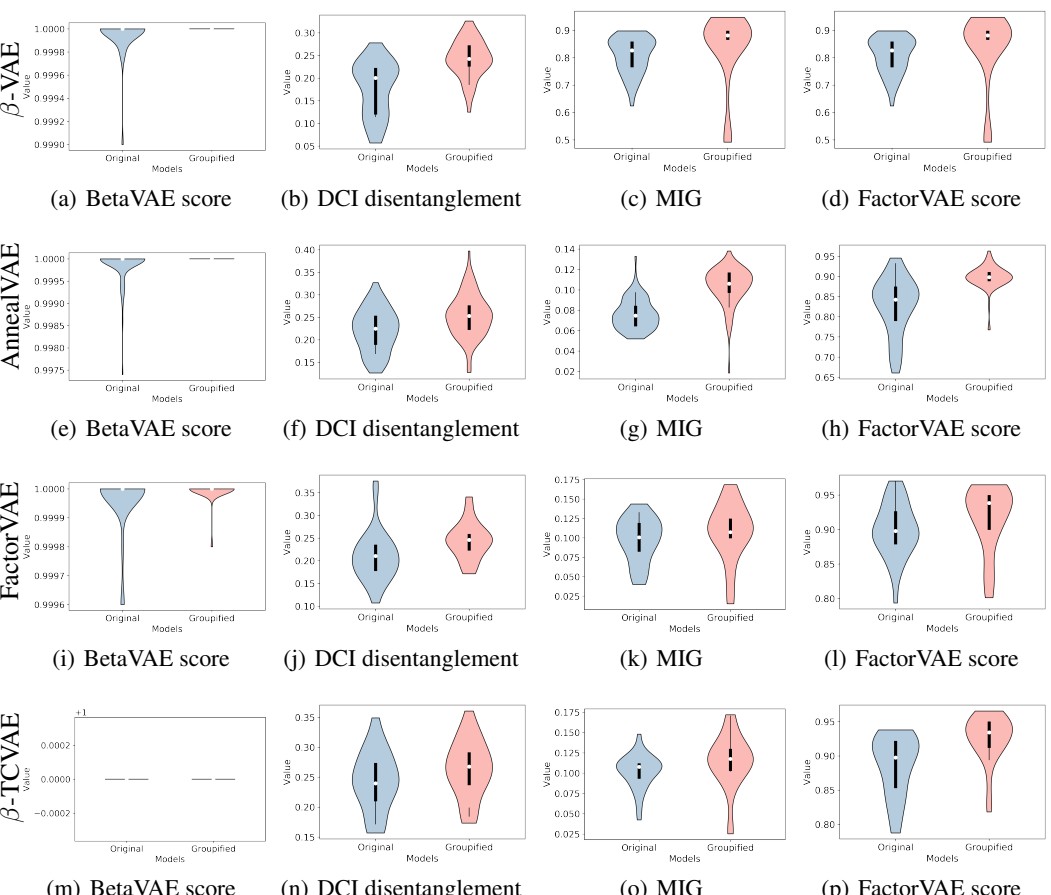

Figure 9: Performance distribution on Cars3d. Variance is due to different hyperparameters and random seeds. We observe that *Groupified* models outperform the Original ones.

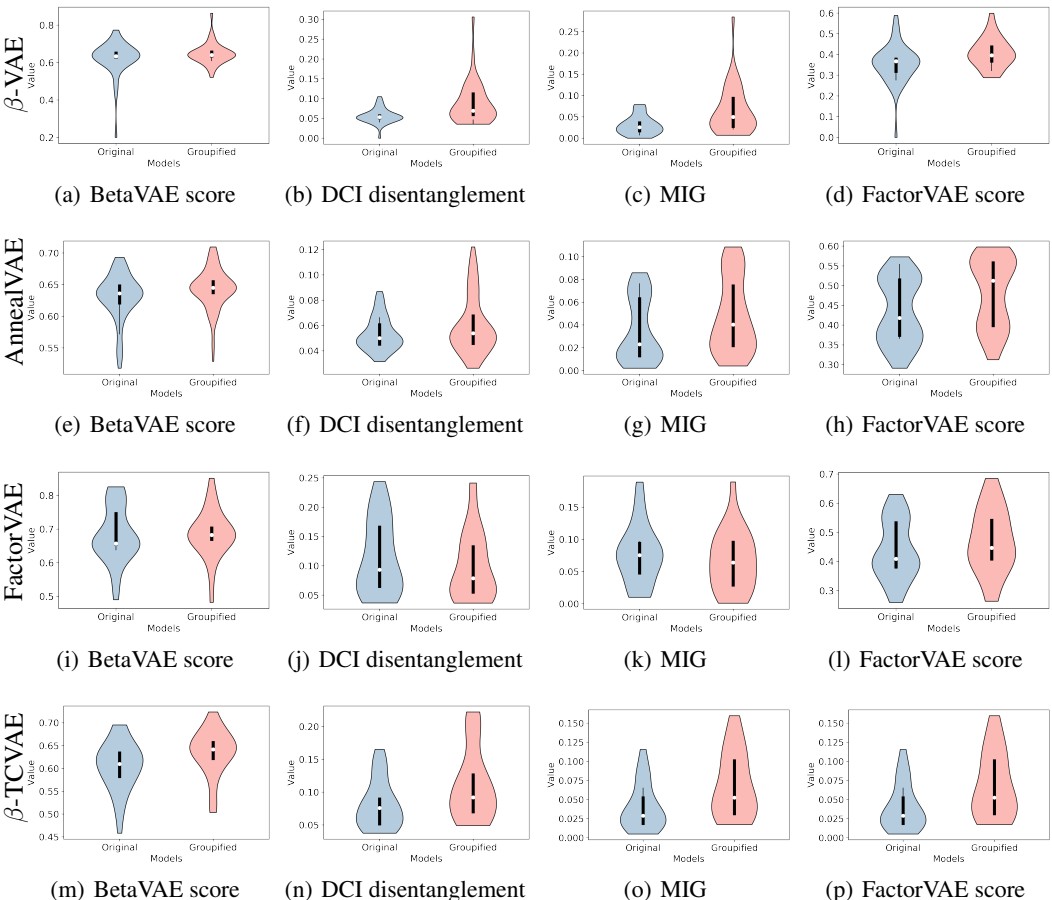

(a) BetaVAE score     (b) DCI disentanglement     (c) MIG     (d) FactorVAE score

(e) BetaVAE score     (f) DCI disentanglement     (g) MIG     (h) FactorVAE score

(i) BetaVAE score     (j) DCI disentanglement     (k) MIG     (l) FactorVAE score

(m) BetaVAE score     (n) DCI disentanglement     (o) MIG     (p) FactorVAE score

Figure 10: Performance distribution on Noisy dSprites. Variance is due to different hyperparameters and random seeds. We observe that *Groupified* VAEs outperform the Original ones.

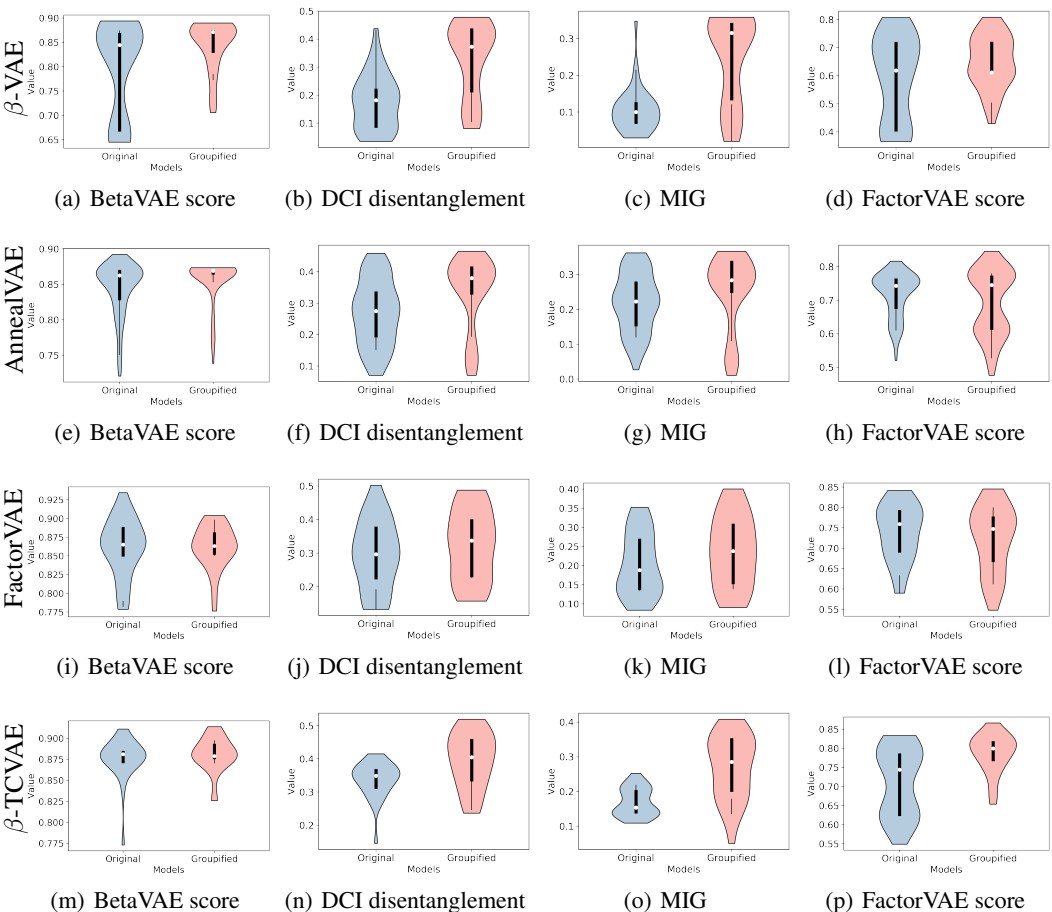

Figure 11: Performance distribution on Color dSprites. Variance is due to different hyperparameters and random seeds. We observe that *Groupified* VAEs outperform the Original ones.

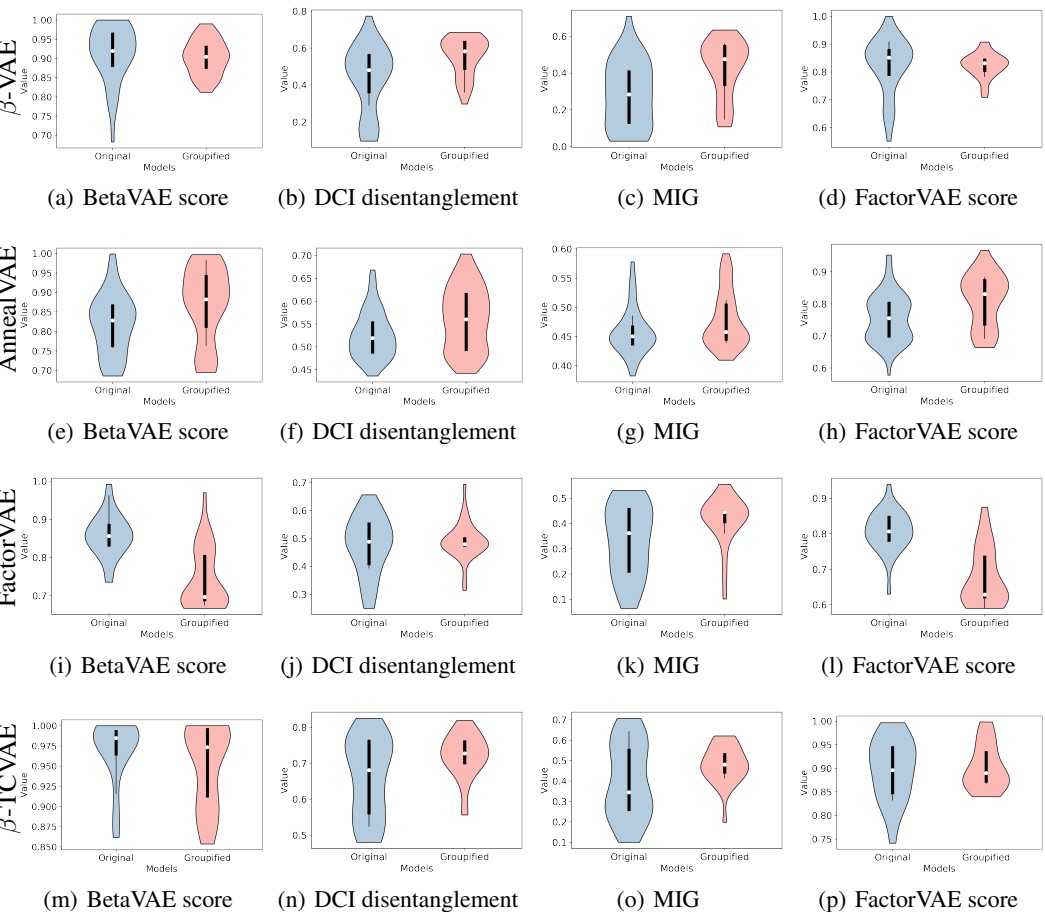

Figure 12: Performance distribution on Shapes3d. Variance is due to different hyperparameters and random seeds. We observe that *Groupified* models outperform the Original ones.

## J    MORE QUALITATIVE RESULTS

We evaluate our methods qualitatively on two typical datasets: Cars3d and Shapes3d. We visualize the traversal results of the Original and *Groupified* FactorVAE and $\beta$-TCVAE. For every factor, we traverse five randomly sampled representations. As shown in Figure 13 and Figure 14, the traversal results of the *Groupified* FactorVAE and $\beta$-TCVAE show that these models learn less entangled representations on Shapes3D (e.g., Orientation of FactorVAE and Scale and Shape of $\beta$-TCVAE).

Similarly, as shown in Figure 15 and Figure 16, the *Groupified* FactorVAE and $\beta$-TCVAE achieve better disentanglement ability on Car3d (e.g., Rotation of FactorVAE and Yaw of $\beta$-TCVAE).

For the real-world dataset, we show the qualitative comparison of the *Groupified* and original AnnealVAE trained on CelebA. As Figure 17 shows, for most of the factors, our model can extract cleaner semantics than the original model. For example, the hair color is entangled with the face shape in the original model, but cleaner in the *Groupified* model.

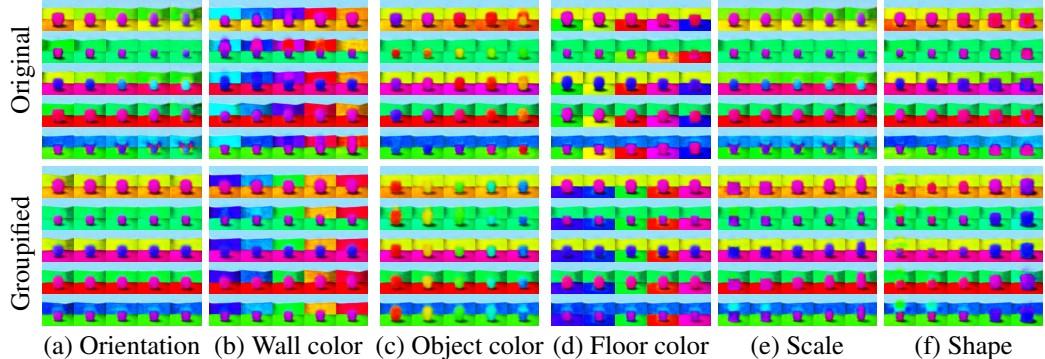

(a) Orientation  (b) Wall color  (c) Object color  (d) Floor color  (e) Scale  (f) Shape

Figure 13:  Learned latent variables using Original and *Groupified* FactorVAE on Shapes3d dataset. The traversal range is (-2, 2).

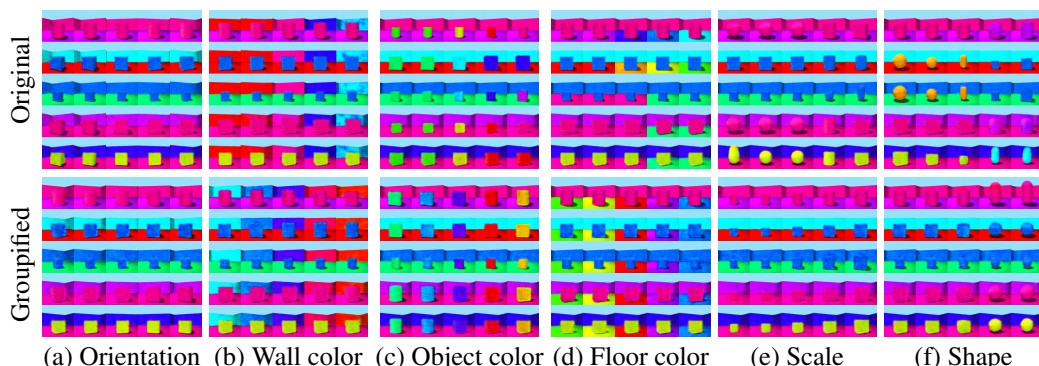

(a) Orientation  (b) Wall color  (c) Object color  (d) Floor color  (e) Scale  (f) Shape

Figure 14: Learned latent variables using Original and *Groupified* $\beta$-TCVAE on Shapes3d dataset. The traversal range is (-2, 2).

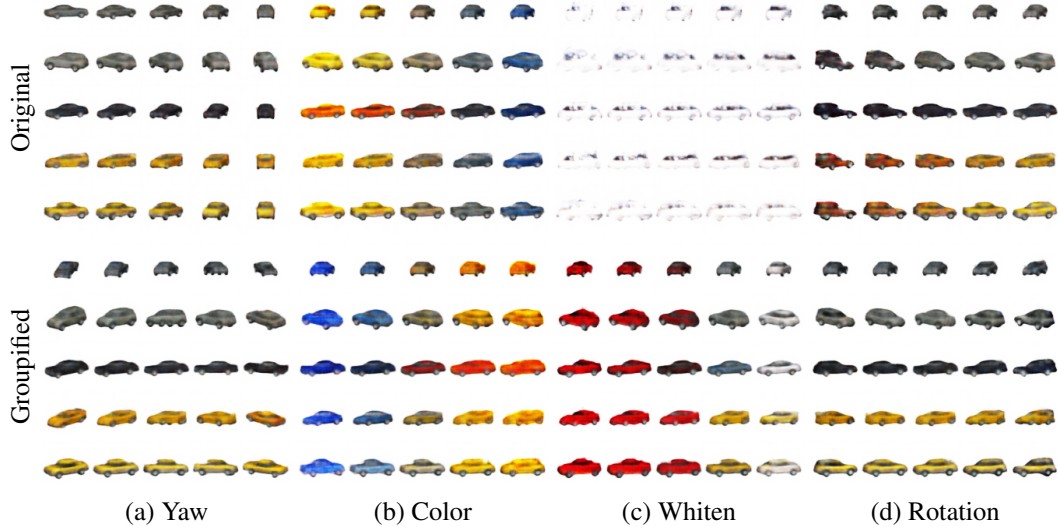

Figure 15: Learned latent variables using Original and *Groupified* FactorVAE on Car3d dataset. The traversal range is (-2, 2).

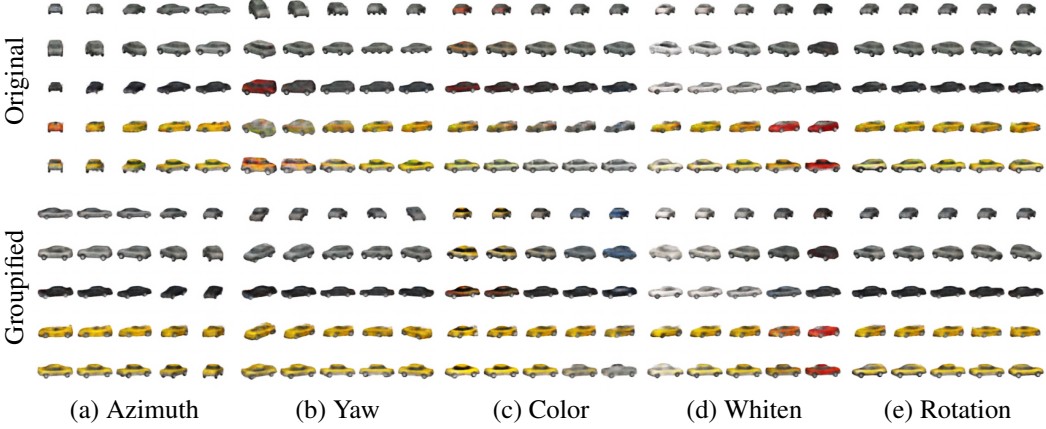

Figure 16: Learned latent variables using Original and *Groupified* $\beta$-TCVAE on Car3d dataset. The traversal range is (-2, 2).

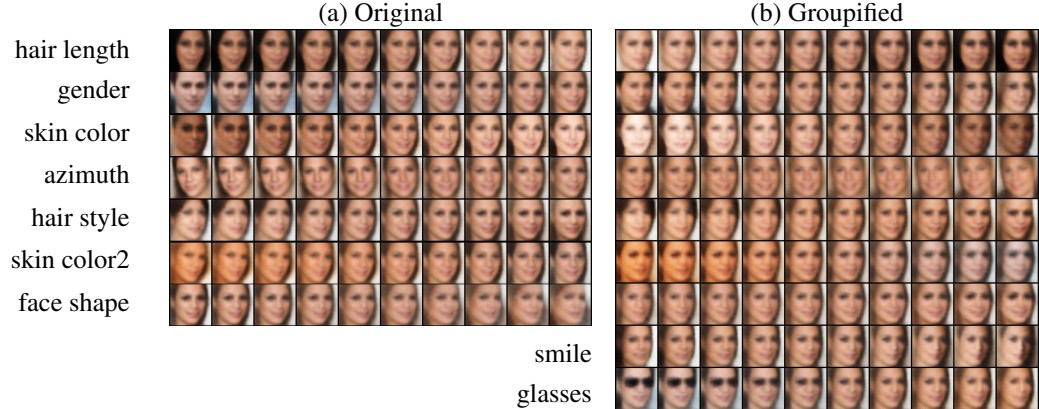

Figure 17: Learned latent variables using Original and *Groupified* AnnealVAE on CeleBa dataset. The traversal range is (-2, 2). The factors learned by *Groupified* model are less entangled.

# K    MEANINGFUL DIMENSION VISUALIZATIONS

When we assign some dimensions to Isomorphism Loss for *Groupified* AnnealVAEs, e.g., the first-5 dimensions, we have an interesting observation that the assigned dimensions are meaningful in *Groupified* AnnealVAEs. The KL divergence increases continuously on these assigned dimensions after the Isomorphism Loss is applied to them. Note that the KL divergence loss in AnnealVAE indicates the amount of information encoded. As Figure 18 shows, the KL divergence of assigned dimensions increases at the beginning of training, which means Isomorphism Loss results in that the assigned dimensions become meaningful. Finally, the assigned first-five dimensions learn to encode the semantics of x position, y position, scale, and orientations. The possible reason is that the latent factors are learned and disentangled in the assigned dimensions due to the punishment of the Isomorphism Loss. To illustrate that controllable dimensions in *Groupified* AnnealVAE are not an exception, we provide more visualizations. The results of covering all hyperparameters settings with two random seeds are shown in Figure 18 to Figure 26, suggesting that their dimensions are controllable.

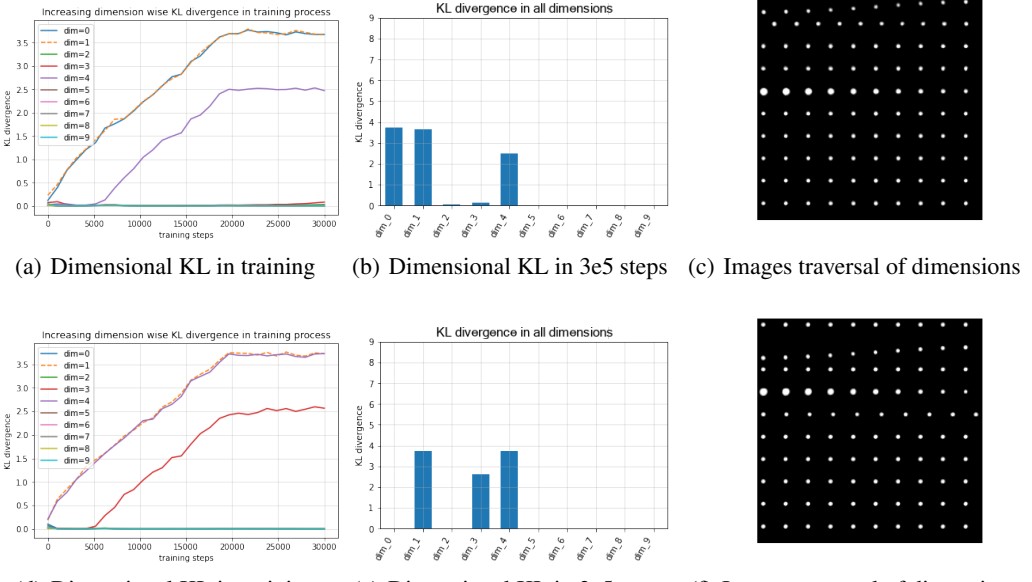

(a) Dimensional KL in training     (b) Dimensional KL in 3e5 steps   (c) Images traversal of dimensions

(d) Dimensional KL in training     (e) Dimensional KL in 3e5 steps   (f) Images traversal of dimensions

Figure 18: Meaningful dimensions visualization for $C = 10$, end = 30000 (different random seeds). The KL divergences of target dimensions (0-4 dimension) increase one by one during training (a). The KL divergences in different dimensions are different after training (b). As the image traversal results (c) show, the meaningful dimensions are learned in 0-4 dims. So are (d), (e), and (f), which are results of a run with a different random seed.

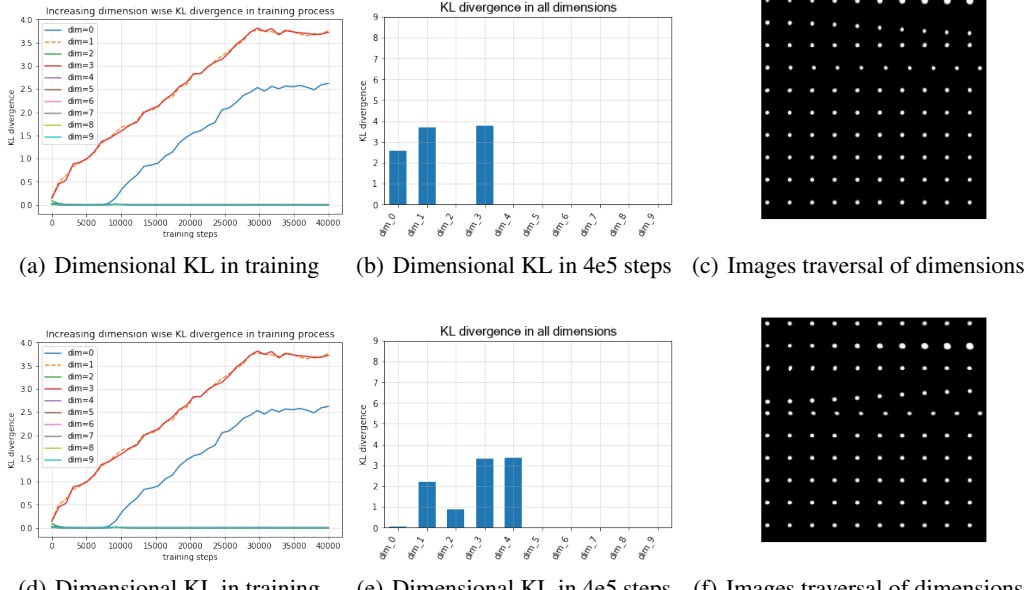

(a) Dimensional KL in training    (b) Dimensional KL in 4e5 steps    (c) Images traversal of dimensions

(d) Dimensional KL in training    (e) Dimensional KL in 4e5 steps    (f) Images traversal of dimensions

Figure 19: Meaningful dimensions visualization for $C = 10$, end = 40000 (different random seeds). The KL divergences of target dimensions (0-4 dimension) increase one by one during training (a). The KL divergences in different dimensions are different amounts after training (b). As the image traversal results (c) show, the meaningful dimensions are learned in 0-4 dims. So as (d), (e), and (f), which are results of a run with a different random seed.

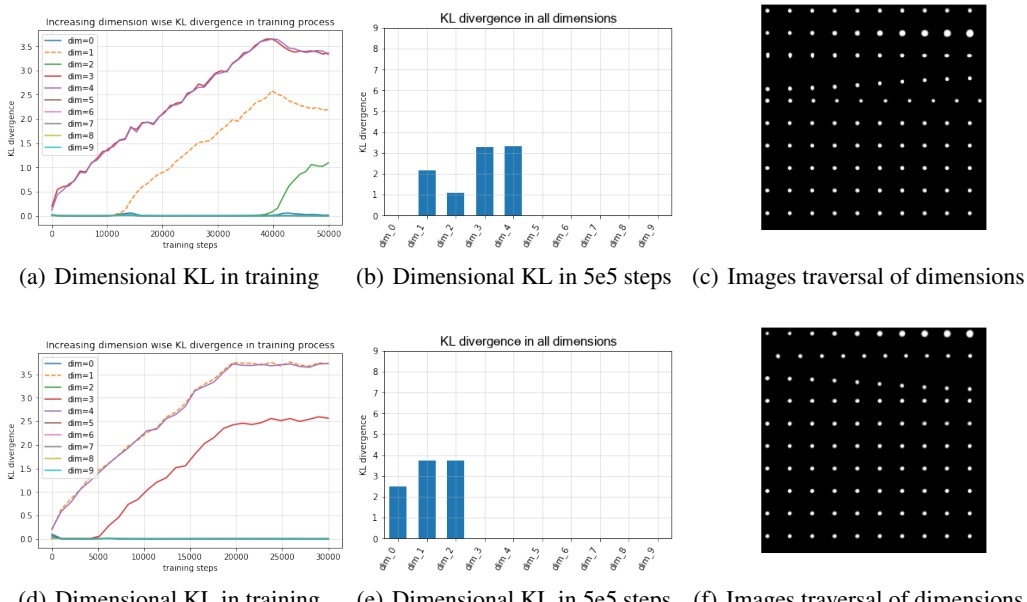

(a) Dimensional KL in training    (b) Dimensional KL in 5e5 steps    (c) Images traversal of dimensions

(d) Dimensional KL in training    (e) Dimensional KL in 5e5 steps    (f) Images traversal of dimensions

Figure 20: Meaningful dimensions visualization for $C = 10$, end = 50000 (different random seeds). The KL divergences of target dimensions (0-4 dimension) increase one by one during training (a). The KL divergences in different dimensions are different after training (b). As the image traversal results (c) show, the meaningful dimensions are learned in 0-4 dims. So are (d), (e), and (f), which are results of a run with a different random seed.

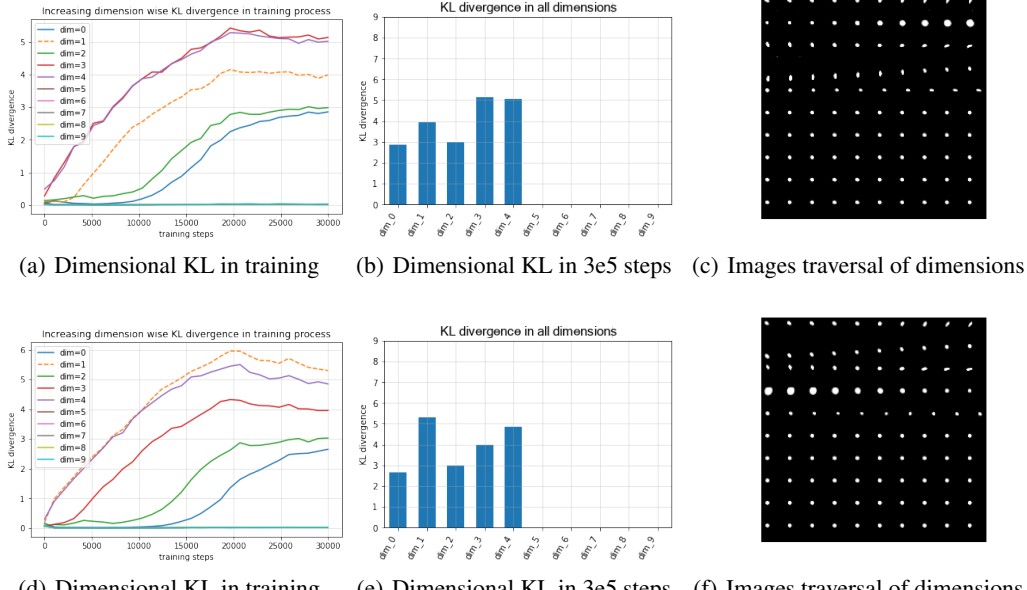

(a) Dimensional KL in training     (b) Dimensional KL in 3e5 steps     (c) Images traversal of dimensions

(d) Dimensional KL in training     (e) Dimensional KL in 3e5 steps     (f) Images traversal of dimensions

Figure 21: Meaningful dimensions visualization for $C = 20$, end = 30000 (different random seeds). The KL divergences of the target dimension (0-4 dimension) increase one by one during training (a). The KL divergences in different dimensions are different amounts after training (b). As the image traversal results (c) show, the meaningful dimensions are learned in 0-4 dims. So as (d), (e), and (f), which have a different random seed.

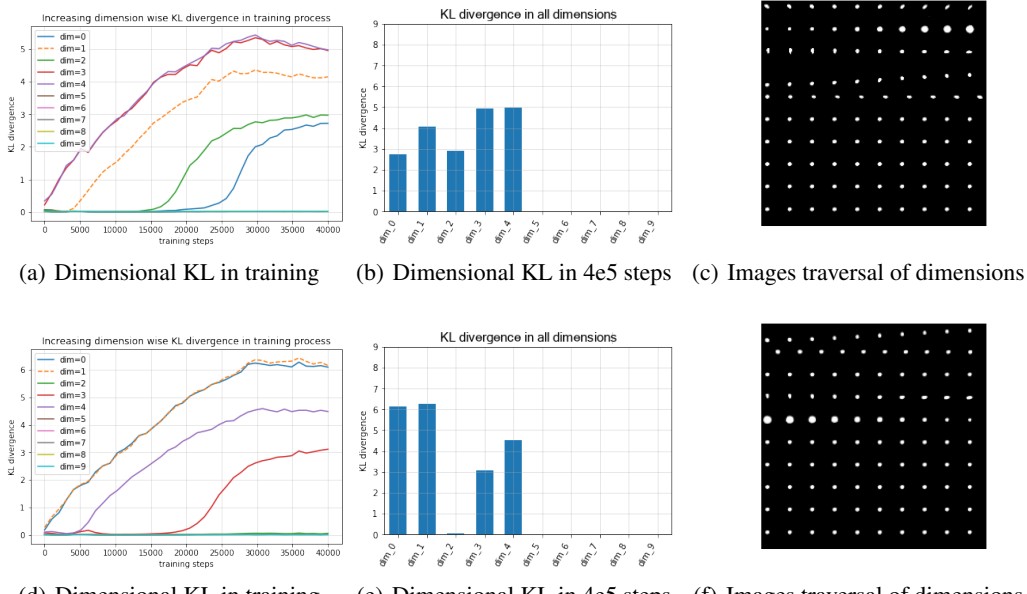

(a) Dimensional KL in training     (b) Dimensional KL in 4e5 steps     (c) Images traversal of dimensions

(d) Dimensional KL in training     (e) Dimensional KL in 4e5 steps     (f) Images traversal of dimensions

Figure 22: Meaningful dimensions visualization for $C = 20$, end = 40000 (different random seeds). The KL divergences of target dimensions (0-4 dimension) increase one by one during training (a). The KL divergences in different dimensions are different after training (b). As the image traversal results (c) show, the meaningful dimensions are learned in 0-4 dims. So are (d), (e), and (f), which are results of a run with a different random seed.

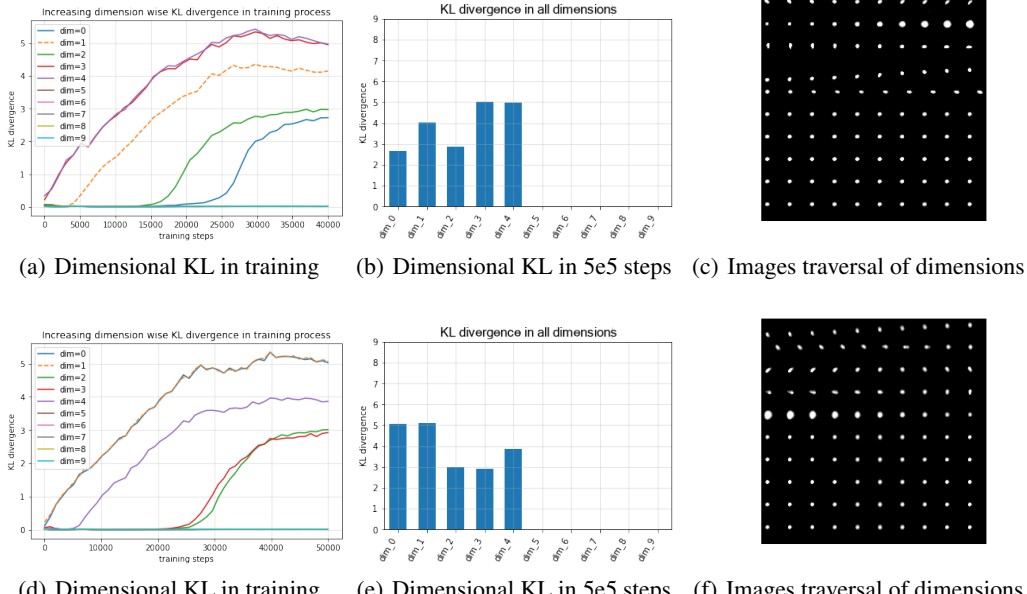

(a) Dimensional KL in training     (b) Dimensional KL in 5e5 steps     (c) Images traversal of dimensions

(d) Dimensional KL in training     (e) Dimensional KL in 5e5 steps     (f) Images traversal of dimensions

Figure 23: Meaningful dimensions visualization for $C = 20$, end = 50000 (different random seeds). The KL divergences of the target dimension (0-4 dimension) increase one by one during training (a). The KL divergences in different dimensions are different after training (b). As the image traversal results (c) show, the meaningful dimensions are learned in 0-4 dims. So are (d), (e), and (f), which are results of a run with a different random seed.

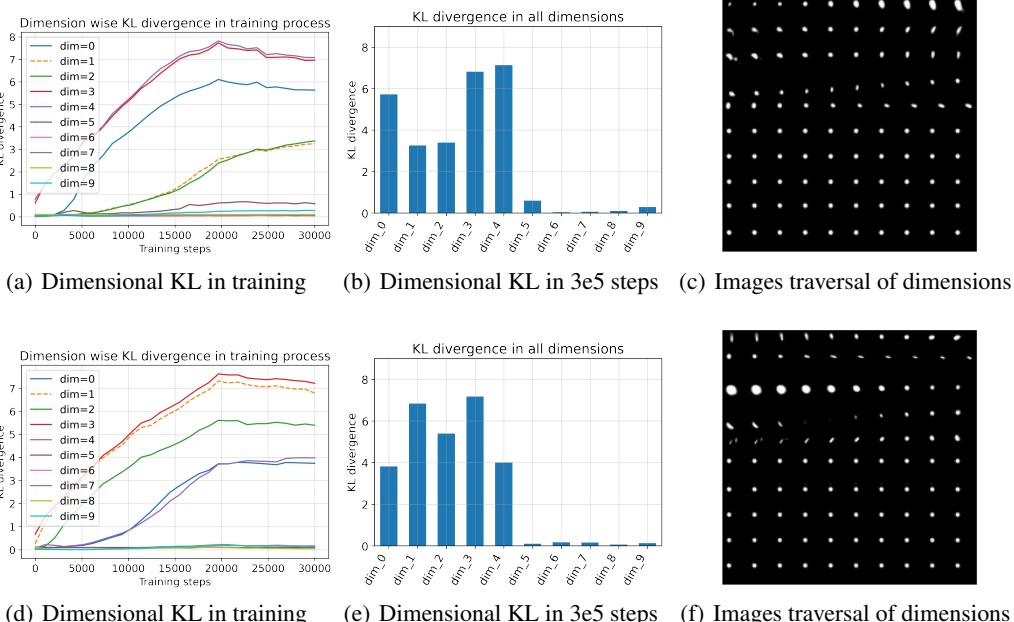

(a) Dimensional KL in training     (b) Dimensional KL in 3e5 steps     (c) Images traversal of dimensions

(d) Dimensional KL in training     (e) Dimensional KL in 3e5 steps     (f) Images traversal of dimensions

Figure 24: Meaningful dimensions visualization for $C = 30$, end = 30000 (different random seeds). The KL divergences of the target dimension (0-4 dimension) increase one by one during training (a). The KL divergences in different dimensions are different after training (b). As the image traversal results (c) show, the meaningful dimensions are learned in 0-4 dims. So are (d), (e), and (f), which are results of a run with a different random seed.

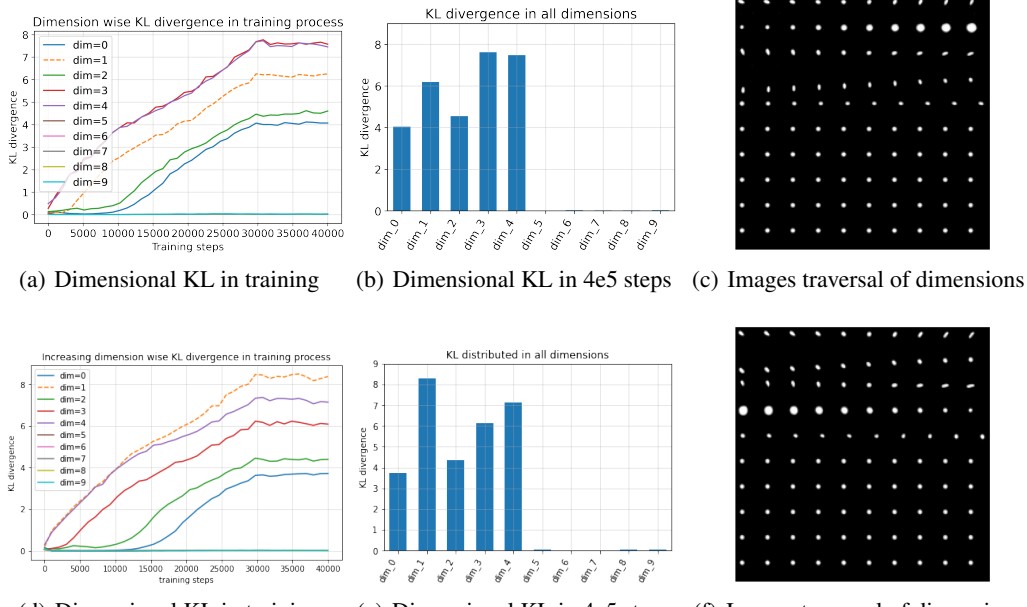

(a) Dimensional KL in training  (b) Dimensional KL in 4e5 steps  (c) Images traversal of dimensions

(d) Dimensional KL in training  (e) Dimensional KL in 4e5 steps  (f) Images traversal of dimensions

Figure 25: Meaningful dimensions visualization for $C = 30$, end $= 40000$ (different random seeds). The KL divergences of the target dimension (0-4 dimension) increase one by one during training (a). The KL divergences in different dimensions are different after training (b). As the image traversal results (c) show, the meaningful dimensions are learned in 0-4 dims. So are (d), (e), and (f), which are results of a run with a different random seed.

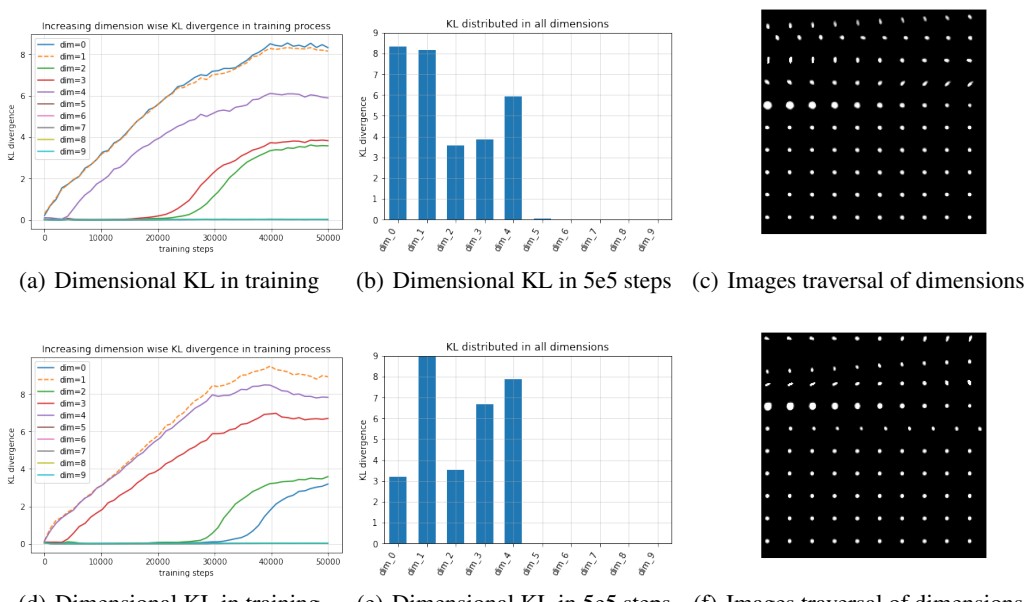

(a) Dimensional KL in training  (b) Dimensional KL in 5e5 steps  (c) Images traversal of dimensions

(d) Dimensional KL in training  (e) Dimensional KL in 5e5 steps  (f) Images traversal of dimensions

Figure 26: Meaningful dimensions visualization for $C = 30$, end $= 50000$ (different random seeds). The KL divergences of target dimensions (0-4 dimension) increase one by one during training (a). The KL divergences in different dimensions are different after training (b). As the image traversal results (c) show, the meaningful dimensions are learned in 0-4 dims. So are (d), (e), and (f), which are results of a run with a different random seed.

## L   MORE REPRESENTATION SPACE VISUALIZATIONS

To illustrate that the Isomorphism Loss suppresses the representation space collapse in VAE-based methods is not an exception, we provide more visualizations of the representation space visualization of *Groupified* VAEs. The Original and *Groupified* $\beta$-VAEs, AnnealVAEs, FactorVAEs, and $\beta$-TCVAEs are shown in Figure 27, Figure 28, Figure 29, and Figure 30, respectively, which implies that the representation space of *Groupified* VAEs are organized better than the original ones.

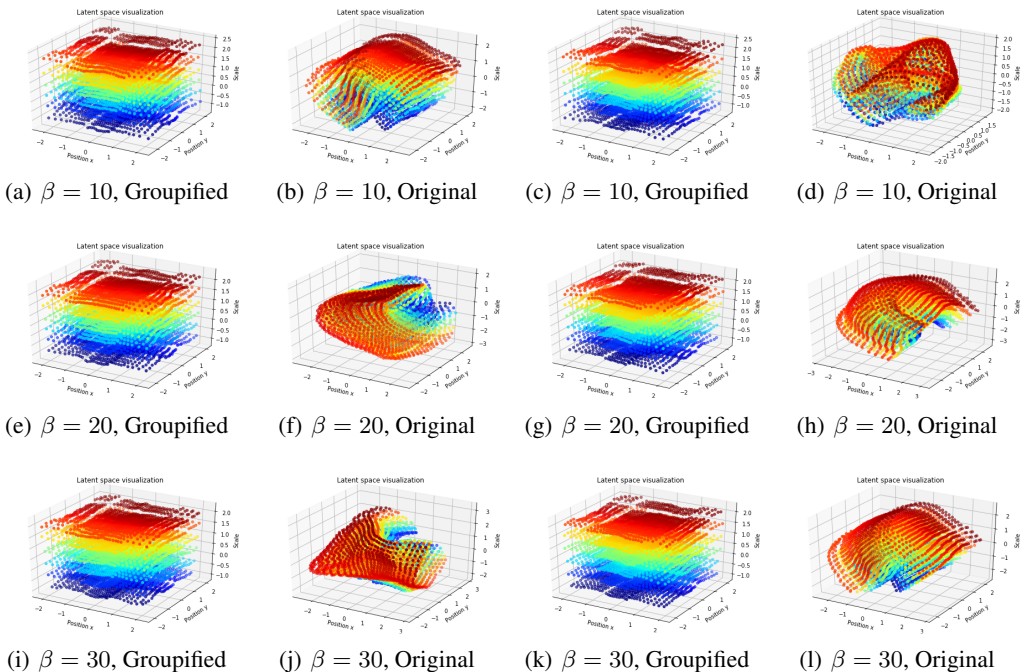

(a) $\beta = 10$, Groupified   (b) $\beta = 10$, Original   (c) $\beta = 10$, Groupified   (d) $\beta = 10$, Original

(e) $\beta = 20$, Groupified   (f) $\beta = 20$, Original   (g) $\beta = 20$, Groupified   (h) $\beta = 20$, Original

(i) $\beta = 30$, Groupified   (j) $\beta = 30$, Original   (k) $\beta = 30$, Groupified   (l) $\beta = 30$, Original

Figure 27:   The representation space span by the *Groupified* and Original $\beta$-VAE. We train the models with the same hyperparameter but different random seeds for different runs. The 3D location of each point is the disentangled representation of the corresponding image. Moreover, an ideal result is that all the points form a cube, and color variation is continuous. Higher hyper-parameter $\beta$ results in the collapse of representation space. The collapse is suppressed by the Isomorphism Loss, which leads to better disentanglement.

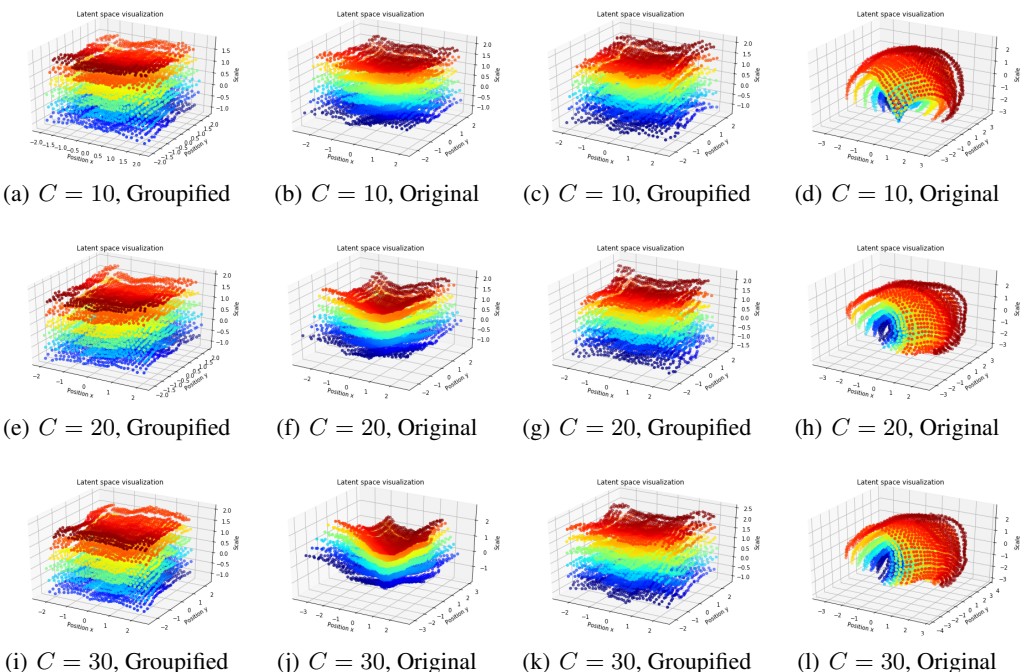

Figure 28: The representation space span by the *Groupified* and Original AnnealVAE. We train the models with the same hyperparameter but different random seeds for different runs. The 3D location of each point is the disentangled representation of the corresponding image. Moreover, an ideal result is that all the points form a cube, and color variation is continuous. Higher hyper-parameter $C$ results in the collapse of representation space. The collapse is suppressed by the Isomorphism Loss, which leads to better disentanglement.

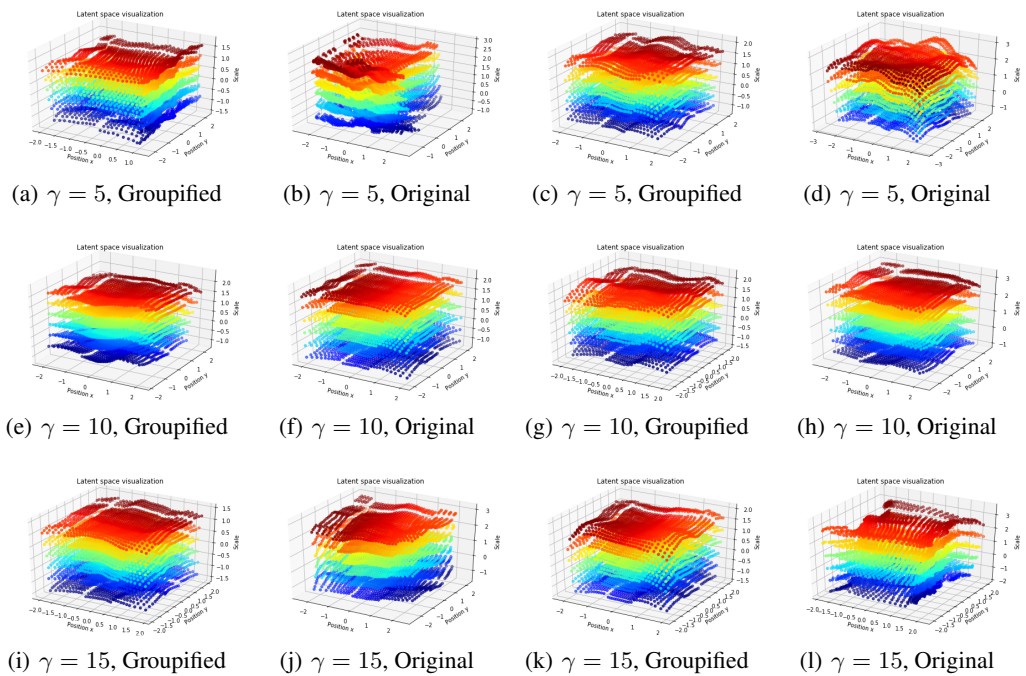

Figure 29: The representation space span by the *Groupified* and Original FactorVAE. We train the models with the same hyperparameter but different random seeds for different runs. The 3D location of each point is the disentangled representation of the corresponding image. Moreover, an ideal result is that all the points form a cube, and color variation is continuous. The collapse is suppressed by the Isomorphism Loss, which leads to better disentanglement.

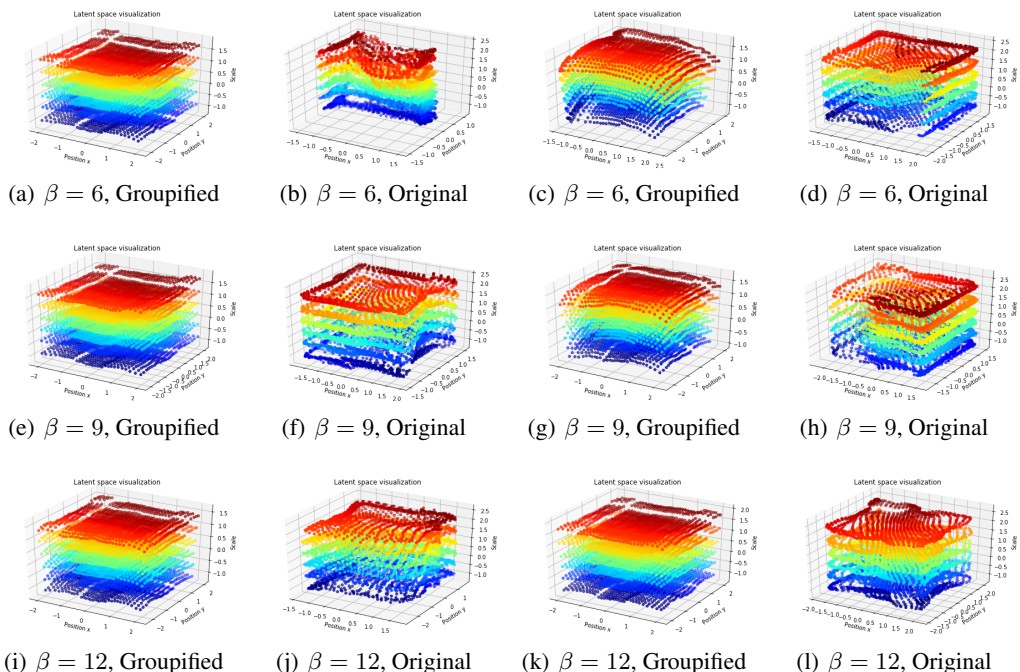

(a) $\beta = 6$, Groupified    (b) $\beta = 6$, Original    (c) $\beta = 6$, Groupified    (d) $\beta = 6$, Original

(e) $\beta = 9$, Groupified    (f) $\beta = 9$, Original    (g) $\beta = 9$, Groupified    (h) $\beta = 9$, Original

(i) $\beta = 12$, Groupified    (j) $\beta = 12$, Original    (k) $\beta = 12$, Groupified    (l) $\beta = 12$, Original

Figure 30: The representation space span by the *Groupified* and Original $\beta$-TCVAE. We train the models with the same hyperparameter but different random seeds for different runs. The 3D location of each point is the disentangled representation of the corresponding image. Moreover, an ideal result is that all the points form a cube, and color variation is continuous. Higher hyper-parameter $C$ results in the collapse of representation space. The collapse is suppressed by the Isomorphism Loss, which leads to better disentanglement.

