# OpenReview forum: "Towards Building A Group-based Unsupervised Representation Disentanglement Framework"
_ICLR.cc/2022/Conference — ICLR 2022 Poster_

### Official Review · Reviewer_PA1y · 2021-10-26

**Correctness:** 4
**Technical Novelty And Significance:** 3
**Empirical Novelty And Significance:** 3
**Recommendation:** 6
**Confidence:** 4

**Main Review:**

Pros:
1. Proposed a new theoretical framework for achieving unsupervised representation disentanglement based on $n$-th dihedral group.
2. Provided theoretical analysis for the proposed framework via model and group structure
3. Conducted extensive experiments to verify the proposed framework

Cons:
1. Missing some latest works on disentanglement. The related work section is a bit weak since it only discussed the VAE models before the year 2019. In fact, there are a lot of related works about disentangled representation learning after 2019, such as TamingVAE, ControlVAE, and other works [1,2,3,4].
2. Please also compare the proposed framework with the baselines in recent work as mentioned above.

3. In Table 1, for dSprites dataset, the MIG score is much lower than the result illustrated in the literature $\beta$-TCVAE and ControlVAE. Please clarify and explain it.

[1] Srivastava, Akash, Yamini Bansal, Yukun Ding, Cole Hurwitz, Kai Xu, Bernhard Egger, Prasanna Sattigeri, Josh Tenenbaum, David D. Cox, and Dan Gutfreund. "Improving the Reconstruction of Disentangled Representation Learners via Multi-Stage Modelling." arXiv preprint arXiv:2010.13187 (2020).

[2] Shao, H., Yao, S., Sun, D., Zhang, A., Liu, S., Liu, D., ... & Abdelzaher, T. (2020, November). ControlVAE: Controllable variational autoencoder. In International Conference on Machine Learning (pp. 8655-8664). PMLR.

[3] Kim, M., Wang, Y., Sahu, P., & Pavlovic, V. (2019). Bayes-factor-vae: Hierarchical bayesian deep auto-encoder models for factor disentanglement. In Proceedings of the IEEE/CVF International Conference on Computer Vision (pp. 2979-2987).

[4] Lezama, J. (2018, September). Overcoming the disentanglement vs reconstruction trade-off via jacobian supervision. In International Conference on Learning Representations.

**Summary Of The Paper:**

This paper proposed a new theoretical framework for achieving unsupervised representation disentanglement based on $n$-th dihedral group. Further, it proved three sufficient conditions on the model, group structure, and data respectively. Evaluations on multiple benchmark datasets demonstrate the proposed framework achieves better mean performance with smaller variances compared to some VAE models.

**Summary Of The Review:**

This paper proposed a new theoretical framework for achieving unsupervised representation disentanglement. My concern is that this work only compares it with the baselines before 2018, so it is hard to say whether it outperforms the recent baselines.

---

> ### Author Response · Authors · 2021-11-14
> **Response to Reviewer PA1y**
>
> Thanks for reviewing our paper and appreciating our idea. Your concerns are addressed   below. We are looking forward to further discussion.
>
> 1. the MIG score is much lower than the result illustrated in the literature $\beta$-TCVAE and ControlVAE.
>
> Our experimental settings are different from $\beta$-TCVAE and ControlVAE. We follow [a] to conduct experiments and verify our method taking into account the impact of hyperparameters and random seeds. This evaluation setting is well accepted in the literature (e.g., Fig. 5 in [b], Fig.3 in [c]). Our method is effective across different hyperparameters and random seeds and better than other methods, including $\beta$-TCVAE and ControlVAE.
>
> We found out that the value of MIG of $\beta$-TCVAE is incorrect. The values here are 0.15±0.067 (original) and 0.21±0.093 (groupified), we have updated the values in the rebuttal version (Highlighted with blue, please check).
>
> 2. Missing some latest works on disentanglement.
>
> We thank the reviewer for providing these related works, and we have followed your suggestion to discuss these works in our rebuttal version and highlighted with blue. We will conduct the experiments on ControlVAE [d] (2020), since 1) Taming VAE(2018), [e] (2018) and [f] (2019) are works before 2020 and 2) MS VAE [h] (2020) [1] is a preprint and did not release their code. Due to the time limit, here we only present the results on dsprites (under the following hyper-parameter settings and random seeds). For results on other datasets, we  will add them in the final version as we finalize these experiments. As the following table shows,   our model still works on ControlVAE.
>
> C_max 15, 25, 35
>
> Random seeds: 0-9
>
> groupified: True, False
>
> Total: 3x10x2 = 60 runs
>
> Other parameters follow the best setting in [d]
>
> | Model | DCI | BetaVAE | MIG | FactorVAE |
> | :-------:| -----: | :----------: | :----: | :-------------: |
> | Original | 0.30 ±0.114    |       0.83±0.084    |    0.15±**0.062**      |     0.62±0.086  |
> | Groupified |  **0.46** ± **0.093**      |    **0.92**±**0.061**    |    **0.26**±0.096     |     **0.73**±**0.072** |
>
> mean±std
>
>
>
>
> [a] Locatello, Francesco, et al. "Challenging common assumptions in the unsupervised learning of disentangled representations." international conference on machine learning. PMLR, 2019.
>
> [b] Rhodes, Travers, and Daniel D. Lee. "Local Disentanglement in Variational Auto-Encoders Using Jacobian $ L_1 $ Regularization." NeurIPS, 2021.
>
> [c] Hälvä, Hermanni, et al. "Disentangling Identifiable Features from Noisy Data with Structured Nonlinear ICA." NeurIPS, 2021.
>
> [d] Shao, Huajie, et al. "ControlVAE: Controllable variational autoencoder." ICML, 2020.
>
> [e] Lezama, J. Overcoming the disentanglement vs reconstruction trade-off via jacobian supervision. ICLR 2018.
>
> [f] Kim, M., Wang, Y., Sahu, P., & Pavlovic, V. Bayes-factor-vae: Hierarchical bayesian deep auto-encoder models for factor disentanglement. ICCV 2019.
>
> [h] Srivastava, Akash, Yamini Bansal, Yukun Ding, Cole Hurwitz, Kai Xu, Bernhard Egger, Prasanna Sattigeri, Josh Tenenbaum, David D. Cox, and Dan Gutfreund. Improving the Reconstruction of Disentangled Representation Learners via Multi-Stage Modelling. Preprint 2020.

---

> > ### Comment · Reviewer_PA1y · 2021-11-22
> > **About MIG**
> >
> > Thanks a lot for adding the latest work and adding new experiments. I still have two questions about the MIG score of ControlVAE and FactorVAE below.
> >
> > [1] I am wondering if you use the MIG of TensorFlow version from [a] to measure ControlVAE and FactorVAE?
> > [2] Why do you set the C_max of ControlVAE to 15, 25, and 35?

---

> > > ### Author Response · Authors · 2021-11-23
> > > **Response to "About MIG"**
> > >
> > > Thanks for your response, for MIG, yes, we use the TensorFlow version from [a] (https://github.com/google-research/disentanglement_lib) and for ControlVAE, we use the model code provided by https://github.com/shj1987/ControlVAE-ICML2020 (official implementation). We follow the default setting as C_max = 25 (in Disentangling/main.py).  We follow [a] to set the hyperparameter interval to 10.
> > >
> > > We can add more results under different settings if you think it is necessary.  For example,  C_max = XX, as reported in the ControlVAE paper.

---

> ### Author Response · Authors · 2021-11-22
> **Looking forward to hearing from you**
>
> Dear Reviewer PA1y,
>
> We want to send you a friendly reminder for the discussion, since the second stage of discussion will be soon concluded.
>
> We thank you again for your valuable comments, and we are happy to extend our response if you have any other concerns left.
>
> Thanks.

---

### Official Review · Reviewer_M8Aa · 2021-10-28

**Correctness:** 2
**Technical Novelty And Significance:** 2
**Empirical Novelty And Significance:** 2
**Recommendation:** 3
**Confidence:** 2

**Main Review:**

Strengths:
- The mathematical part of the paper and experiments appear sound.
- The isomorphism loss is to my knowledge novel, and seems to bring performance gain across datasets and models.

Weaknesses:
-  The paper is at best ambiguous and at worst misleading concerning whether or not it claims to solve the problem of unsupervised representation disentanglement
    - *"we offer an option [...] for the inductive bias that existing VAE-based models lack"*
    - *"Finally, we provide a learning model based on the existing VAE-based methods in an effort to fulfill the three conditions."*
    - *"we are the first to provide a theoretical framework to make the formal group-based mathematical definition of disentanglement practically applicable to unsupervised representation disentanglement."*
   - *"making the learned representation conform to the group- based definition without relying on the environment"*

   But the paper never theoretically shows that the data constraint is actually satisfied, and instead relies on a necessary condition of the data constraint. This greatly undermines the theoretical contribution of the paper.
- I find the mathematical writing to be overall quite obfuscated, which complicates assessment of the theoretical contribution.

As actionnable feedback:
- I would recommend to be more open about which theoretical guarantees the proposed framework actually provides.
- Could you please give intuitive explanations of the role of the isomorphism loss, its novelty and comparison to related work, and why it brings this performance gain?

**Summary Of The Paper:**

The paper proposes a theoretical framework for unsupervised representation disentanglement (in the sense of Higgins et al.) based on three constraints (group structure, data and model). It further describes a learning method for satisfying these constraints, and experimentally shows some performance gain on traditional disentanglement metrics.

My intuitive understanding of the constraints is a follows:
- the model constraint is satisfied by construction of the agent group, whose structure is transferred from the latent space by the model.
- the group structure constraint ensures that this transfer is an isomorphism, and thus that the agent group is actually a group.
- the data constraint ensures that the agent group actually matches the ground truth generative factors.

Please clarify if one of my statements is incorrect.

**Summary Of The Review:**

I find this paper to be overselling the importance of its theoretical contribution, and will therefore recommend rejection. I am looking forward to discussion with the authors, and will reconsider my rating if the authors respond to the two points listed as actionnable feedback in my main review. In particular I'd like to hear more about the isomorphism loss.

---

> ### Author Response · Authors · 2021-11-14
> **Response to Reviewer M8Aa (Part 2)**
>
> 2. Clarification of the Isomorphism Loss:
>
> - The role of Isomorphism Loss
>
> Intuitively speaking, we constrain the isomorphism relation to make the VAEs satisfy the symmetry requirement of the definition as the additional constraint for VAEs, and the symmetry requirement (commutative and cyclic) come from the nature of factors.
>
> - Its novelty and comparison to related work
>
> To the best of our knowledge, we are the first to construct a group in this way (elements are image-to-image mapping and determined by the parameters of VAEs) and the first to derive a loss from the group-based definition in an unsupervised setting.
> The way we construct group $\Phi$: As shown in Fig. 1 (a), the group element $\varphi$ (which is an image-to-image mapping) is the VAE-based mapping that encodes an image to latent space and conducts latent space operation then decodes back to the image space. The group operation is a mapping composition.
>
> The derivation of Isomorphism Loss: As proved in Theorems 1, 2, 3, the sufficient condition of group structure constraint is that $\Phi$ is isomorphic to $G$, and this condition is satisfied when Isomorphism Loss (Abel loss and Order loss) is optimized, which serves as an additional constraint on the original VAE-based systems to help optimize disentanglement.
>
> From the perspective of isomorphism loss, the most related work is [c] to the best of our knowledge. They proposed using the commutative Lie group to represent the latent space. They also require that the group used is commutative. However, we  have the following fundamental differences:
>
> 1) Their group does not come from the group-based definition. Their target is to capture the variation in latent space by a Lie group rather than introducing the group-based definition into the unsupervised setting.
> 2) Their group’s elements are matrix, but ours are image-to-image mappings. Their group operation is matrix multiplication, but ours is function composition.
>
> - why it brings this performance gain
>
> The intuition is that it provides a cycle consistency constraint under these different transformations (group action) of latent space to avoid the collapse of the manifold in the latent space (and reduce the solution space), as  Fig. 3 shows.
>
> In other words, the symmetry property introduced by the isomorphism loss eliminates the distortion of the manifold in the latent space. Specifically, the Able loss constrains the latent space to stay unchanged under the exchange of two different $\varphi_i$, The Order Loss constrains the latent space is cyclic under n compositions of $\varphi_i$. The commutative and cyclic properties reduce the solution space of the VAEs. These properties come from the nature of disentangled factors.
>
>
> [a] Francesco Locatello, Stefan Bauer, Mario Lucic, Gunnar Raetsch, Sylvain Gelly, Bernhard Scholkopf, ¨ and Olivier Bachem. Challenging common assumptions in the unsupervised learning of disentangled representations. In ICML, 2019.
>
> [b] Learning group structure and disentangled representations of dynamical environments. NeurIPS, 2020.
>
> [c] Xinqi Zhu, Chang Xu, and Dacheng Tao. Commutative lie group vae for disentanglement learning. ICML, 2021.
>
> We would appreciate it if you would let us know whether the above clarifies the novelty of our work and changes your assessment of the significance of our work.

---

> > ### Comment · Reviewer_M8Aa · 2021-11-22
> > **Discussion about the isomorphism loss - Final**
> >
> > After reading the author response and re-reading the submission, I still have trouble understanding the intuition of why the isomorphism loss brings this performance gain. Here is a detailed look at the authors explanations
> > - *The intuition is that it provides a cycle consistency constraint under these different transformations (group action) of latent space* -> The mention of "cycle consistency" is quite vague
> > - *to avoid the collapse of the manifold in the latent space (and reduce the solution space), as Fig. 3 shows.* -> What does the collapse of the manifold mean?
> > - *In other words, the symmetry property introduced by the isomorphism loss eliminates the distortion of the manifold in the latent space.* ->  Again, what does the collapse of the manifold mean?
> > - *Specifically, the Able loss constrains the latent space to stay unchanged under the exchange of two different
> > φ_i, The Order Loss constrains the latent space is cyclic under n compositions of φ_i* -> this is mainly paraphrasing the mathematical definitions, which I understood
> > - *The commutative and cyclic properties reduce the solution space of the VAEs.* -> To me this also looks like paraphrasing
> > - *These properties come from the nature of disentangled factors.* -> This does not bring me much information
> >
> > **Final opinion**
> > To me, getting an intuitive idea of why the isomorphism loss brings this performance gain is crucial, as the isomorphism loss is the only model improvement of the submission.  With my current level of understanding, I don't feel confident in recommending acceptance, and I will stick to my reject recommendation. Considering (i) that the other reviewers have a more positive opinion (ii) that the authors have been responsive in the discussion phase, it is certainly possible that I didn't understand some central aspects of the submission, and I have changed my confidence level to 2.

---

> > > ### Author Response · Authors · 2021-11-22
> > > **Response to Reviewer M8Aa**
> > >
> > > Thanks for your response and re-reading. The intuition provided above is derived by inferring from the experimental results, which may make you feel that it is not so strong. Here, we would like to provide another  “top-level” intuitive understanding of why isomorphism loss works. The group theory based definition is motivated by the symmetry property of the data. Based on this, we introduce the isomorphism loss to reorganize the latent space to reflect this symmetry property. Starting from the group-based theoretical definition of disentanglement, we get a quite practical isomorphism loss derived by mathematics proof step by step.  In our last response, we provide some detailed analysis on the experimental results, trying to bring a “bottom-level” intuitive understanding. Maybe it is better to have an intuitive understanding between the top-level and the bottom-level, which we can not provide currently. This may further indicate the value of our theoretical framework, it is hard to directly propose such an intuitive idea only from the perspective of practice itself.

---

> > > ### Author Response · Authors · 2021-11-22
> > > **Further clarification of Isomorphism Loss**
> > >
> > > Although the reviewer has given the final option, considering that our clarification is still not clear enough for the reviewer to fully understand our intuition, we would like to give some further clarification on the intuition. We want to highlight Fig.5 in our rebuttal version.
> > >
> > > **What does the collapse of the manifold mean, and why cycle consistency makes sense?**
> > >
> > > Before we answer the question, we first present the detailed process of plotting Fig. 5.
> > >
> > > In order to have a global view of how the isomorphism loss leads to performance gains, we use the following way to visualize the learned latent space. For a given trained VAE-based model, 1) from the dataset, we select a set of images by traversing all the GT values of three different factors (x-position, y-position, scale), and images are labeled with continuously varying colors.  2) feed the images into the VAE-based model and get the corresponding representations. 3) for each representation we plot a point in 3D space,  whose 3D coordinates are the corresponding representation of x-position, y-position, scale, and whose color is the labeled color of the corresponding image. Thus we plot a manifold for the given VAE-based model as shown in Fig. 5.
> > >
> > >
> > > An ideally disentangled case should be a cube with continuous color variation.
> > > We find the manifold of those original VAEs with poor performance has the following phenomenon:  the manifold is not totally disorganized, but only has clear shape distortion.  (as shown in Fig. 5, (e,f,g,h)). If the distortion can be tackled, we can achieve better disentanglement.
> > >
> > > However, for the manifolds of Groupified VAEs, the distortion is suppressed. Considering that the physical meaning of isomorphism loss is that the latent space after different transformations should be well aligned, which is a kind of cycle consistency constraint (under different transformations, the results should be the same). The distorted spaces are usually asymmetry, resulting in a large value of the loss. By minimizing the loss, this phenomenon is well suppressed.  This observation is quite common, and we provide a lot of results in appendix L.
> > >
> > > Therefore, we derived a **conclusion**: the loss itself is a kind of cycle consistency to avoid the collapse of the manifold (latent space), which leads to factors entanglement in VAE models, as shown in Fig. 5 in the rebuttal version.
> > >
> > > Since this observation indeed helps us understand these gains, we believe this intuition can also help reviewers and other readers to intuitively understand why the isomorphism loss brings this performance gain.

---

> ### Author Response · Authors · 2021-11-14
> **Response to Reviewer M8Aa (Part 1)**
>
> Thanks for providing constructive comments. Your concerns are addressed below. We are looking forward to further discussion.
>
>
> We do not intend to mislead the reviewers, and thanks for pointing out the ambiguity problem in our writing. We regret that it caused your misunderstanding.  We have the word "towards" in our title and other words like "in an effort to," "practically applicable," etc., in the Abstract & Introduction to indicate that we have not completely solved this problem.  We have made the point clearer in the abstract & introduction in the rebuttal version (highlighted with blue, please check) by stating we derive two sufficient conditions for the model and group constraints and a necessary condition for the data constraint. We believe that our rebuttal version can address the ambiguous problem well.
>
> In the following, we provide the explanations and clarifications for your actionable feedback. We believe that our explanation can help you better understand our contribution.
>
> 1. Which theoretical guarantees the proposed framework actually provides.
>
> We are the first to theoretically explore one promising approach (group-based definition) for unsupervised disentanglement, which we believe is very valuable to this community, as also commented by reviewer AKvy.
>
> Specifically, our **theoretical guarantee** is:  our theoretical framework narrows down the landscape of the problem by deriving three sufficient conditions of the group-based definition. Two of them (model, group constraint) can be sufficiently satisfied, and we derive a necessary condition for the third one (data constraint). Please note that, as pointed out in [b], it is not straightforward to reconcile the probabilistic inference methods with the group-based definition framework. Therefore, it is challenging to derive constraints from the group-based definition directly, but with our proposed framework, two of the constraints can be sufficiently satisfied, and the last/third one has a necessary condition. We believe (also pointed out by Reviewer AKvy) that it will have a good discussion potential for the community since our theoretical framework has the potential to derive sufficient conditions for the group-based definition in an unsupervised setting.
>
> In addition, our framework not only demonstrates that the previous VAE-based methods only consider the data constraint (which is in line with the “impossible” conclusion in [a]), but also, it offers the answer of “what inductive bias these VAEs lack” from the perspective of group-based definition: i.e., model and group constraints and data constraint (existing VAEs only satisfy a necessary condition for data constraint).

---

> > ### Comment · Reviewer_M8Aa · 2021-11-18
> > **Discussion about the theoretical claims**
> >
> > I thank the authors for clarifying most of their theoretical claims, and I see that this is has been taken into account in the new version of the submission. However, I still have a concern about the claim that *"[the authors] offer an option for the inductive bias that existing VAE-based models lack"*, which hasn't been modified. Indeed, I would argue that the main problem of existing VAE models (and the main motivation for searching good inductive biases) is that they also only satisfy a necessary condition for the data constraint. In order to support the claim that the submission *"offers an option for the inductive bias these VAEs lack”*, it would be necessary to also show experimentally or theoretically that the data constraint is solved. I would advise the authors to also rephrase this claim

---

> > > ### Author Response · Authors · 2021-11-18
> > > **Response to "Discussion about the theoretical claims"**
> > >
> > > Thanks for your response and pointing this out, we agree with you that the claim still causes ambiguity here. We rephrased the claim to make the point clearer: “provides options from the perspective of group-based definition for the additional inductive bias in existing VAE-based methods.” We have also modified the corresponding claim (also other similar claims) in our rebuttal version (highlighted with blue), please check.
> > >
> > > Specifically, the modifications include (highlighted with blue, please check):
> > > - We changed the claim that "However, it has been proved that ..." into "However, these models lack theoretical ..." in Abstract.
> > > - We changed the claim that "With the first two of the conditions satisfied ..." into "With the first two of the conditions satisfied ...as an additional inductive bias for ..." in Abstract.
> > > - We changed the claim (also other similar claims) that “... for the inductive bias these VAEs lack” into “ … for the additional inductive bias in existing VAE-based methods” in Introduction and also delete “existing VAE-based models only consider...” in Sec.4.

---

> > > > ### Comment · Reviewer_M8Aa · 2021-11-22
> > > > **Discussion about the theoretical claims - Final**
> > > >
> > > > I thank the authors for these clarifications. I appreciate part of the modifications but I would still join Reviewer YTd1 in raising concerns about the multiple references to an **inductive bias**. The following paragraph Reviewer YTd1 reflects my opinion better than I could write myself
> > > >
> > > > *(2) After more detailed thinking, I want to make my point clearer. The key to solving the unidentifiability in [b] is the condition (ii) in your proposed Theorem 1, which links the group ϕ acting on data to the ground-truth generators. However, this data constraint modeling in this paper is solely modeled by the total correlation minimization. The inductive biases introduced in this paper are the group structure constraint and model constraint, which I believe are valuable by themselves, but not related to the inductive biases meant by [b] as they cannot contribute to the unidentifiable problem. I believe it is totally fine the model in this paper cannot solve the unidentifiability problem which is as you said very challenging, but the motivation and discussion in this paper try to convey that this paper proposes ‘inductive biases’ that are wanted by [b] to solve the unidentifiability problem, but the actual ‘biases’ (group structure constraint and model constraint) are not related to this problem, while the only key point ‘data constraint’ that can solve this unidentifiability problem is modeled in the same way as usual methods by the total correlation minimization. Therefore I believe the authors should emphasize the benefits of the group structure constraint and model constraint themselves, rather than trying to describe them as the lacked inductive biases mentioned in [b].*

---

> > > > > ### Author Response · Authors · 2021-11-22
> > > > > **Response to Reviewer M8Aa**
> > > > >
> > > > > Thanks for your response. In our rebuttal version, by “inductive bias”, we mean we provide additional constraints from the perspective of group-based definition.  These constraints are valuable itself, which was agreed by Reviewer YTd1 and AKvy. We regret that the word “inductive bias” used here still caused your concern. We would like to modify the “inductive bias” into “additional constraints” in our final paper if it is necessary to address this ambiguity.  Our main contribution is introducing the group-base definition into the unsupervised setting, it is valuable itself and has good discussion potential for the community (which was agreed by Reviewer AKvy). although we made an ambiguous statement on the “inductive bias”. (This has already been modified and can be further refined, and it will not hurt our main contribution).

---

> > > ### Author Response · Authors · 2021-11-22
> > > **Looking forward to hearing from you**
> > >
> > > Dear Reviewer M8Aa,
> > >
> > > We want to send you a friendly reminder for the discussion, since the second stage of discussion will be soon concluded.
> > >
> > > We thank you again for your valuable comments, and we would appreciate it if you could reconsider the evaluation of our work based on our response. We are happy to extend our response if you have any other concerns left.
> > >
> > > Thanks.

---

### Official Review · Reviewer_YTd1 · 2021-10-29

**Correctness:** 3
**Technical Novelty And Significance:** 3
**Empirical Novelty And Significance:** 2
**Recommendation:** 6
**Confidence:** 4

**Main Review:**

Strengths:
1. The authors provide a theoretical framework on the learning of group-based disentangled representation, which is a step forward to the more general guaranteed disentanglement learning.
2. The experimental results validate the disentanglement effect of the 'Groupify' operation on multiple base models and synthetic datasets.
3. The proposed technique is compatible with all the information-theory-based baseline models.

Weaknesses:
1. The introduction of $\Phi$ group is unnecessarily complex. Do the authors mean the $\Phi$ group is a direct product of $m$ cyclic groups of order n (or even simpler, $m$-dim Torus)? Is each cyclic subgroup represented by $(sin((2\pi z))/n), cos((2\pi z)/n))$, with z being the output of the VAE encoder? The generator of each subgroup is the action of rotating each circle by step $2\pi/n$? I see no necessity of incorporating more concepts from Group Theory like 'congruence class modulo n', 'n-th root unity group', 'additive group of integers modulon'. To me they are all the same in the context of this paper.
2. The mention of 'n-th dihedral group' is not informative, or even misleading. Dihedral groups are generated by only two subgroups, rotation and flip, which is not very related to the group used in this method (a product of m cyclic groups of the same order n).
3. The Abel loss and Order loss can only encourage the representation to maintain a group structure of m-dim torus. However, this does not solve the unidentifiability problem that motivates this method (the impossibility of disentanglement mentioned in the abstract, introduction section, and the paragraph before Sec. 4.1) because the ground-truth correspondence between the group action on the world space and its action effect on the image space is still unknown. The real problem here is not how to learn a group that is isomorphic to an m-dim cyclic group, but how to make sure the learned cyclic subgroups correspond to the interpretable variations shown in the image space. This paper proposes a method to learn a cyclic-group representation, which is valuable by itself in some way, but it still falls into the unidentifiability problem proposed in Locatello et al. 2019b.
4. I don't think $\phi\phi^{-1}=e$ can replace $\phi\phi^{n-1}=e$ in the Order loss. $\phi\phi^{n-1}=e$ ensures the cyclic structure of order n, but $\phi\phi^{-1}=e$ holds in any group with arbitrary order. The equation $\phi^{-1}=\phi^{n-1}$ holds when the cyclic structure has **already been learned**, but $\phi\phi^{-1}=e$ alone cannot help the learning of a cyclic structure.
5. It looks like the proposed VAE model is a special case of the Commutative Lie Group VAE proposed in https://arxiv.org/abs/2106.03375 with the learned group representation being replaced by a pre-defined direct product of cyclic groups. They should be more clearly discussed and compared in the main paper.
6. Currently only the synthetic datasets provided in the open-sourced disentanglement library (https://github.com/google-research/disentanglement_lib) are used in the experiments. More results on some real-world datasets like CelebA should be provided to validate the model's generalization ability.
7. The paper only shows the effectiveness of the method as an incremental technique to improve the existing disentanglement models, but does not show its stand-alone performance. If the proposed constraints are effective enough, they should outperform the existing SOTA models by being directly added to a vanilla VAE (as all the baseline models do).

**Summary Of The Paper:**

This paper provides a theoretical framework on the learning of group-based disentanglement representations. It proposes a method to learn a cyclic group representation with the Abel loss and Order loss based on VAEs. Experiments validate the effectiveness of the proposed method on improving existing disentanglement VAE models.

**Summary Of The Review:**

I believe the theoretical contribution of this paper is valuable to the group-based disentanglement learning (or general disentanglement learning) community, but I still have concerns listed in the main review section. I currently rate this paper as weak reject, but I will be glad to improve my score if my concerns are properly solved.

======

After rebuttal:
Thanks for the authors' response to my comments. I think a large part of my concerns has been addressed. I will increase the score to 6 weak accept.

---

> ### Author Response · Authors · 2021-11-14
> **Response to Reviewer YTd1 (Part 2)**
>
> 5. Commutative Lie Group VAE should be more clearly discussed and compared in the main paper.
>
> We thank the reviewer for providing this related work, we will add the following discussions to the final version. First of all, we emphasize the building of a theoretical framework to use group definition towards solving unsupervised disentanglement. However, [a] 's contribution is more about a constraint in the latent space  ([a] proposed using the commutative Lie group to represent the latent space), which does not come from group-based definition. Specifically, the definitions of the group in our paper are different from [a]. We use the autoencoder's encoding action and decoding process to define the elements of the group, as Fig. 1 shows. In addition, our framework is compatible with their proposed method. The Lie group representation can be applied to extend our framework into a lie group version.
>
> In addition, the motivation of “commutative” of ours and [a] are totally different. In [a], the “commutative” is motivated by the decomposition requirement of the Lie group (one-parameter subgroup decomposition in Sec. 4.2 in [a]). Each subgroup characterizes one kind of variation. Ours “commutative” is motivated by the symmetry requirement of the group-based definition, which comes from the nature of factors.
>
> 6. More results on some real-world datasets.
>
> Thanks for the constructive suggestion. We have added the qualitative experimental results on CelebA in the rebuttal version (as done in [a, g]) and highlighted with blue, please check. We nevertheless want to emphasize here that we have conducted extensive experiments (as commented by other reviewers AKvy & PA1y) to demonstrate that the improvements of our theoretical framework are cross-data sets, cross-models, and cross hyper-parameters, our model’s effectiveness is already validated with strong evidence.
>
> 7. The paper does not show its stand-alone performance.
>
> Thank you for raising the question about the stand alone performance of our model. This is an interesting concern. However, we can not fully agree with the comment that our method should outperform the existing SOTA models by being directly added to a vanilla VAE. The reasons are twofold:
>
> - According to our  theoretical framework, in order to achieve disentanglement, our method requires the minimization of TC,as stated in the second paragraph of section 4, “the minimization of total correlation is… to fulfill the data constraint to some extent for the unsupervised setting.”  Technically, please note that since the baseline models propose a substituted objective for Total correlation minimization, they directly add their constraint to a vanilla VAE. However, we propose complementary techniques to those models rather than a substituted one, thus using our new constraints alone does not make sense.
> - Many other techniques are also based on the existing disentanglement models or regularization. For example,  Lie group VAE [ a] without hessian penalty [c] can not outperform the existing SOTA models (Tab. 4 in [a] showing that DCI 19.7, but BetaTCVAE DCI 35), and [b] requires beta-VAE objective function to work.
>
> Having said that, we do believe it is an interesting ablation study. Therefore, we conduct and compare it (i.e., being directly added to a vanilla VAE) with Lie group VAE (without Hessian penalty) and vanilla VAE on dsprites. From the table below we observe that our model works even on vanilla VAE and is comparable to the Lie group VAE (without hessian penalty), under the same experiment setting as Lie group VAE.
>
>
> | Models | DCI | MIG |
> | :-----:| :----: | :----: |
> | vanilla VAE | 0.081±0.041 | 0.078±0.064 |
> | Lie group VAE | 0.197±0.046 | **0.254±0.061** |
> | Groupified       | **0.221±0.042**  | 0.182± 0.041 |
>
> Note that the results of the Lie group VAE report here is the best result after tuning the hyperparameters (see Tab.3 in [a]). Due to the time limitation, we use the default parameters of groupified vanilla VAE.
>
> [a] Xinqi Zhu, Chang Xu, and Dacheng Tao. Commutative lie group vae for disentanglement learning. ICML, 2021.
>
> [b] Shao, Huajie, et al. "ControlVAE: Controllable variational autoencoder." ICML, 2020.
>
> [c] Peebles, William, et al. "The hessian penalty: A weak prior for unsupervised disentanglement." ECCV, 2020.
>
> [d] Rhodes, Travers, and Daniel D. Lee. "Local Disentanglement in Variational Auto-Encoders Using Jacobian $ L_1 $ Regularization." NeurIPS, 2021.
> [e] Ross, Andrew Slavin, and Finale Doshi-Velez. "Benchmarks, Algorithms, and Metrics for Hierarchical Disentanglement." ICML, 2021.
>
> [f] Wei, Yuxiang, et al. "Orthogonal Jacobian Regularization for Unsupervised Disentanglement in Image Generation." ICCV, 2021.
>
> [g] Chen, Ricky TQ, et al. "Isolating sources of disentanglement in variational autoencoders." NeurIPS, 2018.

---

> ### Author Response · Authors · 2021-11-14
> **Response to Reviewer YTd1 (Part 1)**
>
> Thanks for providing constructive comments. Your concerns are addressed as below. We are looking forward to further discussion.
>
> 1. The introduction of $\Phi$ group is unnecessarily complex.
>
> We regret to note that “$\Phi$ group is a direct product of m cyclic groups of order n” is a misunderstanding of $\Phi$. We would like to present a better way of understanding  $\Phi$ from why we introduce $\Phi$, and make further clarification about why we introduce those concepts.
>
> Please note that our definition of $\Phi$ is: 1) its elements are the process of encoding, action on latent space, and decoding. 2) the operation is function composition (As Fig. 1 shows). This group is determined/defined by the parameters of our VAE. In our paper, the group $G$ to which we constrain $\Phi$ to be isomorphic is m cyclic groups of order n. (which is enforced by the optimization of the proposed isomorphism loss). Therefore, in order to make the above clear, the introduction of $\Phi$ group is quite necessary.
>
> Although in group theory they are all isomorphic groups, we believe it is necessary to use these concepts in our paper for rigorousness and simplicity.
> - For the theoretical part, $Z$ is not a group (there is no operator), and we need to use 'congruence class modulo n' (same elements as in G) to define the elements in $Z$. We use 'element-wise addition' to define the group action $G$ on $Z$. In addition,  the "action of rotating each circle by step $2π/n$" is unfriendly to those readers unfamiliar with group theory (unlike 'element-wise addition', therefore here we use 'element-wise addition').
> - For the implementation part, we chose the n-th root unity group to instantiate these elements in $Z$ (whose elements are differentiable). The additive group of integers modulo n does not show up in the main manuscript. This name is only used to introduce $Z/nZ$ in the appendix.
>
> 2. The mention of 'n-th dihedral group' is not informative, or even misleading.
>
> We regret that you have a misunderstanding, and we have clarified it in the rebuttal version. Here we introduce Dihedral groups in an attempt to help readers better understand the relation between the elements of a group and disentangled factors.
> Specifically, by “inspiring by Dihedral groups,” we mean the rotation and flip can be analogies to the factors in disentanglement, and they are also analogies to the generators in our settings. In addition, flip and rotation are transformations and also the elements of the group. These facts inspire us to construct the group $\Phi$.
>
> 3. The Abel loss and Order loss does not solve the unidentifiability problem that motivates this method.
>
> We agree with you that it still falls into the unidentifiability problem proposed in Locatello et al. 2019b (Note that we never claim that we solved the problem), but we believe our theoretical framework and method have made a big step towards solving the problem from the perspective of group-based definition.  The reasons are twofold:
> - please note that the unidentifiability problem does not exist in the group-based definition. (Only identifiable solutions exist in the group-based definition)
> - Our theoretical framework presents three sufficient conditions for the definition. We proposed a solution that satisfied two of these conditions (model and structure) and met a necessary condition for the third one (data).
>
> In addition, the unidentifiability problem is believed to be quite challenging in unsupervised settings. Note that the recent progress of unsupervised disentanglement also did not solve the unidentifiability problem [a, b, d, e, f] and even not discuss it [a, f]. They all fall into the unidentifiability problem.
>
> 4. I don't think $\varphi \varphi^{−1}=e$ can replace $\varphi \varphi^{n−1}=e$, in the Order loss.
>
> Thanks for pointing out，the notation here is misleading. We have clarified this in the rebuttal version (highlighted with blue, please check). By using $\varphi^{-1}$, we intended to represent the process in Fig. 7 (b), which is not the inverse element but an approximation of $\varphi^{n-1}$. When the autoencoder can do the reconstruction well，this approximation holds.

---

> > ### Comment · Reviewer_YTd1 · 2021-11-19
> > **More Discussion**
> >
> > 1. About my ‘misunderstanding’: I understood $\phi$ is defined on data space. But since it relies on the autoencoder mapping, which is already commonly used in all autoencoder-based disentanglement models, the only component that distinguishes $\phi$ from existing models is the cyclic structure of the latent code, which is a product of cyclic groups, therefore I used ‘$\phi$ group is a direct product of m cyclic groups of order n’ in my comment.
> > The overcomplexity problem: I don’t remember the 'congruence class modulo n’ has been mentioned in [a]. What new things have been brought here by re-describing the cyclic group into the ‘integer modulo n’ group? I still think the authors should stick to the group description in [a] which is simple and easy to understand, instead of reintroducing this group again with 'congruence classes modulo n’ in Sec. 3.2, as it only causes me extra confusion though I am already familiar with the group definition in [a].
> >
> > 3. (1) Yes, I understood it well. The group-based definition directly used the correspondence between the ground-truth group structure (in W state space) and the representation. So technically this can only ‘describe’ how the ideal solution should look like but it does not provide any method to achieve it.
> > (2) After more detailed thinking, I want to make my point clearer.
> > The key to solving the unidentifiability in [b] is the condition (ii) in your proposed Theorem 1, which links the group $\phi$ acting on data to the ground-truth generators. However, this data constraint modeling in this paper is solely modeled by the total correlation minimization. The inductive biases introduced in this paper are the group structure constraint and model constraint, which I believe are valuable by themselves, but not related to the inductive biases meant by [b] as they cannot contribute to the unidentifiable problem. I believe it is totally fine the model in this paper cannot solve the unidentifiability problem which is as you said very challenging, but the motivation and discussion in this paper try to convey that this paper proposes ‘inductive biases’ that are wanted by [b] to solve the unidentifiability problem, but the actual ‘biases’ (group structure constraint and model constraint) are not related to this problem, while the only key point ‘data constraint’ that can solve this unidentifiability problem is modeled in the same way as usual methods by the total correlation minimization. Therefore I believe the authors should emphasize the benefits of the group structure constraint and model constraint themselves, rather than trying to describe them as the lacked inductive biases mentioned in [b].
> >
> > [a] Irina Higgins, David Amos, David Pfau, Sebastien Racaniere,Loic Matthey, Danilo Rezende, Alexander Lerchner. "Towards a Definition ofDisentangled Representations” 2018
> > [b] Francesco Locatello, Stefan Bauer, Mario Lucic, Gunnar Raetsch, Sylvain Gelly, Bernhard Scholkopf, ¨ and Olivier Bachem. Challenging common assumptions in the unsupervised learning of disentangled representations. In ICML, 2019.

---

> > > ### Author Response · Authors · 2021-11-19
> > > **Response to "More Discussion"**
> > >
> > > 1. We fully understand your response to "the misunderstanding", the statement makes sense in this context.
> > >
> > > 2. Thanks for pointing out that we should stick to the group description in [a]. With this description, we indeed can describe our idea without "congruence classes modulo n". We have removed "congruence classes modulo n" in our rebuttal version (highlighted with blue in Sec 3.2 & 4.1, please check).
> > >
> > > 3. Thanks for your detailed thinking, we fully agree with your insight and followed your suggestion that "we should emphasize the benefits of the group structure constraint and model constraint themselves, rather than trying to describe them as lacking inductive biases". We appreciate your insightful comments and suggestions that help a lot to make the point more precise.
> > >
> > > Specifically, the modifications include (highlighted with blue, please check):
> > > - We removed "congruence classes modulo n" in Sec 3.2 & 4.1.
> > > - We changed the claim (also other similar claims) that “... for the inductive bias these VAEs lack” into “ … for the additional inductive bias in existing VAE-based methods” in Introduction and also delete “existing VAE-based models only consider...” in Sec.4.
> > >
> > > - We add “The inductive bias encourages ...” in Introduction to emphasize the benefits (role) of the group structure constraint and model constraint.
> > > - We changed the claim that "However, it has been proved that ..." into "However, these models lack theoretical ..." in Abstract.
> > > - We changed the claim that "With the first two of the conditions satisfied ..." into "With the first two of the conditions satisfied ...as an additional inductive bias for ..." in Abstract.
> > >
> > > [a] Irina Higgins, David Amos, David Pfau, Sebastien Racaniere,Loic Matthey, Danilo Rezende, Alexander Lerchner. "Towards a Definition of disentangled Representations” 2018

---

> > > ### Author Response · Authors · 2021-11-22
> > > **Looking forward to hearing from you**
> > >
> > > Dear Reviewer YTd1,
> > >
> > > We want to send you a friendly reminder for the discussion, since the second stage of discussion will be soon concluded.
> > >
> > > We thank you again for your valuable comments, and we would appreciate it if you could reconsider the evaluation of our work based on our response. We are happy to extend our response if you have any other concerns left.
> > >
> > > Thanks.

---

> > > ### Author Response · Authors · 2021-11-23
> > > **Further response to "More Discussion"**
> > >
> > > Dear Reviewer YTd1:
> > >
> > > After further thinking about your concern, we rewrote substantially (highlighted with blue) to address your concern that “the motivation and discussion in this paper try to convey that this paper proposes ‘inductive biases’ that are wanted by [b] to solve the unidentifiability problem”. We believe that our modified version may address this “ambiguous problem” well.
> > >
> > > - Remove the ambiguous statements related to the unidentifiability problem in Abstract & Introduction
> > > - Add some clarifications on that our work still falls into the unidentifiability problem in Related Works.
> > > - Remove the paragraph before Sec. 4.1.
> > > - Add “we only provide … as future work” as our limitation in Conclusion.
> > >
> > > We want to highlight the value of our theoretical framework. “the key to solving the unidentifiability in [b] is the condition (ii)” also reflects the value of our theoretical framework. From the group-definition perspective, we make this point clearer by providing a theoretical guarantee from our proposed Theorem 1.

---

> > > > ### Comment · Reviewer_YTd1 · 2021-11-26
> > > > **Thanks for your response**
> > > >
> > > > Thanks for your response. I believe these modifications make the paper clearer. I will increase the score to 6 weak accept.

---

> > > > > ### Author Response · Authors · 2021-11-30
> > > > > **Thank you for updating the score!**
> > > > >
> > > > > Your feedback was extremely valuable and helped improve the paper substantially.

---

### Official Review · Reviewer_AKvy · 2021-11-02

**Correctness:** 4
**Technical Novelty And Significance:** 4
**Empirical Novelty And Significance:** 4
**Recommendation:** 8
**Confidence:** 4

**Main Review:**

Strength:

This paper appears to be the first in bringing and realizing the concept of group-based disentanglement in unsupervised VAE. The method was rigorously motivated and backed by theoretical analysis. The experiments were relatively thorough in terms of the datasets, metrics, and comparison disentanglement VAE methods considered. The improvement of metrics, while moderate to marginal in some cases, were consistent across most metrics and datasets.

The visuals of the cyclic representation shown in section 5.2 and Fig 5 are especially interesting, as well the latent space shown in Fig 3 demonstrating the benefit of the isomorphism loss.

Weakness:

Some of the metrics seem to be weaker than usually reported in the comparison methods. E.g., The MIG metrics reported for dSprites seemed to be rather low for each of the "original" VAE methods compared to those seen in literature (e.g., Fig 3 in [1]). Please clarify, especially since the margin of improvements in many metrics were less than 1 standard deviation of the statistics.

While it is appreciated that the presented work considers a "unsupervised" learning of representations in comparison to existing works that uses the interaction with the environment as supervision to adopt the group-based definition, it would be still desirable to see a comparison of performance to understand what is the price to pay to go from a supervised to unsupervised setting. This is somewhat related to the comment above regarding avoiding using weak baselines.

In the final manuscript, please adjust the figures and tables such that they 1) appear in the order they're referred to in the text and 2) are spaced as close to the text description as possible. Right now Fig 4 and Fig 5 are referred to first, before Fig 3 and 2 are discussed in the text.


**Summary Of The Paper:**

This paper presents an unsupervised approach to achieve group-based disentangled representation learning, as opposed to existing environment-based approaches. A theoretical framework for the group-based VAE was proposed, and implementations of the group and isomorphism reported. Experiments were performed on four common benchmark datasets, with comparisons to several representative VAE-based disentanglement framework (beta-vae, annealVAE, factorVAE, beta-TCVAE), demonstrating the groupified VAE achieves better disentanglement in many metrics. Qualitative evaluations demonstrate the cyclic representation space learned by the groupified VAE.

**Summary Of The Review:**

This is an overall well written manuscript describing a novel method backed by in-depth theoretical analysis. The experiments were well designed and evaluations relatively thorough. There are some questions that can be addressed and improvements that can be made to the current manuscript, but overall I consider this to be an interesting work that will have good discussion potential for the community.

---

> ### Author Response · Authors · 2021-11-14
> **Response to Reviewer AKvy**
>
> Thanks for your extremely valuable comments. We appreciate the positive feedback for our work and are happy to include the suggested changes into the rebuttal version. We are looking forward to further discussion.
>
> 1. The MIG metrics reported for dSprites seemed to be rather low for each of the "original" VAE methods compared to those seen in literature
>
> The settings of our experiments are different from [a] (or other baselines). We follow [b] (The baseline performance is in Figure 13 of [b] appendix) to conduct experiments and verify our method taking into account the impact of hyperparameters and random seeds. This evaluation setting is well accepted in the literature (e.g.,Fig. 5 in [b], Fig.3 in [c]). Our method is effective across different hyperparameters and random seeds and better than other methods, including [a].
>
> We found out that the value of MIG of $\beta$-TCVAE is incorrect. The values here are 0.15±0.067 (original) and 0.21±0.093 (groupified), we have updated the values in the rebuttal version (Highlighted with blue, please check).
>
> 2. A comparison of performance to understand what is the price to pay to go from a supervised to unsupervised setting.
>
> Thanks for this interesting suggestion. We fully agree with your insight and will add such an experiment in the final version later, as we finalize the experiments.
>
> 3. Adjust the figures and tables.
>
> We have followed your suggestions in the rebuttal version (Please check).
>
> [a] Chen, Ricky TQ, et al. "Isolating sources of disentanglement in variational autoencoders." NeurIPS 2018.
>
> [b] Locatello, Francesco, et al. "Challenging common assumptions in the unsupervised learning of disentangled representations." ICML, 2019.
>
> [c] Rhodes, Travers, and Daniel D. Lee. "Local Disentanglement in Variational Auto-Encoders Using Jacobian $ L_1 $ Regularization." NeurIPS 2021.
>
> [d] Hälvä, Hermanni, et al. "Disentangling Identifiable Features from Noisy Data with Structured Nonlinear ICA." NeurIPS 2021.
>
> If the reference [1] in your comment is not the reference [a] here, please let us know.

---

> > ### Comment · Reviewer_AKvy · 2021-11-22
> > **Post-discussion comments**
> >
> > Thanks for your response.
> >
> > Thanks for clarifying on the discrepancy between the results and some published works may be due to the fact that the results were reported across multiple hyperparameters and random seeds. I did look into [b] and feel that there were still some difference in the reported numbers between Table 2 of the presented manuscript and those in [b], e.g., the numbers reported on Car3D in Fig 3 of [b] and dSprits and 3DShape in Fig 13 of [b]. I understand that such fluctuation may be due to the use of different hyperparameters and random seeds, which highlights the importance of a better way of presenting the "improvement" obtained by the groupified methods in this paper -- especially since the improvement was in most cases within 1 std away from the mean. If the results were highly overlapping and the improvement was less than the fluctuation itself, it'd not be very convincing.
> >
> > In such cases, it may be more beneficial if the authors could control the factors causing fluctuation and then demonstrate the improvement of the presented method, e.g., comparing differences at different levels of regularization parameters.
> >
> > I also carefully read through the discussion among the authors and the other reviewers. While I personally are in favor of this manuscript, I understood the concerns of the other reviewers and could not strongly vouch against their concerns, especially on the points of inductive biases and the intuition of isomorphism loss.

---

> > > ### Author Response · Authors · 2021-11-23
> > > **Response to Reviewer AKvy**
> > >
> > > Thank you for your response, and constructive suggestion. Due to the time limitation, we would like to follow your suggestion to present our results in final version.

---

> ### Author Response · Authors · 2021-11-22
> **Looking forward to hearing from you**
>
> Dear Reviewer AKvy,
>
> We want to send you a friendly reminder for the discussion, since the second stage of discussion will be soon concluded.
>
> We thank you again for your valuable comments, and we are happy to extend our response if you have any other concerns left.
>
> Thanks.

---

### Author Response · Authors · 2021-11-15
**Looking forward to the discussion**

Dear all reviewers,

We are looking forward to further discussion.

Best,

Paper10 Authors

---

### Decision · Program_Chairs · 2022-01-20

**Decision:**

Accept (Poster)

**Comment:**

The submission provides a theoretical framework on the learning of group-based disentanglement representations and proposes a novel method to learn such representations.

The reviewers appreciated the novel perspective of the paper in introducing the concept of group-based disentanglement in unsupervised VAE. Furthermore, the approach was considered to be soundly theoretically motivated and experiments to be extensive. There was a lively discussion between reviewers and authors about certain ambiguities in the manuscript; however, they seem to have been largely resolved to the reviewers' satisfaction.

While there was a reviewer with a very low confidence recommending rejection, this paper brings an indisputably interesting novel perspective to the learning of unsupervised representations and I thus recommend acceptance.